# Adv-SSL: Adversarial Self-Supervised Representation Learning with Theoretical Guarantees

**Chenguang Duan**[1]    **Yuling Jiao**[2,3,4]    **Huazhen Lin**[5,6,7,8*]
**Wensen Ma**[1]    **Jerry Zhijian Yang**[8,3,1,4]

[1]School of Mathematics and Statistics, Wuhan University
[2]School of Artificial Intelligence, Wuhan University
[3]National Center for Applied Mathematics in Hubei, Wuhan University
[4]Hubei Key Laboratory of Computational Science, Wuhan University
[5]Center of Statistical Research, Southwestern University of Finance and Economics
[6]School of Statistics and Data Science, Southwestern University of Finance and Economics
[7]New Cornerstone Science Laboratory, Southwestern University of Finance and Economics
[8]Institute for Math & AI, Wuhan University
{cgduan.math, yulingjiaomath, vincen, zjyang.math}@whu.edu.cn
linhz@swufe.edu.cn

## Abstract

Learning transferable data representations from abundant unlabeled data remains a central challenge in machine learning. Although numerous self-supervised learning methods have been proposed to address this challenge, a significant class of these approaches aligns the covariance or correlation matrix with the identity matrix. Despite impressive performance across various downstream tasks, these methods often suffer from biased sample risk, leading to substantial optimization shifts in mini-batch settings and complicating theoretical analysis. In this paper, we introduce a novel **Adv**ersarial **S**elf-**S**upervised Representation **L**earning (Adv-SSL) for unbiased transfer learning with no additional cost compared to its biased counterparts. Our approach not only outperforms the existing methods across multiple benchmark datasets but is also supported by comprehensive end-to-end theoretical guarantees. Our analysis reveals that the minimax optimization in Adv-SSL encourages representations to form well-separated clusters in the embedding space, provided there is sufficient upstream unlabeled data. As a result, our method achieves strong classification performance even with limited downstream labels, shedding new light on few-shot learning.

## 1 Introduction

Collecting unlabeled data is considerably more convenient and cost-effective than gathering labeled data in real-world applications. Representations learned from such abundant data can be effectively transferred to various downstream tasks, thereby enhancing model performance or reducing the amount of labeled data required. Consequently, learning representations from abundant unlabeled data is both highly valuable and challenging.

Recently, self-supervised contrastive learning has emerged as a leading approach for learning representations from unlabeled data. This method aims to produce representations that are invariant to data augmentation. However, solely minimizing the distance between similar pairs can result in

---

*Corresponding author

trivial solutions, known as model collapse. To address this issue, researchers have proposed various strategies, which can be broadly categorized into three types.

The first strategy treats augmented views from different images as negative pairs, ensuring that their representations remain dissimilar [36, 23, 8, 9, 21, 38]. However, these methods require large batch sizes to provide sufficient negative samples, resulting in substantial computational and memory demands that may be prohibitive in many applications. Additionally, by treating augmented views from different images as negative pairs, these approaches overlook semantic similarities between distinct images, potentially forcing apart representations of conceptually related content. As noted by [12, 11], this design can degrade representation performance.

The second strategy prevents model collapse through asymmetric network architectures [19, 10, 6, 7]. Although these methods eliminate the need for negative pairs, they exhibit significant sensitivity to architectural design choices, where minor modifications can lead to collapsed solutions [19, 10]. Furthermore, these approaches introduce considerable challenges for interpretability.

The third line of work prevents model collapse by introducing a regularization term that aligns the covariance or correlation matrix with the identity matrix [37, 14, 4, 22, 3, 20, 24, 39], thereby encouraging the separation of class centers. These methods do not require negative samples and also offer clearer theoretical interpretability. A typical regularization term employed in these approaches [37, 22, 20, 24] is formulated as:

$$\mathcal{R}(f) = \left\| \mathbb{E}_{\boldsymbol{x}_s \sim \mathbb{P}_s} \mathbb{E}_{\mathbf{x}_{s,1}, \mathbf{x}_{s,2} \in \mathcal{A}(\boldsymbol{x}_s)} \left\{ f(\mathbf{x}_{s,1}) f(\mathbf{x}_{s,2})^\top \right\} - I_{d^*} \right\|_F^2, \tag{1}$$

where $f : \mathbb{R}^d \to \mathbb{R}^{d^*}$ denotes a representation mapping from the original image space to the representation space, $\| \cdot \|_F$ is the Frobenius norm, $\boldsymbol{x}_s$ represents an original image following a distribution $\mathbb{P}_s$, and $\mathcal{A}(\boldsymbol{x}_s)$ denotes the collection of all possible augmented views yielded from $\boldsymbol{x}_s$. The terms $\mathbf{x}_{s,1}, \mathbf{x}_{s,2} \in \mathcal{A}(\boldsymbol{x}_s)$ refer to two augmented views independently sampled from the uniform distribution on $\mathcal{A}(\boldsymbol{x}_s)$, while $I_{d^*}$ is a $d^* \times d^*$ identity matrix.

The population risk defined in equation (1) is typically intractable. In [22, 20, 37, 24], the researchers estimate (1) using the following sample-level regularization:

$$\widehat{\mathcal{R}}(f) = \left\| \frac{1}{n_s} \sum_{i=1}^{n_s} f(\mathbf{x}_{s,1}^{(i)}) f(\mathbf{x}_{s,2}^{(i)})^\top - I_{d^*} \right\|_F^2, \tag{2}$$

where $\{\boldsymbol{x}_s^{(i)}\}_{i \in [n_s]}$ denotes the original dataset, and $\widetilde{D}_s = \{(\mathbf{x}_{s,1}^{(i)}, \mathbf{x}_{s,2}^{(i)}) \in \mathcal{A}(\boldsymbol{x}_s^{(i)})\}_{i \in [n_s]}$ represents the augmented dataset for learning representations. Unfortunately, it is evident that $\widehat{\mathcal{R}}(f)$ is a biased estimator of $\mathcal{R}(f)$, i.e., $\mathbb{E}_{\widetilde{D}_s}\{\widehat{\mathcal{R}}(f)\} \neq \mathcal{R}(f)$, due to the non-commutativity between the expectation and the Frobenius norm. This inherent bias gives rise to two significant challenges.

Firstly, the biased estimator (2) used in [22, 20, 37] introduces significant optimization deviations during training. Although theoretically $\mathbb{E}_{\widetilde{D}_s}\{\widehat{\mathcal{R}}(f)\}$ converges to $\mathcal{R}(f)$ as $n$ approaches infinity, practical memory constraints necessitate the use of mini-batch samples. As a result, the bias introduces an offset in the optimization direction at each iteration. Moreover, this offset can accumulate over successive training steps, since each gradient direction depends on the previous one. Ultimately, this compounding effect may cause the learned representation to diverge significantly from the true minimizer of the population risk, thereby impairing practical performance, as demonstrated in Table 1.

Secondly, this inherent bias presents significant obstacles to establishing end-to-end theoretical guarantees, which is crucial for addressing several fundamental questions: *How quickly does the downstream task error converge with respect to both the number of unlabeled samples in the source domain and the number of labeled samples in the target domain? What is the mechanism by which unlabeled data in self-supervised learning contributes to downstream task performance? Why do self-supervised learning methods remain effective even when downstream labeled data is limited?*

Although recent theoretical studies have significantly advanced the understanding of self-supervised learning, several issues remain unresolved. These studies can be broadly categorized into two main lines of research. The first line [15, 22, 2, 24] focuses on analyzing the population risk of self-supervised learning methods. However, fundamental questions remain incompletely addressed due to the lack of discussion at the sample level. A comprehensive theoretical analysis requires bridging

the gap between population-level (1) and sample-level (2) risks, which is challenging due to the bias inherent in methods such as [37, 22, 20].

A second line of theoretical research analyzes generalization error using Rademacher complexity [31, 21, 1, 26, 20], but frequently overlooks the approximation error. This omission is critical, as overall learning performance is determined by the total error, which is the sum of both generalization and approximation errors. Consequently, by focusing on only one component, these analyses may provide an incomplete picture of model performance.

In this study, we propose a novel self-supervised learning framework, **Adv**ersarial **S**elf-**S**upervised Representation **L**earning (Adv-SSL). Adv-SSL introduces an innovative iterative scheme that eliminates the bias between the population risk (1) and its sample-level estimator (2), thereby addressing two critical challenges: training deviation and theoretical limitations caused by bias. Through comprehensive end-to-end analysis, we demonstrate how the amount of unlabeled data in the self-supervised pre-training phase enhances downstream task performance. Specifically, we show that representation learning with Adv-SSL enables downstream data to be effectively clustered by category in the representation space, provided that the upstream unlabeled sample size is sufficiently large. As a result, Adv-SSL achieves outstanding classification performance even with only a few downstream labeled samples, offering valuable insights for few-shot learning.

## 1.1 Contributions

Our main contributions can be summarized as follows:

- We introduce Adv-SSL, a novel unbiased self-supervised transfer learning method. This approach learns representations from unlabeled data by solving a min-max optimization problem that corrects the bias inherent in existing methods [22, 37]. Through extensive experiments, we demonstrate that Adv-SSL significantly outperforms previous biased sample risk (Table 1), as well as several existing self-supervised learning approaches (Table 3).

- We establish comprehensive end-to-end theoretical guarantees for Adv-SSL in transfer learning scenarios under misspecified setting (Theorem 1). Our theoretical analysis shows that representations learned by Adv-SSL, through minimax optimization, enable downstream data to be clustered by category in the representation space, provided that the upstream unlabeled sample size is sufficiently large. Consequently, Adv-SSL achieves outstanding classification performance even with only a few downstream labeled samples, offering valuable insights for few-shot learning.

## 1.2 Preliminaries

Given an integer $n \in \mathbb{N}$, we use $[n]$ to represent the integer set $\{1, 2, \cdots, n\}$. For any vector $\boldsymbol{x}$, we denote $\|\boldsymbol{x}\|_2$ and $\|\boldsymbol{x}\|_\infty$ as the 2-norm and $\infty$-norm of $\boldsymbol{x}$ respectively. Let $A, B \in \mathbb{R}^{d_1 \times d_2}$ be two matrices, we define their Frobenius inner product by $\langle A, B \rangle_F = \mathrm{tr}(A^\top B)$. Moreover, we denote $\|A\|_F$ as the Frobenius norm of $A$, which is the norm induced by Frobenius inner product, and $\|A\|_\infty = \sup_{\|\boldsymbol{x}\|_\infty \leq 1} \|A\boldsymbol{x}\|_\infty$ as the $\infty$-norm of $A$, which is the maximum 1-norm of the rows of $A$. For a vector-valued map $f$, we adopt $\mathrm{dom}(f)$ to represent its domain. Further, given $0 \leq a_1 \leq a_2$, we use $a_1 \leq \|f\|_2 \leq a_2$ to denote $a_1 \leq \inf_{\boldsymbol{x} \in \mathrm{dom}(f)} \|f(\boldsymbol{x})\|_2 \leq \sup_{\boldsymbol{x} \in \mathrm{dom}(f)} \|f(\boldsymbol{x})\|_2 \leq a_2$. Besides that, the Lipschitz norm of $f$ is given by $\|f\|_{\mathrm{Lip}} = \sup_{\boldsymbol{x} \neq \boldsymbol{y}} \frac{\|f(\boldsymbol{x}) - f(\boldsymbol{y})\|_2}{\|\boldsymbol{x} - \boldsymbol{y}\|_2}$. Additionally, we use $f \in \mathrm{Lip}(L)$ to represent $\|f\|_{\mathrm{Lip}} \leq L$. Finally, if $X$ and $Y$ are two quantities, for ease of presentation, we employ $X \lesssim Y$ or $Y \gtrsim X$ to indicate the statement that $X \leq CY$ for some $C > 0$ and denote $X \asymp Y$ when $X \lesssim Y \lesssim X$ throughout this paper.

We subsequently adopt the following neural networks as the hypothesis space.

**Definition 1** (ReLU neural networks). Let $d_1, d_2 \in \mathbb{N}$ and $L, N_1, \ldots, N_L \in \mathbb{N}$. A ReLU neural network with depth $L$ and width $W := \max\{N_1, \ldots, N_L\}$ has the following form:

$$f_{\boldsymbol{\theta}}(\boldsymbol{x}) = A_L \sigma(A_{L-1} \sigma(\cdots \sigma(A_0 \boldsymbol{x} + b_0)) + b_{L-1}), \tag{NN}$$

where $A_i \in \mathbb{R}^{N_{i+1} \times N_i}$, $b_i \in \mathbb{R}^{N_{i+1}}$, and $\sigma(\cdot)$ is the element-wise ReLU activation function. Denote by $\boldsymbol{\theta} := ((A_0, b_0), \ldots, (A_{L-1}, b_{L-1}), A_L)$ the collection of parameters of the neural network (NN).

Furthermore, define $\kappa(\boldsymbol{\theta}) = \|A_L\|_\infty \prod_{l=0}^{L-1} \max\{\|(A_l, b_l)\|_\infty, 1\}$. For $\mathcal{K} > 0$, it follows from Appendix G.1 that $\|f_{\boldsymbol{\theta}}\|_{\mathrm{Lip}} \leq \mathcal{K}$, provided that $\kappa(\boldsymbol{\theta}) \leq \mathcal{K}$.

Let $\mathcal{K} > 0$ and $0 < B_1 < B_2$, a ReLU network class $\mathcal{NN}_{d_1, d_2}(W, L, \mathcal{K}, B_1, B_2)$ is defined as

$$\left\{ f_{\boldsymbol{\theta}} \text{ has the form (NN)}: N_0 = d_1,\ N_{L+1} = d_2,\ \kappa(\boldsymbol{\theta}) \leq \mathcal{K},\ B_1 \leq \|f_{\boldsymbol{\theta}}\|_2 \leq B_2 \right\}.$$

Finally, for two given measures $\mu$ and $\nu$, we define the 1-Wasserstein distance as $\mathcal{W}(\mu, \nu) := \max_{g \in \mathrm{Lip}(1)} \mathbb{E}_{X \sim \mu}\{g(X)\} - \mathbb{E}_{Y \sim \nu}\{g(Y)\}$.

## 1.3 Organization

The rest of this paper is structured as follows: Section 2 introduces the core concept of Adv-SSL and presents our alternating optimization algorithm. In Section 3, we develop a comprehensive end-to-end theoretical guarantee for Adv-SSL with proof details in Section F. Section 4 demonstrates Adv-SSL's effectiveness through extensive experimental evaluations across diverse datasets and metrics. Section 5 summarizes the conclusions of this work.

The appendices provide a review of existing studies (Appendix A), a notation summary (Appendix B), experimental details (Appendix C), additional numerical experiments (Appendix D), discussions on assumptions (Appendix E), and complete theoretical proofs (Appendices F to G).

## 2 Adversarial Self-Supervised Representation Learning

### 2.1 Notations

Throughout this paper, we use $d$ and $d^*$ to represent the dimensions of the original image space and the representation space, respectively. We use the letter $\boldsymbol{x}_s$ and its variants to denote image instances from the source domain $\mathcal{X}_s \subseteq [0, 1]^d$ with the source distribution $\mathbb{P}_s$. Correspondingly, we use the letter $\boldsymbol{x}_t$ and its variants for image instances from the target domain $\mathcal{X}_t \subseteq [0, 1]^d$, while $\mathbb{P}_t$ represents the measure regarding the entry $(\boldsymbol{x}_t, y)$ with label $y \in [K]$. In this context, we can independently and identically sample a total of $n_s$ source image instances from $\mathbb{P}_s$ and $n_t$ labeled downstream samples from $\mathbb{P}_t$, and refer to them as $D_s = \{\boldsymbol{x}_s^{(i)}\}_{i \in [n_s]}$ and $D_t = \{(\boldsymbol{x}_t^{(i)}, y_i)\}_{i \in [n_t]}$, respectively.

Since the primary objective of contrastive learning is to learn a representation that is invariant to different augmentations, data augmentation plays a crucial role in this field. A augmentation $A : \mathbb{R}^d \to \mathbb{R}^d$ is essentially a predefined transformation applied to original images. Common augmentations include the composition of random transformations, such as RandomCrop, HorizontalFlip, and Color Distortion [8]. We refer to $\mathcal{A} = \{A_i(\cdot)\}_{i \in [m]}$ as the collection of used data augmentations, where $m$ is the total number of data augmentations, which is finite since only a finite number of augmentations will be used in practice. Based on it, we can construct an augmented dataset $\widetilde{D}_s = \{\tilde{\mathbf{x}}_s^{(i)}\}_{i \in [n_s]}$, where $\tilde{\mathbf{x}}_s^{(i)} = (\mathbf{x}_{s,1}^{(i)}, \mathbf{x}_{s,2}^{(i)}) = (A_{i,1}(\boldsymbol{x}_s^{(i)}), A_{i,2}(\boldsymbol{x}_s^{(i)}))$, and $A_{i,1}$ and $A_{i,2}$ are independently drawn from the uniform distribution on $\mathcal{A}$. The Appendix B summarizes the notations used throughout this work for easy reference and cross-checking.

### 2.2 Adversarial self-supervised learning

The regularization term $\mathcal{R}(f)$ defined in (1) has been adopted in various studies [22, 20, 24] to prevent model collapse. Specifically, we aim to identify an encoder that is as close as possible to $f^*$:

$$f^* \in \underset{f: B_1 \leq \|f\|_2 \leq B_2}{\arg\min} \mathcal{L}(f) = \mathcal{L}_{\mathrm{align}}(f) + \lambda \mathcal{R}(f),$$

$$\mathcal{L}_{\mathrm{align}}(f) = \mathbb{E}_{\boldsymbol{x}_s \sim \mathbb{P}_s} \mathbb{E}_{\mathbf{x}_{s,1}, \mathbf{x}_{s,2} \in \mathcal{A}(\boldsymbol{x}_s)} \left\{ \left\| f(\mathbf{x}_{s,1}) - f(\mathbf{x}_{s,2}) \right\|_2^2 \right\}.$$

Intuitively, imposing the constraint $B_1 \leq \|f\|_2 \leq B_2$ does not impair encoder performance, as the key aspect of data representation is the ability to distinguish between features rather than the scale of their values. As we will demonstrate, this constraint actually can actually facilitate the theoretical analysis of Adv-SSL.

Since the expectation in this regularization term is challenging to compute practically, it is necessary to approximate it using an empirical average based on the collected samples. One of the most commonly-used empirical versions is $\widehat{\mathcal{R}}(f)$ defined in (2) [22, 20]. However, as stated in Section 1, $\widehat{\mathcal{R}}(f)$ is a biased estimation of $\mathcal{R}(f)$, i.e., $\mathbb{E}_{\widetilde{D}_s}\{\widehat{\mathcal{R}}(f)\} \neq \mathcal{R}(f)$, which introduces optimization deviation from $f^*$ and hinders establishing a theoretical understanding of the empirical risk minimizer.

To address these issues, we propose a novel *unbiased* sample-level estimator for the population risk (1). A key observation that motivates Adv-SSL is that we can rewrite $\mathcal{R}(f)$ as

$$\mathcal{R}(f) = \sup_{G \in \mathcal{G}(f)} \mathcal{R}(f, G), \tag{3}$$

where $G \in \mathbb{R}^{d^* \times d^*}$ is a matrix variable, and

$$\mathcal{R}(f, G) = \left\langle \mathbb{E}_{\boldsymbol{x} \sim \mathbb{P}_s} \mathbb{E}_{\mathbf{x}_{s,1}, \mathbf{x}_{s,2} \in \mathcal{A}(\boldsymbol{x}_s)} \{ f(\mathbf{x}_{s,1}) f(\mathbf{x}_{s,2})^\top \} - I_{d^*}, G \right\rangle_F,$$
$$\mathcal{G}(f) = \{ G \in \mathbb{R}^{d^* \times d^*} : \|G\|_F \leq \sqrt{\mathcal{R}(f)} \}.$$

The equation (3) holds because of the fact that $\langle A, B \rangle_F \leq \|A\|_F \|B\|_F$ for any matrices $A, B$ of same dimension, with equality holding if and only if $A = B$. Correspondingly, its sample-level counterpart defined in (2) can be rewritten as

$$\widehat{\mathcal{R}}(f) = \sup_{G \in \widehat{\mathcal{G}}(f)} \widehat{\mathcal{R}}(f, G),$$

where

$$\widehat{\mathcal{R}}(f, G) = \left\langle \frac{1}{n_s} \sum_{i=1}^{n_s} f(\mathbf{x}_{s,1}^{(i)}) f(\mathbf{x}_{s,2}^{(i)})^\top - I_{d^*}, G \right\rangle_F,$$
$$\widehat{\mathcal{G}}(f) = \left\{ G \in \mathbb{R}^{d^* \times d^*} : \|G\|_F \leq \sqrt{\widehat{\mathcal{R}}(f)} \right\}.$$

It can be shown that $\widehat{\mathcal{R}}(\cdot, \cdot)$ is an unbiased estimator of the population risk $\mathcal{R}(\cdot, \cdot)$, that is, for each fixed $f$ and auxiliary variable $G$,

$$\mathcal{R}(f, G) = \mathbb{E}_{\widetilde{D}_s}\{\widehat{\mathcal{R}}(f, G)\}.$$

Hence, the equivalent transformation (3) help us avoid the issue introduced by bias. Specifically, with the equation (3) and its empirical version, we learn a representation through Adv-SSL at the sample level, which can be formulated as a mini-max problem as follows:

$$\min_{f \in \mathcal{F}} \max_{G \in \widehat{\mathcal{G}}(f)} \widehat{\mathcal{L}}(f, G) = \widehat{\mathcal{L}}_{\text{align}}(f) + \lambda \widehat{\mathcal{R}}(f, G),$$
$$\widehat{\mathcal{L}}_{\text{align}}(f) = \frac{1}{n_s} \sum_{i=1}^{n_s} \left\| f(\mathbf{x}_{s,1}^{(i)}) - f(\mathbf{x}_{s,2}^{(i)}) \right\|_2^2,$$

where $\mathcal{F}$ is defined as $\mathcal{NN}(W, L, \mathcal{K}, B_1, B_2)$. We will specify the appropriate parameters $(W, L, \mathcal{K}, B_1, B_2)$ to satisfy the theoretical requirements in Section 3. The term $\widehat{\mathcal{L}}_{\text{align}}(f)$ embodies the core idea of contrastive learning: learning a representation that is invariant to different augmentations. Additionally, $\lambda > 0$ serves as the regularization hyperparameter.

This mini-max problem naturally suggests solving it via an alternating optimization algorithm, in which $G$ is held fixed during the optimization of the encoder $f$, and $f$ is held fixed during the optimization of $G$. This procedure is detailed in Algorithm 1.

*Remark* 1 (Detach technique). It is important to note that $G_\tau$ in Algorithm 1 has been detached from the computational graph when updating the encoder parameters $\boldsymbol{\theta}$, which implies that the gradient regarding $\boldsymbol{\theta}$ is given by the Step 8 of Algorithm 1, rather than $\nabla_{\boldsymbol{\theta}} \| \frac{1}{N} \sum_{i=1}^N f_{\boldsymbol{\theta}}(\mathbf{x}_{s,1}^{(n_i^{(\tau)})}) f_{\boldsymbol{\theta}}(\mathbf{x}_{s,2}^{(n_i^{(\tau)})})^\top - I_{d^*} \|_F^2$, which is the mini-batch gradient of $\widehat{\mathcal{R}}(f)$. In this regard, *such a mini-max iteration format will yield a distinctly different encoder in the mini-batch scenario compared to previous studies* [37, 22].

**Algorithm 1** Alternative Optimization Algorithm

---

**Require:** Unlabeled dataset $D_s = \{x_s^{(i)}\}_{i \in [n_s]}$, initial encoder parameter $\theta_0$, iteration horizon $T$, mini-batch size $N$, learning rate $\eta$.

1: Construct an augmented dataset $\widetilde{D}_s = \{\tilde{x}_s^{(i)}\}_{i \in [n_s]}$.

2: **for** $\tau \in \{0\} \cup [T-1]$ **do**

3:      Sample a mini-batch $\mathcal{B}_\tau = \{\tilde{x}_s^{(n_i^{(\tau)})}\}_{i \in [N]} \subseteq D_s$ of size $N$, where $n_i^{(\tau)}$ represents the index of the $i$-th sample in the mini-batch $\mathcal{B}_\tau$ within $D_s$.

4:      **if** $\tau = 0$ **then**

5:          $G_0 = \sum_{i=1}^N f_{\theta_0}(x_{s,1}^{(n_i^{(\tau)})}) f_{\theta_0}(x_{s,2}^{(n_i^{(\tau)})})^\top - I_{d^*}$.

6:          Detach: $G_0 \leftarrow G_0.\text{detach}()$.

7:      **end if**

8:      Update encoder $\theta_{\tau+1} = \theta_\tau - \eta \Delta_\theta$, where $\Delta_\theta$ is given by

$$
\Delta_\theta = \nabla_\theta \frac{1}{N} \sum_{i=1}^N \left\| f_\theta(x_{s,1}^{(n_i^{(\tau)})}) - f_\theta(x_{s,2}^{(n_i^{(\tau)})}) \right\|_2^2 + \left\langle \nabla_\theta \frac{1}{N} \sum_{i=1}^N f_\theta(x_{s,1}^{(n_i^{(\tau)})}) f_\theta(x_{s,2}^{(n_i^{(\tau)})})^\top - I_{d^*}, G_\tau \right\rangle_F .
$$

9:      $G_{\tau+1} = \sum_{i=1}^N f_{\theta_{\tau+1}}(x_{s,1}^{(n_i^{(\tau)})}) f_{\theta_{\tau+1}}(x_{s,2}^{(n_i^{(\tau)})})^\top - I_{d^*}$.

10:      Detach: $G_{\tau+1} \leftarrow G_{\tau+1}.\text{detach}()$.

11: **end for**

12: **return** The learned encoder $f_{\theta_T}$.

---

The natural question that arises is *whether this adversarial iteration format will lead to better performance?*

To answer this question, we compare Adv-SSL against two biased self-supervised learning methods: Barlow Twins [37] and the approach proposed by [22], across multiple benchmark datasets. The experimental results, summarized in Table 1, demonstrate that Adv-SSL significantly improves downstream classification accuracy compared to both baseline methods, which are implemented using our repository, with a total training of 1000 epochs and a representation dimension of 512. It worth mentioning that our results are close to those reported in the well-known Python package LightlySSL, suggesting our results align with expectations.

| Method | CIFAR-10 | | CIFAR-100 | | Tiny ImageNet | |
| --- | --- | --- | --- | --- | --- | --- |
| | Linear | $k$-nn | Linear | $k$-nn | Linear | $k$-nn |
| Barlow Twins[37] | 87.32 | 84.74 | 55.88 | 46.41 | 41.52 | 27.00 |
| Beyond Separability[22] | 86.95 | 82.04 | 56.48 | 48.62 | 41.04 | 31.58 |
| Adv-SSL | **93.01** | **90.97** | **68.94** | **58.50** | **50.21** | **37.40** |

Table 1: Top-1 Accuracy Comparison with Biased SSL Methods.

The experimental details can be found in Appendix C. In addition, more ablation studies are deferred to Appendices D.1, D.2, D.3 and D.4, which respectively involve the choice of the regularization parameter $\lambda$, the impact of data augmentations, the influence of the alignment term and effectiveness in terms of transfer learning.

Furthermore, it naturally raises a question of *whether the minimax iteration in Adv-SSL incurs any additional training cost?* Intuitively, the extra cost from adversarial updates is negligible, as the inner maximization problem admits an analytical solution, as shown in Step 9 of Algorithm 1. To further support this view, we provide a detailed comparison of the timing and memory costs in Table 2.

All experiments were conducted on a single Tesla V100 GPU. The time mentioned refers to the training time spent per epoch. As we seen, this observation aligns well with our intuition.

| Method | CIFAR-10 | | CIFAR-100 | | Tiny ImageNet | |
|---|---|---|---|---|---|---|
| | Memory | Time | Memory | Time | Memory | Time |
| Barlow Twins | 5598 MiB | 68s | 5598 MiB | 74s | 8307 MiB | 386s |
| Adv-SSL | **5585 MiB** | **51s** | **5585 MiB** | **52s** | **8282 MiB** | **352s** |

Table 2: Comparison of Training Memory and Time Costs Between Barlow Twins and Adv-SSL.

## 3 End-to-End Theoretical Guarantee

### 3.1 Problem formulation

We first define $\hat{f}_{n_s}$ as the empirical risk minimizer for Adv-SSL as (4) and hope to establish a rigorous theoretical guarantee for that.

$$\hat{f}_{n_s} \in \arg\min_{f \in \mathcal{F}} \max_{G \in \widehat{\mathcal{G}}(f)} \widehat{\mathcal{L}}(f, G) = \widehat{\mathcal{L}}_{\text{align}}(f) + \lambda \widehat{\mathcal{R}}(f, G) \tag{4}$$

Moreover, following the similar process to that used for obtaining $\widetilde{D}_s$, we can construct the downstream augmented dataset $\widetilde{D}_t = \{(\tilde{\mathbf{x}}_t^{(i)}, y_i)\}_{i \in [n_t]}$, where $\tilde{\mathbf{x}}_t^{(i)} = (\mathbf{x}_{t,1}^{(i)}, \mathbf{x}_{t,2}^{(i)}) \in \mathbb{R}^{2d}$ with $\mathbf{x}_{t,1}^{(i)} = A_{i,1}(\boldsymbol{x}_t^{(i)})$, $\mathbf{x}_{t,2}^{(i)} = A_{i,2}(\boldsymbol{x}_t^{(i)})$. Therein, $A_{i,1}$, $A_{i,2}$ are independently and identically distributed samples drawn from the uniform distribution defined on $\mathcal{A}$. In this context, for a testing sample $\boldsymbol{x}$, we construct the following linear probe as a classifier:

$$Q_{\hat{f}_{n_s}}(\boldsymbol{x}) = \arg\max_{k \in [K]} \left( \widehat{W} \hat{f}_{n_s}(\boldsymbol{x}) \right)_k, \tag{5}$$

where the $k$-th row of $\widehat{W}$ is given by $\widehat{\mu}_t(k) = \frac{1}{2n_t(k)} \sum_{i=1}^{n_t} \left( \hat{f}_{n_s}(\mathbf{x}_{t,1}^{(i)}) + \hat{f}_{n_s}(\mathbf{x}_{t,2}^{(i)}) \right) \mathbb{1}\{y_i = k\}$, therein, $n_t(k) = \sum_{i=1}^{n_t} \mathbb{1}\{y_i = k\}$. Here the Adv-SSL estimator $\hat{f}_{n_s}$ is defined as (4). The classifier defined in (5) indicates that by calculating the average representations for each class, we build a template for each downstream class individually. Whenever a new sample needs to be classified, it is assigned to the category of the template that it most closely resembles. Furthermore, we use the following misclassification rate to evaluate the quality of $\hat{f}_{n_s}$.

$$\text{Err}(Q_{\hat{f}_{n_s}}) = \mathbb{P}_t\{Q_{\hat{f}_{n_s}}(\boldsymbol{x}_t) \neq y\}, \tag{6}$$

Appendix B summarizes the notations used throughout this paper for easy cross-checking.

### 3.2 Theoretical limitation induced by bias

In this section, we aim to elucidate the limitations imposed by bias from theoretical perspective. We first assert that $\mathbb{E}_{\widetilde{D}_s, \widetilde{D}_t}\{\text{Err}(Q_{\hat{f}_{n_s}})\} \lesssim \sqrt{\mathbb{E}_{\widetilde{D}_s}\{\mathcal{L}(\hat{f}_{n_s})\}}$ under specific conditions, the details of which can be found in F.4.7. Consequently, analyzing the sample complexity of $\mathbb{E}_{\widetilde{D}_s}\{\mathcal{L}(\hat{f}_{n_s})\}$ is essential for establishing an end-to-end theoretical guarantee for $\hat{f}_{n_s}$. However, the bias between $\mathcal{L}(f)$ and its empirical counterpart $\widehat{\mathcal{L}}(f)$ presents a significant challenge for this analysis.

In fact, in the field of learning theory, the condition $\mathbb{E}_{\widetilde{D}_s}\{\widehat{\mathcal{L}}(f)\} = \mathcal{L}(f)$ is quite important to explore the sample complexity of $\mathbb{E}_{\widetilde{D}_s}\{\mathcal{L}(\hat{f}_{n_s})\}$. Specifically, let $\bar{f}$ satisfy $\mathcal{L}(\bar{f}) - \mathcal{L}(f^*) = \inf_{f \in \mathcal{F}}\{\mathcal{L}(f) - \mathcal{L}(f^*)\}$, where we recall $\widehat{\mathcal{R}}(f)$ is given by (2), then

$$\mathcal{L}(\hat{f}_{n_s}) = \{\mathcal{L}(\hat{f}_{n_s}) - \widehat{\mathcal{L}}(\hat{f}_{n_s})\} + \{\widehat{\mathcal{L}}(\hat{f}_{n_s}) - \mathcal{L}(\bar{f})\} + \{\mathcal{L}(\bar{f}) - \mathcal{L}(f^*)\} + \mathcal{L}(f^*)$$

$$\leq \{\mathcal{L}(\hat{f}_{n_s}) - \widehat{\mathcal{L}}(\hat{f}_{n_s})\} + \{\widehat{\mathcal{L}}(\bar{f}) - \mathcal{L}(\bar{f})\} + \{\mathcal{L}(\bar{f}) - \mathcal{L}(f^*)\} + \mathcal{L}(f^*)$$

$$\leq 2 \sup_{f \in \mathcal{F}} \left| \mathcal{L}(f) - \widehat{\mathcal{L}}(f) \right| + \inf_{f \in \mathcal{F}}\{\mathcal{L}(f) - \mathcal{L}(f^*)\} + \mathcal{L}(f^*),$$

where the first inequality follows from the fact that $\hat{f}_{n_s}$ minimizes the empirical risk $\widehat{\mathcal{L}}(f)$ over $\mathcal{F}$. Taking the expectation with respect to $\widetilde{D}_s$ on both sides yields $\mathbb{E}_{\widetilde{D}_s}\{\mathcal{L}(\hat{f}_{n_s})\} \leq \mathcal{L}(f^*) +$

$2\mathbb{E}_{\widetilde{D}_s}\big\{\sup_{f\in\mathcal{F}}\big|\mathcal{L}(f)-\widehat{\mathcal{L}}(f)\big|\big\}+\inf_{f\in\mathcal{F}}\{\mathcal{L}(f)-\mathcal{L}(f^*)\}$. Standard techniques from empirical process [17] can be used to estimate the second term in unbiased settings. However, the presence of bias complicates their direct application. In contrast, leveraging the unbiased nature of Adv-SSL, we develop a novel error decomposition as follows:

$$\mathbb{E}_{\widetilde{D}_s}\Big\{\mathcal{L}(\hat{f}_{n_s})\Big\}\lesssim\mathcal{L}(f^*)+\mathbb{E}_{\widetilde{D}_s}\Big\{\sup_{f\in\mathcal{F},G\in\widehat{\mathcal{G}}(f)}\Big|\mathcal{L}(f,G)-\widehat{\mathcal{L}}(f,G)\Big|\Big\}+\inf_{f\in\mathcal{F}}\Big\{\mathcal{L}(f)-\mathcal{L}(f^*)\Big\}$$
$$+\mathbb{E}_{\widetilde{D}_s}\Big[\sup_{f\in\mathcal{F}}\Big\{G^*(f)-\widehat{G}(f)\Big\}\Big], \tag{7}$$

where $G^*(f)=\mathbb{E}_{\boldsymbol{x}_s\sim\mathbb{P}_s}\mathbb{E}_{\mathrm{x}_{s,1},\mathrm{x}_{s,2}\in\mathcal{A}(\boldsymbol{x}_s)}\big\{f(\mathrm{x}_{s,1})f(\mathrm{x}_{s,2})^\top\big\}-I_{d^*}\in\mathbb{R}^{d^*}$ and its sample counterpart $\widehat{G}(f)=\frac{1}{n_s}\sum_{i=1}^{n_s}f(\boldsymbol{x}_1^{(i)})f(\boldsymbol{x}_2^{(i)})^\top-I_{d^*}$. We defer the corresponding proof to Section F.4.1. This decomposition allows us to directly apply empirical process methods to handle the second term on the right-hand side, as presented in Section F.4.3. Regarding the other terms, the first term vanishes under Assumption 2, as demonstrated in Section F.4.2. The third term, known as the approximation error, quantifies the error introduced by using $\mathcal{F}$ to approximate $f^*$; this can be controlled using existing results from [25], as shown in Section F.4.4. The last term can be reformulated as a standard problem concerning the rate of convergence of the empirical mean to the population mean, as discussed in Section F.4.5. By combining these results, we leverage the adversarial formulation of Adv-SSL to successfully establish a end-to-end theoretical guarantee for $\hat{f}_{n_s}$.

## 3.3 Assumptions

We begin with defining the Hölder class, which plays a key role in bounding the approximation error.

**Definition 2.** Let $d\in\mathbb{N}$ and $\alpha=r+\beta>0$, where $r\in\mathbb{N}_0$ and $\beta\in(0,1]$. We assert $f:\mathbb{R}^d\to\mathbb{R}$ belongs to the Hölder class $\mathcal{H}^\alpha(\mathbb{R}^d)$ if and only if

$$|\partial^{\boldsymbol{s}}f(\boldsymbol{x})|\le 1 \text{ and } \max_{\|\boldsymbol{s}\|_1=r}\sup_{\boldsymbol{x}\neq\boldsymbol{y}}\frac{\partial^{\boldsymbol{s}}f(\boldsymbol{x})-\partial^{\boldsymbol{s}}f(\boldsymbol{y})}{\|\boldsymbol{x}-\boldsymbol{y}\|_\infty^\beta}\le 1,$$

where for a multi-index $\boldsymbol{s}=(s_1,\ldots,s_d)\in\mathbb{N}^d$ and $f:\mathbb{R}^d\to\mathbb{R}$, the symbol $\partial^{\boldsymbol{s}}f$ denotes the partial differential operator $\partial^{\boldsymbol{s}}=\frac{\partial^{s_1}}{\partial x_1^{s_1}}\frac{\partial^{s_2}}{\partial x_1^{s_2}}\cdots\frac{\partial^{s_d}}{\partial x_d^{s_d}}$. Furthermore, we define $\mathcal{H}^\alpha:=\{f:[0,1]^d\to\mathbb{R},f\in\mathcal{H}^\alpha(\mathbb{R}^d)\}$ as the restriction of $\mathcal{H}^\alpha(\mathbb{R}^d)$ to $[0,1]^d$.

In this context, we make following assumption on $f^*$:

**Assumption 1.** There exists $\alpha=r+\beta$ with $r\in\mathbb{N}$ and $\beta\in(0,1]$ s.t $f_i^*\in\mathcal{H}^\alpha$ for each $i\in[d^*]$.

Assumption 1 is a standard assumption in the nonparametric statistics [33, 32].

As for the term $\mathcal{L}(f^*)$ in eq (7), we need following Assumption 2 to justify $\mathcal{L}(f^*)=0$.

**Assumption 2.** Assume there exists a measurable partition $\{\mathcal{P}_1,\ldots,\mathcal{P}_{d^*}\}$ of $\mathcal{X}_s$ satisfying $\mathbb{P}_s(\mathcal{P}_i)\in[\frac{1}{B_2^2},\frac{1}{B_1^2}]$ for each $i\in[d^*]$.

Assumption 2 requires that the source data distribution is not overly singular. In particular, all common continuous distributions defined on the Borel algebra satisfy this condition, as the measure of any single point is zero. Further details regarding the vanishing of $\mathcal{L}(f^*)$ are provided in Section F.4.2.

Additionally, we introduce two assumptions regarding the data augmentations.

**Assumption 3.** Assume any data augmentation $A_i\in\mathcal{A}$ is $M$-Lipschitz continuous, that is, $\|A_i(\boldsymbol{x})-A_i(\boldsymbol{y})\|_2\le M\|\boldsymbol{x}-\boldsymbol{y}\|_2$ for any $\boldsymbol{x},\boldsymbol{y}\in[0,1]^d$.

The most commonly used augmentations, including cropping, horizontal mirroring, color jittering, grayscale conversion, and Gaussian blurring, actually all satisfy this assumption. See Section E.1.

In addition to the Lipschitz property of data augmentation, we adopt Definition 3 to mathematically quantify the quality of data augmentations. To present it, we define $C_t(k)$ as a set such that $\boldsymbol{x}_t\in C_t(k)$ if and only if $\boldsymbol{x}_t$ belongs to the $k$-th class. Correspondingly, similar to [24], we assume that any upstream instance $\boldsymbol{x}_s$ can be categorized into one or more latent classes $\{C_s(k)\}_{k\in[K]}$.

**Definition 3.** A data augmentations $\mathcal{A}$ is referred to as a $(\sigma_s, \sigma_t, \delta_s, \delta_t)$-augmentations if for each $k \in [K]$, there exists two subsets $\widetilde{C}_s(k) \subseteq C_s(k)$ and $\widetilde{C}_t(k) \subseteq C_t(k)$ such that (i) $\mathbb{P}_s\big(\boldsymbol{x}_s \in \widetilde{C}_s(k)\big) \geq \sigma_s \mathbb{P}_s\big(\boldsymbol{x}_s \in C_s(k)\big)$, (ii) $\sup_{\boldsymbol{x}_{s,1}, \boldsymbol{x}_{s,2} \in \widetilde{C}_s(k)} \min_{\mathrm{x}_{s,1} \in \mathcal{A}(\boldsymbol{x}_{s,1}), \mathrm{x}_{s,2} \in \mathcal{A}(\boldsymbol{x}_{s,2})} \big\| \mathrm{x}_{s,1} - \mathrm{x}_{s,2} \big\|_2 \leq \delta_s$; (iii) $\mathbb{P}_t\big(\boldsymbol{x}_t \in \widetilde{C}_t(k)\big) \geq \sigma_t \mathbb{P}_t\big(\boldsymbol{x}_t \in C_t(k)\big)$, (iv) $\sup_{\boldsymbol{x}_{t,1}, \boldsymbol{x}_{t,2} \in \widetilde{C}_t(k)} \min_{\mathrm{x}_{t,1} \in \mathcal{A}(\boldsymbol{x}_{t,1}), \mathrm{x}_{t,2} \in \mathcal{A}(\boldsymbol{x}_{t,2})} \big\| \mathrm{x}_{t,1} - \mathrm{x}_{t,2} \big\|_2 \leq \delta_t$ and (v) $\mathbb{P}_t\big( \cup_{k=1}^K \widetilde{C}_t(k) \big) \geq \sigma_t$, where $\sigma_s, \sigma_t \in (0, 1]$ and $\delta_s, \delta_t \geq 0$.

Broadly speaking, this definition emphasizes that robust data augmentation should consistently produce distance-closed augmented views for semantically similar original images. We refer to Section E.2 for further explanations. In this context, we introduce the following assumption to delineate the data augmentation necessary for the end-to-end theoretical guarantee of Adv-SSL.

**Assumption 4** (Existence of augmentation sequence). Assume there exists a sequence of $(\sigma_s^{(n)}, \sigma_t^{(n)}, \delta_s^{(n)}, \delta_t^{(n)})$-data augmentations $\mathcal{A}_n = \{A_i^{(n)}\}_{i \in [m]}$ such that (i) $\max\{\delta_s^{(n)}, \delta_t^{(n)}\} \lesssim n^{-\frac{\epsilon_\mathcal{A} + d + 1}{2(\alpha + d + 1)}}$ holds for some $\epsilon_\mathcal{A} > 0$, (ii) $\min\{\sigma_s^{(n)}, \sigma_t^{(n)}\} \to 1$ as $n \to \infty$.

It is noteworthy that this assumption closely aligns with [21, Assumption 3.5] and [20, Assumption 3.6], both of which require the augmentations must be sufficiently robust to ensure the internal connections within latent classes remain strong enough to prevent the separation of instance clusters.

We next introduce the assumption related to distribution shift. Prior to characterizing the transferability from the source domain to the target domain, we must first quantify the similarity between these domains. For ease of presentation, let $p_s(k) = \mathbb{P}_s\big(\boldsymbol{x}_s \in C_s(k)\big)$ and denote $\mathbb{P}_s(k)(\cdot)$ as the probability distribution of the source data that categorized into the $k$-th latent class $C_s(k)$, i.e., $\mathbb{P}_s(k)(\cdot) = \mathbb{P}_s\big( \cdot \mid \boldsymbol{x}_s \in C_s(k)\big)$. Similarly, let $p_t(k) = \mathbb{P}_t\big(\boldsymbol{x}_t \in C_t(k)\big)$ and $\mathbb{P}_t(k)(\cdot) = \mathbb{P}_t\big( \cdot \mid \boldsymbol{x}_t \in C_t(k)\big)$. In this context, we make following assumption:

**Assumption 5** (Domain shift). There exists a $\epsilon_{\mathrm{ds}} > 0$ such that both $\max_{k \in [K]} \mathcal{W}\big(\mathbb{P}_s(k), \mathbb{P}_t(k)\big) \lesssim n_s^{-\frac{\epsilon_{\mathrm{ds}} + d + 1}{2(\alpha + d + 1)}}$ and $\max_{k \in [K]} |p_s(k) - p_t(k)| \lesssim n_s^{-\frac{\epsilon_{\mathrm{ds}}}{2(\alpha + d + 1)}}$.

Generally speaking, smaller $\epsilon_{\mathrm{ds}}$ indicates less discrepancy between the source and target domains. Similar assumptions using alternative divergence measures have been proposed in [5, 16, 13]. To help readers quickly understand this assumption, we discuss it more specifically in Section E.3.

### 3.4 End-to-end theoretical guarantee

We present the end-to-end theoretical guarantee as follow and defer its proof to Appendix F.

**Theorem 1.** *Under certain Assumptions, set* $W \gtrsim n_s^{\frac{2d + \alpha}{4(\alpha + d + 1)}}$, $L \geq 2\lceil \log_2(d + r) \rceil + 2, \mathcal{K} \asymp n_s^{\frac{d + 1}{2(\alpha + d + 1)}}$ *and* $\mathcal{A} = \mathcal{A}_{n_s}$ *(excellent data augmentation), then we have*

$$\mathbb{E}_{\widetilde{D}_s, \widetilde{D}_t}\big\{\mathrm{Err}(Q_{\hat{f}_{n_s}})\big\} \lesssim \big(1 - \sigma_s^{(n_s)}\big) + n_s^{-\frac{\min\{\alpha, \epsilon_\mathcal{A}, \epsilon_{\mathrm{ds}}\}}{32(\alpha + d + 1)}} + \frac{1}{\min_k \sqrt{n_t(k)}}$$

*for sufficiently large* $n_s$.

**Interpretation of sample complexity regarding** $n_s$   The upper bound of misclassification error in Theorem 1 offers several key insights into the convergence behavior regarding $n_s$. First, as the data dimensionality $d$ increases, the convergence rate with respect to the sample size $n_s$ slows down, reflecting the curse of dimensionality. In contrast, as the augmentation quality $\epsilon_\mathcal{A}$ increases, indicating better augmentation, the convergence rate of the upper bound on the misclassification rate with respect to $n_s$ improves. Similarly, $\epsilon_{\mathrm{ds}}$ measures the extent of the shift between the source and target domains. A larger $\epsilon_{\mathrm{ds}}$ indicates the difference between the domains is smaller, a smaller domain difference, which makes the transfer learning task easier and further improves the convergence rate regarding $n_s$ increases. Finally, when both $\epsilon_\mathcal{A}$ and $\epsilon_{\mathrm{ds}}$ exceed $\alpha$, the convergence rate adopts a typical form found in nonparametric statistics [32], specifically $-\frac{\alpha}{32(\alpha + d + 1)}$.

**Few-shot learning**   Theorem 1 demonstrates how the abundance of unlabeled data in the source domain leveraged by Adv-SSL benefits downstream tasks in the target domain. Specifically, the

classification error of downstream tasks consists of three components: the first depends on the data distribution, the second diminishes as the number of unlabeled data in the source domain increases, and the third approaches zero as the quantity of labeled data in downstream tasks grows. Furthermore, when a sufficiently large number of unlabeled samples in the source domain is available, such that $n_s \gtrsim \min_k n_t(k)^{-\frac{16(\alpha+d+1)}{\min\{\alpha,\epsilon,\mathcal{A},\epsilon_{\mathrm{ds}}\}}}$, then $\mathbb{E}_{\widetilde{D}_s,\widetilde{D}_t}\big\{\mathrm{Err}(Q_{\hat{f}_{n_s}})\big\} \lesssim (1-\sigma_s^{(n_s)}) + \frac{1}{\min_k \sqrt{n_t(k)}}$. This finding indicates that classifiers powered by the representation learned by Adv-SSL can achieve excellent performance with minimal labeled samples, thereby providing rigorous theoretical understanding for few-shot learning [28, 30, 35, 27].

**How does the rate change as $m$ increases**    If we consider the case where $m$ increases with $n_s$, according to our theoretical guarantee, we have following conclusions:

$$\mathbb{E}_{\widetilde{D}_s,\widetilde{D}_t}\big\{\mathrm{Err}(Q_{\hat{f}_{n_s}})\big\} \lesssim (1-\sigma_s^{(n_s)}) + m^2 n_s^{-\frac{\min\{\alpha,\epsilon,\mathcal{A},\epsilon_{\mathrm{ds}}\}}{32(\alpha+d+1)}} + \frac{1}{\min_k \sqrt{n_t(k)}}.$$

First of all, as long as $m \lesssim n_s^{\frac{\min\{\alpha,\epsilon,\mathcal{A},\epsilon_{\mathrm{ds}}\}}{64(\alpha+d+1)}}$, the desired asymptotic property can be guaranteed. However, as the growth rate of $m$ increases, the convergence rate with respect to $n_s$ becomes slower. This result is intuitive: a larger $m$ implies that more potential knowledge must be learned from the data, which in turn requires a larger sample size to maintain the same level of misclassification rate. For instance, if we set $m \asymp n_s^{\frac{\min\{\alpha,\epsilon,\mathcal{A},\epsilon_{\mathrm{ds}}\}}{128(\alpha+d+1)}}$, the resulting convergence rate is $n_s^{-\frac{\min\{\alpha,\epsilon,\mathcal{A},\epsilon_{\mathrm{ds}}\}}{64(\alpha+d+1)}}$.

## 4    Comparison with Existing Methods

As the experiments conducted in existing self-supervised learning methods, we pretrain the representation on CIFAR-10, CIFAR-100 and Tiny ImageNet, and subsequently conduct fine-tuning on each dataset with annotations. Table 3 shows the classification accuracy of representations learned by Adv-SSL, compared with baseline methods including SimCLR [8], BYOL [19], WMSE [14], where the results has been reported in [14]. In addition, we also compare Adv-SSL with VICReg [4] and LogDet [39]. The presented result shows that Adv-SSL consistently outperforms previous mainstream self-supervised methods. The experimental details are deferred to Section C and the implementation can be found in https://github.com/vincen-github/Adv-SSL.

| Method | CIFAR-10 | | CIFAR-100 | | Tiny ImageNet | |
|---|---|---|---|---|---|---|
| | Linear | $k$-nn | Linear | $k$-nn | Linear | $k$-nn |
| SimCLR | 91.80 | 88.42 | 66.83 | 56.56 | 48.84 | 32.86 |
| BYOL | 91.73 | 89.45 | 66.60 | 56.82 | **51.00** | 36.24 |
| WMSE2 | 91.55 | 89.69 | 66.10 | 56.69 | 48.20 | 34.16 |
| WMSE4 | 91.99 | 89.87 | 67.64 | 56.45 | 49.20 | 35.44 |
| VICReg | 91.23 | 89.15 | 67.61 | 57.04 | 48.55 | 35.62 |
| LogDet | 92.47 | 90.19 | 67.32 | 57.56 | 49.13 | 35.78 |
| Adv-SSL | **93.01** | **90.97** | **68.94** | **58.50** | 50.21 | **37.40** |

Table 3: Top-1 Accuracy Comparison for Different SSL Methods.

## 5    Conclusion

In this paper, we propose a novel adversarial contrastive learning method for unsupervised transfer learning. Our approach achieves state-of-the-art classification accuracy on various real datasets, outperforming existing self-supervised learning methods under both fine-tuned linear probe and $k$-NN protocols. Additionally, we provide an end-to-end theoretical guarantee for downstream classification tasks in misspecified and over-parameterized settings. Our analysis shows that the misclassification rate depends primarily on the strength of data augmentation applied to large amounts of unlabeled data, and offers new theoretical insights into the effectiveness of few-shot learning for downstream tasks with limited samples.

## 6  Acknowledgments

This work has been funded by the National Key Research and Development Program of China (No. 2024YFA1014200, No. 2023YFA1000103, No. 2022YFA1003702), the National Natural Science Foundation of China (Nos. 123B2019, 12125103, U24A2002, 12371441, 12426309), and the Fundamental Research Funds for the Central Universities.

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

# Contents

## A   Related Works

**Self-supervised contrastive loss**   The loss function proposed by [22] can be regarded as a special version of Adv-SSL with the constraint $\mathrm{x}_{s,1} = \mathrm{x}_{s,2}$. The main difference between Adv-SSL and the approach by [22] lies in the iteration format. As stated in Section 1, optimization deviation can accumulate with each iteration, particularly in the mini-batch scenario, while Adv-SSL employs adversarial training to mitigate this issue. The same problem is encountered by [37], which can be loosely regarded as a biased sample version of (1).

**Self-supervised theory**   Recent theoretical studies can be categorized into two main lines of research. The first line [15, 22, 2, 24] focuses on analyzing the population risk of self-supervised learning methods, which can not characterize how the error in downstream tasks diminishes with increasing sample size. The second line of research [31, 21, 1, 26, 20] studies the generalization error through Rademacher complexity without the consideration of approximation error. However, the absence of approximation error renders the resulting generalization error analysis ineffective. Specifically, ignoring the approximation error by simply supposing $f$ belonging to a deep neural network class, the Rademacher complexity can be significantly reduced by controlling the scale of the network class, leading to impressive upper bounds. However, this controlled neural network class intuitively limits its approximation capacity. The increasing approximation error results in a larger overall error. Therefore, these studies cannot provide theoretical guidance for hypothesis class selection nor fully characterize the total error of self-supervised learning methods. In contrast, our work provides a comprehensive convergence analysis that characterizes how the downstream task error converges with respect to both the number of unlabeled samples in the source domain and labeled samples in the target domain.

## B   Notation List

To reduce confusion and enhance comprehension regarding the symbols used in this study, we have created a list to provide readers with a convenient reference. This list directs readers to the first occurrence of each symbol in the relevant sections or equations. Within this table, the symbol $\square$ indicates an option for $s$ or $t$, representing the source domain and the target domain, respectively.

| Symbol | Description | Reference |
|---|---|---|
| $D_{\square}$ | dataset | Section 2.1 |
| $d^*$ | representation dimension | Section 2.1 |
| $n_t(k)$ | sample size of $k$-th target class | Equation (5) |
| $\mathcal{A}$ | data augmentation | Section 2.1 |

*Continued on next page*

| Symbol | Description | Reference |
|--------|-------------|-----------|
| $m$ | number of augmentations | Section 2.1 |
| $M$ | Lipschitz constant of augmentations | Assumption 3 |
| $\mathbf{x}_\square^{(i)}$ | augmented view | Section 2.1 |
| $\tilde{\mathbf{x}}_\square^{(i)}$ | concatenated augmented view | Section 2.1 |
| $\widetilde{D}_s$ | source augmented dataset | Section 2.1 |
| $\widetilde{D}_t$ | target augmented dataset | Section D.4 |
| $K$ | the number classes | Section 2.1 |
| $C_\square(k)$ | $k$-th source/target class | Definition 3 |
| $\widetilde{C}_\square(k)$ | main part of $C_\square(k)$ | Definition 3 |
| $\mathbb{P}_\square$ | data distribution | Section 2.1 |
| $\mathbb{P}_\square(k)$ | distribution conditioned on $\boldsymbol{x}_\square \in C_\square(k)$ | Assumption 5 |
| $p_\square(k)$ | probability of $\boldsymbol{x}_\square \in C_\square(k)$ | Assumption 5 |
| $\mu_\square(k)$ | $k$-th representation center | Lemma 1 and 2 |
| $\hat{\mu}_t(k)$ | $k$-th empirical center | Equation (5) |
| $f^*$ | population optimal encoder | Equation (2.2) |
| $\hat{f}_{n_s}$ | sample optimal encoder | Equation (4) |
| $Q_{\hat{f}_{n_s}}$ | classifier based on $\hat{f}_{n_s}$ | Equation (5) |
| Err | misclassification error | Equation (6) |
| $\mathcal{W}$ | Wasserstien Distance | Section 2.1 |
| $\mathcal{F}$ | neural network hypothesis space | Equation (4) |
| $\mathcal{G}(f)$ | feasible set of $G$ | Section (2.2) |
| $\mathcal{L}_{\text{align}}$ | alignment term | Section 3.1 |
| $\mathcal{R}(f, G)$ | regularization term | Definition 3 |
| $\mathcal{R}(f)$ | regularization term | Definition 1 |
| $\alpha$ | parameter of Hölder class | Definition 2 |
| $\epsilon_{\mathcal{A}}$ | augmentation parameter | Assumption 4 |
| $\epsilon_{\text{ds}}$ | distribution shift parameter | Assumption 5 |
| $\sigma_\square, \delta_\square$ | parameters of augmentation | Definition 3 |
| $\epsilon_1, \epsilon_2$ | distribution shift | Equation (17) |
| $W, L, \mathcal{K}, B_1, B_2$ | parameters of neural network | Definition 1 |

Table 4: Summary of Symbols

## C   Experimental Details

**Implementation details.** Except for tuning $\lambda$ for different datasets, all other hyperparameters used in our experiments align with [14]. We train for $1,000$ epochs with a learning rate of $3 \times 10^{-3}$ for CIFAR-10 and CIFAR-100, and $2 \times 10^{-3}$ for Tiny ImageNet. A learning rate warm-up is applied for the first $500$ iterations of the optimizer, in addition to a $0.2$ learning rate drop at $50$ and $25$ epochs before the training end. We use a mini-batch size of $256$, and the dimension of the hidden layer in the projection head is set to $1024$. The weight decay is set to $10^{-3}$. We adopt an embedding size ($d^*$) of $512$. The backbone network used in our implementation is ResNet-18.

**Image transformation details.** We randomly apply crops with sizes ranging from $0.08$ to $1.0$ of the original area and aspect ratios ranging from $3/4$ to $4/3$ of the original aspect ratio. Furthermore, we apply horizontal mirroring with a probability of $0.5$. Additionally, color jittering is applied with a configuration of $(0.4; 0.4; 0.4; 0.1)$ and a probability of $0.8$, while grayscaling is applied with a probability of $0.2$. For CIFAR-10 and CIFAR-100, random Gaussian blurring is adopted with a probability of $0.5$ and a kernel size of $0.1$. During testing, only one crop is used for evaluation.

**Evaluation protocol.** During evaluation, we freeze the network encoder and remove the projection head after pretraining, then train a supervised linear classifier on top of it, which is a fully-connected layer followed by softmax. we train the linear classifier for $500$ epochs using the Adam optimizer with corresponding labeled training set without data augmentation. The learning rate is exponentially decayed from $10^{-2}$ to $10^{-6}$. The weight decay is set as $10^{-4}$. we also include the accuracy of a k-nearest neighbors classifier with $k = 5$, which does not require fine tuning.

All experiments were conducted using a single Tesla V100 GPU unit. The PyTorch implementations can be found in supplementary material.

## D    Additional Numerical Experiments

### D.1    Ablation study on the regularization parameter

The regularization parameter $\lambda$ in Adv-SSL balances the alignment term $\mathcal{L}_{\mathrm{align}}(\cdot)$ and the regularization term $\mathcal{R}(\cdot)$. Our theoretical analysis suggests that $\lambda = \mathcal{O}(1)$. Specifically,

- In Lemma 4, we demonstrate that the alignment factor $R_t(\varepsilon, f)$ can be bounded by the alignment term $\mathcal{L}_{\mathrm{align}}(\cdot)$, while the divergence factor $\max_{i \neq j} |\mu_t(t)^\top \mu_t(j)|$ is bounded by the regularization term $\mathcal{R}(\cdot)$.
- Based on the definition of the population risk $\mathcal{L}(f) = \mathcal{L}_{\mathrm{align}}(f) + \lambda \mathcal{R}(f)$, we find

$$\mathcal{L}_{\mathrm{align}}(f) \leq \mathcal{L}(f) \quad \text{and} \quad \mathcal{R}(f) \leq \lambda^{-1} \mathcal{L}(f) \lesssim \mathcal{L}(f),$$

where we used $\lambda = \mathcal{O}(1)$. This allows us to bound both the alignment factor $R_t(\varepsilon, f)$ and the divergence factor $\max_{i \neq j} |\mu_t(t)^\top \mu_t(j)|$ in terms of the population risk $\mathcal{L}(f)$, which leads to the conclusion in Lemma 5.

| Regularization parameter | CIFAR-10 Linear | CIFAR-10 $k$-nn | CIFAR-100 Linear | CIFAR-100 $k$-nn |
|---|---|---|---|---|
| $\lambda = 5.0 \times 10^{-5}$ | 90.11 | 87.72 | 67.59 | 57.34 |
| $\lambda = 1.0 \times 10^{-4}$ | 90.53 | 88.12 | 68.01 | 57.59 |
| $\lambda = 5.0 \times 10^{-4}$ | 92.24 | 89.99 | 68.24 | 58.35 |
| $\lambda = 1.0 \times 10^{-3}$ | 92.01 | 90.18 | 67.88 | 57.89 |
| $\lambda = 5.0 \times 10^{-3}$ | 92.11 | 90.01 | 68.12 | 57.66 |
| $\lambda = 1.0 \times 10^{-2}$ | 92.47 | 90.33 | **68.94** | **58.50** |
| $\lambda = 5.0 \times 10^{-2}$ | **93.01** | **90.97** | 67.68 | 57.13 |
| $\lambda = 1.0 \times 10^{-1}$ | 92.77 | 90.38 | 67.82 | 57.49 |
| $\lambda = 1.0$ | 91.75 | 89.76 | 66.78 | 56.76 |

Table 5: Comparisons of Adv-SSL with different regularization parameters.

### D.2    Ablation study on the data augmentations

| random cropping | grayscale | color distortion | random horizontal flipping | CIFAR-10 Linear | CIFAR-10 $k$-nn |
|---|---|---|---|---|---|
| ✓ | ✓ | ✓ | ✓ | **93.01** | **90.97** |
| ✓ | ✓ | ✓ | | 91.03 | 88.34 |
| ✓ | ✓ | | | 89.18 | 85.65 |
| ✓ | | | | 79.32 | 69.81 |

Table 6: Downstream performance of Adv-SSL under different richness of augmentations.

### D.3    Ablation study on the alignment term

The loss function (4) proposed in this work consists of an alignment term and a regularization term. In contrast, Barlow Twins [37] does not require the alignment term. In this subsection, we show whether this additional alignment term necessary for this method.

[24, Lemma 4.1] has demonstrated that the the diagonal part of the cross-correlation matrix serves as a alignment term under certain conditions. Indeed,

$$\mathbb{E}_{\boldsymbol{x}_s \sim \mathbb{P}_s} \mathbb{E}_{\mathbf{x}_{s,1}, \mathbf{x}_{s,2} \in \mathcal{A}(\boldsymbol{x}_s)} \left[ \left\{ f_i(\mathbf{x}_{s,1}) - f_i(\mathbf{x}_{s,2}) \right\}^2 \right]$$

$$= \mathbb{E}_{\boldsymbol{x}_s \sim \mathbb{P}_s} \mathbb{E}_{\mathbf{x}_{s,1}, \mathbf{x}_{s,2} \in \mathcal{A}(\boldsymbol{x}_s)} \big\{ f_i^2(\mathbf{x}_{s,1}) + f_i^2(\mathbf{x}_{s,2}) \big\} - 2 \mathbb{E}_{\boldsymbol{x}_s \sim \mathbb{P}_s} \mathbb{E}_{\mathbf{x}_{s,1}, \mathbf{x}_{s,2} \in \mathcal{A}(\boldsymbol{x}_s)} \big\{ f_i(\mathbf{x}_{s,1}) f_i(\mathbf{x}_{s,2}) \big\}$$
$$= 2 \mathbb{E}_{\boldsymbol{x}_s \sim \mathbb{P}_s} \mathbb{E}_{\mathbf{x}_s \in \mathcal{A}(\boldsymbol{x})} \big\{ f_i^2(\mathbf{x}_s) \big\} - 2 \mathbb{E}_{\boldsymbol{x}_s \sim \mathbb{P}_s} \mathbb{E}_{\mathbf{x}_{s,1}, \mathbf{x}_{s,2} \in \mathcal{A}(\boldsymbol{x}_s)} \big\{ f_i(\mathbf{x}_{s,1}) f_i(\mathbf{x}_{s,2}) \big\},$$

where the second equality holds from that $\mathbf{x}_{s,1}$ and $\mathbf{x}_{s,2}$ follow the same distribution. The alignment risk is then related to the diagonal part of the cross-correlation matrix as:

$$\mathcal{L}_{\text{align}}^2(f) = \Big( \sum_{i=1}^{d} \mathbb{E}_{\boldsymbol{x}_s \sim \mathbb{P}_s} \mathbb{E}_{\mathbf{x}_{s,1}, \mathbf{x}_{s,2} \in \mathcal{A}(\boldsymbol{x}_s)} \big\{ \big( f_i(\mathbf{x}_{s,1}) - f_i(\mathbf{x}_{s,2}) \big)^2 \big\} \Big)^2$$

$$= 4 \Big( \sum_{i=1}^{d} \Big[ \mathbb{E}_{\boldsymbol{x}_s \sim \mathbb{P}_s} \mathbb{E}_{\mathbf{x}_s \in \mathcal{A}(\boldsymbol{x})} \big\{ f_i^2(\mathbf{x}_s) \big\} - \mathbb{E}_{\boldsymbol{x}_s \sim \mathbb{P}_s} \mathbb{E}_{\mathbf{x}_{s,1}, \mathbf{x}_{s,2} \in \mathcal{A}(\boldsymbol{x}_s)} \big\{ f_i(\mathbf{x}_{s,1}) f_i(\mathbf{x}_{s,2}) \big\} \Big] \Big)^2$$

$$\leq 4d \sum_{i=1}^{d} \Big[ \mathbb{E}_{\boldsymbol{x}_s \sim \mathbb{P}_s} \mathbb{E}_{\mathbf{x}_s \in \mathcal{A}(\boldsymbol{x})} \big\{ f_i^2(\mathbf{x}_s) \big\} - \mathbb{E}_{\boldsymbol{x}_s \sim \mathbb{P}_s} \mathbb{E}_{\mathbf{x}_{s,1}, \mathbf{x}_{s,2} \in \mathcal{A}(\boldsymbol{x}_s)} \big\{ f_i(\mathbf{x}_{s,1}) f_i(\mathbf{x}_{s,2}) \big\} \Big]^2,$$

where the last inequality follows from Cauchy-Schwarz inequality. It is crucial that the right-hand side of the inequality is consistent with the alignment term in the loss function of Barlow Twins, provided that $\mathbb{E}_{\boldsymbol{x}_s \sim \mathbb{P}_s} \mathbb{E}_{\mathbf{x}_s \in \mathcal{A}(\boldsymbol{x})} \big\{ f_i^2(\mathbf{x}_s) \big\} = 1$ for each $i \in \{1, \ldots, d\}$. However, this condition does not hold generally.

Therefore, from a theoretical perspective, the cross-correlation loss alone, as used in Barlow Twins [37], is insufficient for learning representations invariant to augmentations, as previously discussed. To address this, we introduce an additional explicit alignment term in the loss, as also suggested by [22, 20].

From a practical perspective, we conduct an ablation study comparing Adv-SSL with and without the explicit alignment term. Our results shown in Table 7 indicate that the inclusion of the explicit alignment term improves Adv-SSL's performance.

| Method | CIFAR-10 | | CIFAR-100 | |
|---|---|---|---|---|
| | Linear | $k$-nn | Linear | $k$-nn |
| Adv-SSL without alignment | 92.42 | 90.01 | 67.27 | 58.10 |
| Adv-SSL with alignment | **93.01** | **90.97** | **68.94** | **58.50** |

Table 7: Comparisons of Adv-SSL without and with the alignment term.

### D.4 Transfer learning

To align the experiments with the theoretical settings, we conduct a simple additional experiment in terms of transfer learning. Specifically, we transfer the representation trained on CIFAR-100 to CIFAR-10. Compared to the performance of Barlow Twins [37], we can see that Adv-SSL indeed has strong transferability in practice, as demonstrated in our theory.

| Methods | Linear | $k$-nn |
|---|---|---|
| Barlow Twins[37] | 73.56 | 66.34 |
| Beyond Separability[22] | 74.11 | 66.79 |
| Adv-SSL | **80.57** | **73.41** |

Table 8: Transfer learning from CIFAR-100 to CIFAR-10

## E   Additional Discussions on Assumptions

### E.1   Discussions on assumption 3

Assumption 3 is mild and numerous commonly-used augmentation methods satisfy this assumption. Specifically, all of the augmentation methods used in our experiments meet this requirement.

As outlined in Section C, the data augmentations used in our experiments, including crops, horizontal mirroring, color jittering, gray scaling, and Gaussian blurring, are indeed Lipschitz continuous. In the following, we provide a detailed justification for each of these transformations.

**Crops:** For each image $\boldsymbol{x} \in \mathbb{R}^d$, a crop of this image is defined as $\mathrm{Crop}(\boldsymbol{x}; I) = \boldsymbol{x}_I$, for some index set $I \subseteq \{1, \ldots, d\}$. The Lipschitz continuity follows from the fact that:

$$\| \mathrm{Crop}(\boldsymbol{x}; I) - \mathrm{Crop}(\boldsymbol{y}; I) \|_2^2 = \| \boldsymbol{x}_I - \boldsymbol{y}_I \|_2^2 = \sum_{i \in I} (\boldsymbol{x}_i - \boldsymbol{y}_i)^2$$

$$\leq \sum_{i=1}^d (\boldsymbol{x}_i - \boldsymbol{y}_i)^2 = \| \boldsymbol{x} - \boldsymbol{y} \|_2^2.$$

Thus crop is 1-Lipschitz.

**Horizontal mirroring:** For each image $\boldsymbol{x} \in \mathbb{R}^d$, a horizontal mirror of this image can be formulated as $\mathrm{Mirror}(\boldsymbol{x}) = \boldsymbol{x}_I$, where the index set $I$ is a rearrangement of $\{1, \ldots, d\}$. We find

$$\| \mathrm{Mirror}(\boldsymbol{x}) - \mathrm{Mirror}(\boldsymbol{y}) \|_2^2 = \| \boldsymbol{x}_I - \boldsymbol{y}_I \|_2^2 = \sum_{i \in I} (\boldsymbol{x}_I - \boldsymbol{y}_I)^2$$

$$= \sum_{i=1}^d (\boldsymbol{x}_I - \boldsymbol{y}_I)^2 = \| \boldsymbol{x} - \boldsymbol{y} \|_2^2.$$

Thus horizontal mirroring is 1-Lipschitz.

**Color jittering:** As an example, consider the brightness adjustment for color jittering. The color jittering operator is defined as $\mathrm{Jitter}(\boldsymbol{x}) = \mathrm{clip}(\alpha \boldsymbol{x}, 0, 1)$ for some adjustment factor $\alpha > 0$. If $\alpha > 1$, the image becomes brighter; if $\alpha < 1$, the image becomes darker. Then

$$\| \mathrm{Jitter}(\boldsymbol{x}) - \mathrm{Jitter}(\boldsymbol{y}) \|_2^2 = \sum_{i=1}^d \big( \mathrm{clip}(\alpha \boldsymbol{x}_I, 0, 1) - \mathrm{clip}(\alpha \boldsymbol{y}_I, 0, 1) \big)^2$$

$$\leq \sum_{i=1}^d (\alpha \boldsymbol{x}_I - \alpha \boldsymbol{y}_I)^2 \leq \alpha^2 \| \boldsymbol{x} - \boldsymbol{y} \|_2^2.$$

Thus color jittering is $\alpha$-Lipschitz.

**Grayscaling:** The grayscale transformation is a weighted sum of the RGB channels. For a RGB image $\boldsymbol{x} \in \mathbb{R}^d$, the red channel is defined as $\boldsymbol{x}_R := (\boldsymbol{x}_i)_{i=1}^{d/3}$, the green channel is defined as $\boldsymbol{x}_G := (\boldsymbol{x}_i)_{i=d/3+1}^{2d/3}$, and the blue channel is defined as $\boldsymbol{x}_B := (\boldsymbol{x}_i)_{i=2d/3}^d$. The grayscaling of this image is defined as $\mathrm{Gray}(\boldsymbol{x}) = \alpha \boldsymbol{x}_R + \beta \boldsymbol{x}_G + \gamma \boldsymbol{x}_B$ for some $\alpha, \beta, \gamma \in (0, 1)$. The Lipschitz continuity follows from:

$$\| \mathrm{Gray}(\boldsymbol{x}) - \mathrm{Gray}(\boldsymbol{y}) \|_2 \leq \alpha \| \boldsymbol{x}_R - \boldsymbol{y}_R \|_2 + \beta \| \boldsymbol{x}_G - \boldsymbol{y}_G \|_2 + \gamma \| \boldsymbol{x}_B - \boldsymbol{y}_B \|_2$$
$$\leq \max\{\alpha, \beta, \gamma\} \| \boldsymbol{x} - \boldsymbol{y} \|_2,$$

where the first inequality holds from Jensen's inequality. Thus grayscaling is $\max\{\alpha, \beta, \gamma\}$-Lipschitz.

**Gaussian blurring:** Gaussian blurring applies a Gaussian kernel to smooth the image, reducing high-frequency noise and detail. The blurred image $\mathrm{GaussianBlur}(\boldsymbol{x}; \sigma) = \boldsymbol{x} * K_\sigma$ is defined by convolving the original image $\boldsymbol{x}$ with a Gaussian kernel $K_\sigma$. Convolution is a linear operation, and it is well-known that convolution with a Gaussian kernel is Lipschitz continuous, where the Lipschitz constant depends on the kernel size and $\sigma$. Therefore, Gaussian blurring is Lipschitz continuous with a constant that depends on the kernel size and $\sigma$.

### E.2 Discussions on assumption 4

The concept of $(\sigma_s, \delta_s, \sigma_t, \delta_t)$-augmentation is introduced to quantify the concentration of augmented data, which is a extensive version of $(\sigma_s, \delta_s)$ augmentations proposed by [24, Definition 1] in terms of transfer learning. We now provide a step-by-step explanation:

- Augmentation distance: for a given augmentation set $\mathcal{A}$, the augmentation distance between two samples $\boldsymbol{x}$ and $\boldsymbol{y}$ are defined as the minimum distance between their augmented views: $\|\boldsymbol{x}-\boldsymbol{y}\|_{\mathcal{A}} := \min_{\mathrm{x}\in\mathcal{A}(\boldsymbol{x}),\mathrm{y}\in\mathcal{A}(\boldsymbol{y})}\|\mathrm{x}-\mathrm{y}\|_2$. Since augmentations can capture partial semantic meanings of the original sample through various views, this augmentation distance reflects the maximal semantic similarity between the two samples.

- $\sigma_s$-main-part of the latent class: for a latent class in the source domain $C_s(k)$, the $\sigma_s$-main-part is defined as $\widetilde{C}_s(k) \subseteq C_s(k)$ satisfying $\mathbb{P}_s\big(\boldsymbol{x}\in\widetilde{C}_s(k)\big) \geq \sigma_s\mathbb{P}_s\big(\boldsymbol{x}\in C_s(k)\big)$. The parameter $\sigma_s$ quantifies the concentration of the distribution $\mathbb{P}_s(k)(\cdot) := \mathbb{P}_s\big(\,\cdot\,|\boldsymbol{x}\in C_s(k)\big)$ of this latent class. Specifically, for fixed sets $\widetilde{C}_s(k)$ and $C_s(k)$, a larger value of $\sigma_s$ indicates a higher concentration of the distribution $\mathbb{P}_s(k)$.

- Augmentation diameter of the $\sigma_s$-main-part: the parameter $\delta_s$ is defined as the diameter of the $\sigma_s$-main-part in augmentation distance, that is, $\sup_{\boldsymbol{x},\boldsymbol{y}\in\widetilde{C}_s(k)}\|\boldsymbol{x}-\boldsymbol{y}\|_{\mathcal{A}}$. For a fixed distribution $\mathbb{P}_s$ and a fixed parameter $\sigma_s$, the smaller value of the diameter $\delta_s$ means a higher concentration of the augmented distribution, as well as greater similarity between augmented data samples.

- Summary for $(\sigma_s,\delta_s)$-augmentations: The concentration of the augmented distribution, as measured by the pair of parameters $(\sigma_s,\delta_s)$, depends on both the distribution $\mathbb{P}_s(k)$ and the augmentation set $\mathcal{A}$. Specifically, for a fixed augmentation set $\mathcal{A}$, a smaller value of $\sigma_s$ and a higher concentration of $\mathbb{P}_s(k)$ result in a smaller $\sigma_s$-main-part $\widetilde{C}_s(k)$, leading to a smaller value of $\delta_s$. Additionally, for a fixed distribution $\mathbb{P}_s(k)$, a smaller value of $\sigma_s$ and a larger augmentation set $\mathcal{A}$ lead to smaller augmentation distances $\|\boldsymbol{x}-\boldsymbol{y}\|_{\mathcal{A}}$ for each pair of samples $(\boldsymbol{x},\boldsymbol{y})$, resulting in a smaller value of $\delta_s$.

- The conditions (i)-(iv) in Definition 3 can be considered an extensive version that takes into account the difference between the source domain and the target domain.

- The extra condition (v) in Definition 3 replaces the assumption $\mathcal{A}\big(C_t(i)\big) \cap \mathcal{A}\big(C_t(j)\big) = \emptyset$ required by [24]. This implies that the augmentation methods used should be intelligent enough to recognize objects that align with the image labels in multi-objective images. A straightforward alternative to this requirement is to assume that different classes $C_t(k)$ are pairwise disjoint, meaning that for all $i \neq j, C_t(i) \cap C_t(j) = \emptyset$, which implies that

$$\mathbb{P}_t\big(\cup_{k=1}^K \widetilde{C}_t(k)\big) = \sum_{k=1}^K \mathbb{P}_t\big(\widetilde{C}_t(k)\big) \geq \sigma_t \sum_{k=1}^K \mathbb{P}_t\big(C_t(k)\big) = \sigma_t.$$

To ensure readers can get quickly understanding for the Definition 3, we provide following example:

**Example 1.** Suppose the samples in the $k$-th latent class follows the uniform distribution on $[0, R]$, i.e., $C_s(k) = [0, R]$ and $\mathbb{P}_s(k) = \mathsf{unif}(0, R)$. For each $\sigma_s \in (0, 1]$, we can find a $\sigma_s$-main-part of $C_s(k)$ as $\widetilde{C}_s(k) = [0, \sigma_s R]$. Further, we define the augmentation set as $\mathcal{A}(x) = \{\mathrm{x} \in \mathbb{R} : |x - \mathrm{x}| \leq r\}$ for each $x \in \mathcal{X}$. Then the augmentation diameter $\delta_s$ of the $\sigma_s$-main-part is given as

$$\sup_{x,y\in\widetilde{C}_s(k)} \|x - y\|_{\mathcal{A}} = \max\{\sigma_s R - 2r, 0\} =: \delta_s.$$

The parameters $\sigma_s$, $\delta_s$, $r$ and $R$ are interrelated by this equality. Note that the parameter $R$ reflects the concentration of the distribution $\mathbb{P}_s(k)$ within the latent class. A smaller value of $R$ indicates a higher concentration of $\mathbb{P}_s(k)$, which in turn leads to a smaller value of the augmentation diameter $\delta_s$. Additionally, a larger augmentation set, i.e., a larger value of $r$, results in a smaller value of the augmentation diameter $\delta_s$.

### E.3 Discussions on assumption 5

Assumption 5 is common in the theory of transfer learning, such as [5, 16, 13]. We now provide a concrete example for more intuition. Consider the following example using one-dimensional Gaussian mixtures. Specifically, we define the source and target distributions as follows:

$$\mathbb{P}_s := \sum_{k=1}^K w_s(k)\mathbb{P}_s(k), \quad \sum_{k=1}^K w_s(k) = 1,$$

$$\mathbb{P}_t := \sum_{k=1}^{K} w_t(k) \mathbb{P}_t(k), \quad \sum_{k=1}^{K} w_t(k) = 1,$$

where the distributions of each latent class are Gaussian:

$$\mathbb{P}_s(k) := N\big(\mu_s(k), \sigma^2\big), \quad \mathbb{P}_t(k) := N\big(\mu_t(k), \sigma^2\big), \quad 1 \le k \le K.$$

Then the parameter $\epsilon_1$ is the maximum distance between the means of the source and target distributions for each latent class:

$$\epsilon_1 = \max_{k \in [K]} \mathcal{W}\big(\mathbb{P}_s(k), \mathbb{P}_t(k)\big) = \max_{k \in [K]} \big\{ |\mu_s(k) - \mu_t(k)| \big\}$$

Additionally, the parameter $\epsilon_2$ is the maximum distance between the mixture weights of the source and target distributions:

$$\epsilon_2 = \max_{k \in [K]} |w_s(k) - w_t(k)|.$$

Thus, Assumption 5 not only requires that the source and target distributions for each latent class are close in terms of their means, but also that their mixture weights are similar.

# F  Proof of Theorem 1

## F.1  Proof sketch

In this section, we focus on providing the proof sketch for Theorem 1. Based on [24], we begin by exploring the sufficient condition regarding the downstream error bound, as shown in Lemma 1. Specifically, it reveals that the error bound $\mathrm{Err}\big(Q_f\big) \le (1 - \sigma_t) + R_t(\varepsilon, f)$ holds under the condition $\max_{i \ne j} \mu_t(i)^\top \mu_j < B_2^2 \psi(\sigma_t, \delta_t, \varepsilon, f)$. The naturally raised question is whether minimaxing the risk of Adv-SSL can help us meet the required condition. To answer this question, we establish Lemma 4 in Section F.3, which reveals that minimizing the risk of Adv-SSL can achieve the requirement $\max_{i \ne j} \mu_t(i)^\top \mu_j < B_2^2 \psi(\sigma_t, \delta_t, \varepsilon, f)$. Meanwhile, its directly induced corollary 1 indicates that we should explore the sample complexity of $\mathbb{E}_{\widetilde{D}_s}\big\{\mathcal{L}(f)\big\}$. To this end, we begin by developing a novel error decomposition approach in Section F.4.1, which decouples $\mathbb{E}_{\widetilde{D}_s}\big\{\mathcal{L}(f)\big\}$ into four terms: $\mathcal{L}(f^*)$, the statistical error $\mathcal{E}_{\mathrm{sta}}$ regarding the neural network class $\mathcal{F}$, the approximation error $\mathcal{E}_{\mathcal{F}}$, and the error induced by the dual variable $\mathcal{E}_{\mathcal{G}}$. We then deal with them individually in Sections F.4.2, F.4.3, F.4.4, and F.4.5 respectively. Subsequently, we conduct the tradeoff to obtain the sample complexity of $\mathbb{E}_{\widetilde{D}_s}\big\{\mathcal{L}(f)\big\}$ and the corresponding parameters of the network, including width, depth, and norm constraint. Based on these results, we can derive the desired error upper bound for Adv-SSL, as shown in Theorem 1, which completes the proof.

## F.2  Sufficient condition of small misclassification rate

To begin with, let $\mu_t(k) := \mathbb{E}_{\boldsymbol{x}_t \in C_t(k)} \mathbb{E}_{\mathbf{x}_t \in \mathcal{A}(\boldsymbol{x}_t)} \big\{ f(\mathbf{x}_t) \big\} = \frac{1}{p_t(k)} \mathbb{E}_{\boldsymbol{x}_t \sim \mathbb{P}_t} \mathbb{E}_{\mathbf{x}_t \in \mathcal{A}(\boldsymbol{x}_t)} \big[ f(\mathbf{x}_t) \mathbb{1}\big\{ \boldsymbol{x}_t \in C_t(k) \big\} \big]$, inspired by [24], we have following lemma.

**Lemma 1.** *Given a $(\sigma_s, \sigma_t, \delta_s, \delta_t)$-augmentation, if the encoder $f$ such that $B_1 \le \|f\|_2 \le B_2$ is $\mathcal{K}$-Lipschitz and*

$$\mu_t(i)^\top \mu_t(j) < B_2^2 \psi(\sigma_t, \delta_t, \varepsilon, f),$$

*holds for any pair of $(i, j)$ with $i \ne j$, then the downstream error rate of $Q_f$*

$$\mathrm{Err}(Q_f) \le (1 - \sigma_t) + R_t(\varepsilon, f),$$

*where $R_t(\varepsilon, f) = \mathbb{P}_t\big(\boldsymbol{x}_t \in \mathcal{X}_t : \sup_{\mathbf{x}_{t,1}, \mathbf{x}_{t,2} \in \mathcal{A}(\boldsymbol{x}_t)} \|f(\mathbf{x}_{t,1}) - f(\mathbf{x}_{t,2})\|_2 > \varepsilon\big)$, $\psi(\sigma_t, \delta_t, \varepsilon, f) = \Gamma_{\min}(\sigma_t, \delta_t, \varepsilon, f) - \sqrt{2 - 2\Gamma_{\min}(\sigma_t, \delta_t, \varepsilon, f)} - \frac{1}{2}\Big(1 - \frac{\min_{k \in [K]} \|\hat{\mu}_t(k)\|_2^2}{B_2^2}\Big) - \frac{2\max_{k \in [K]} \|\hat{\mu}_t(k) - \mu_t(k)\|_2}{B_2}$, herein, $\Gamma_{\min}(\sigma_t, \delta_t, \varepsilon, f) = \Big(\sigma_t - \frac{R_t(\varepsilon, f)}{\min_i p_t(i)}\Big)\Big(1 + \big(\frac{B_1}{B_2}\big)^2 - \frac{\mathcal{K}\delta_t}{B_2} - \frac{2\varepsilon}{B_2}\Big) - 1.$*

*Proof.* For any encoder $f$, let $S_t(\varepsilon, f) := \{\boldsymbol{x}_t \in \mathcal{X}_t : \sup_{\mathbf{x}_{t,1}, \mathbf{x}_{t,2} \in \mathcal{A}(\boldsymbol{x}_t)} \|f(\mathbf{x}_{t,1}) - f(\mathbf{x}_{t,2})\|_2 \le \varepsilon\}$, if any $\boldsymbol{x}_t \in \{\widetilde{C}_t(1) \cup \cdots \cup \widetilde{C}_t(K)\} \cap S_t(\varepsilon, f)$ can be correctly classified by $Q_f$, it turns out that $\mathrm{Err}(Q_f)$ can be bounded by $(1 - \sigma_t) + R_t(\varepsilon, f)$. In fact,

$$
\begin{aligned}
\mathrm{Err}(Q_f) = \mathbb{P}_t\Big\{Q_f(\boldsymbol{x}_t) \ne y\Big\} &\le \mathbb{P}_t\Big[\{\widetilde{C}_t(1) \cup \cdots \cup \widetilde{C}_t(K) \cap S_t(\varepsilon, f)\}^c\Big] \\
&= \mathbb{P}_t\Big[\{\widetilde{C}_t(1) \cup \cdots \cup \widetilde{C}_t(K)\}^c \cup \{S_t(\varepsilon, f)\}^c\Big] \le (1 - \sigma_t) + \mathbb{P}_t\Big[\{S_t(\varepsilon, f)\}^c\Big] \\
&= (1 - \sigma_t) + R_t(\varepsilon, f).
\end{aligned}
$$

The first row is derived from the definition of $\mathrm{Err}(Q_f)$. Since any $\boldsymbol{x}_t \in \{\widetilde{C}_t(1) \cup \cdots \cup \widetilde{C}_t(K)\} \cap S_t(\varepsilon, f)$ can be correctly classified by $Q_f$, we obtain the second row. De Morgan's laws imply the third row. The fourth row follows from Definition 3. Finally, noting that $R_t(\varepsilon, f) = \mathbb{P}_t[\{S_t(\varepsilon, f)\}^c]$ yields the last line.

Hence it suffices to show for given $i \in [K]$, $\boldsymbol{x}_t \in \widetilde{C}_t(i) \cap S_t(\varepsilon, f)$ can be correctly classified by $Q_f$ if for any $j \ne i$,

$$
\begin{aligned}
\mu_t(i)^\top \mu_t(j) < B_2^2\Big(\Gamma_i(\sigma_t, \delta_t, \varepsilon, f) &- \sqrt{2 - 2\Gamma_i(\sigma_t, \delta_t, \varepsilon, f)} - \frac{1}{2}\Big(1 - \frac{\min_{k \in [K]} \|\hat{\mu}_t(k)\|_2^2}{B_2^2}\Big) \\
&- \frac{\|\hat{\mu}_t(i) - \mu_t(i)\|_2}{B_2} - \frac{\|\hat{\mu}_t(j) - \mu_t(j)\|_2}{B_2}\Big),
\end{aligned}
$$

where $\Gamma_i(\sigma_t, \delta_t, \varepsilon, f) = \Big(\sigma_t - \frac{R_t(\varepsilon, f)}{p_t(i)}\Big)\Big(1 + \Big(\frac{B_1}{B_2}\Big)^2 - \frac{K\delta_t}{B_2} - \frac{2\varepsilon}{B_2}\Big) - 1$.

To this end, without losing generality, consider the case $i = 1$. To turn out $\boldsymbol{x}_t \in \widetilde{C}_t(1) \cap S_t(\varepsilon, f)$ can be correctly classified by $Q_f$, by the definition of $\widetilde{C}_t(1)$ and $S_t(\varepsilon, f)$, It just need to show $\forall k \ne 1, \|f(\boldsymbol{x}_t) - \hat{\mu}_t(1)\|_2 < \|f(\boldsymbol{x}_t) - \hat{\mu}_t(k)\|_2$, which is equivalent to

$$
f(\boldsymbol{x}_t)^\top \hat{\mu}_t(1) - f(\boldsymbol{x}_t)^\top \hat{\mu}_t(k) - \Big(\frac{1}{2}\|\hat{\mu}_t(1)\|_2^2 - \frac{1}{2}\|\hat{\mu}_t(k)\|_2^2\Big) > 0.
$$

We first deal with the term $f(\boldsymbol{x}_t)^\top \hat{\mu}_t(1)$,

$$
\begin{aligned}
f(\boldsymbol{x}_t)^\top \hat{\mu}_t(1) &= f(\boldsymbol{x}_t)^\top \mu_t(1) + f(\boldsymbol{x}_t)^\top \big(\hat{\mu}_t(1) - \mu_t(1)\big) \\
&\ge f(\boldsymbol{x}_t)^\top \mathbb{E}_{\boldsymbol{x}_t \in C_t(1)} \mathbb{E}_{\mathbf{x}_t \in \mathcal{A}(\boldsymbol{x}_t)}\big\{f(\mathbf{x}_t)\big\} - \big\|f(\boldsymbol{x}_t)\big\|_2 \big\|\hat{\mu}_t(1) - \mu_t(1)\big\|_2 \\
&\ge \frac{1}{p_t(1)} f(\boldsymbol{x}_t)^\top \mathbb{E}_{\boldsymbol{x}_t \sim \mathbb{P}_t} \mathbb{E}_{\mathbf{x}_t \in \mathcal{A}(\boldsymbol{x}_t)}\Big[f(\mathbf{x}_t)\mathbb{1}\big\{\boldsymbol{x}_t \in C_t(1)\big\}\Big] - B_2\big\|\hat{\mu}_t(1) - \mu_t(1)\big\|_2 \\
&= \frac{1}{p_t(1)} f(\boldsymbol{x}_t)^\top \mathbb{E}_{\boldsymbol{x}_t \sim \mathbb{P}_t} \mathbb{E}_{\mathbf{x}_t \in \mathcal{A}(\boldsymbol{x}_t)}\Big[f(\mathbf{x}_t)\mathbb{1}\big\{\boldsymbol{x}_t \in C_t(1) \cap \widetilde{C}_t(1) \cap S_t(\varepsilon, f)\big\}\Big] \\
&\quad + \frac{1}{p_t(1)} f(\boldsymbol{x}_t)^\top \mathbb{E}_{\boldsymbol{x}_t \sim \mathbb{P}_t} \mathbb{E}_{\mathbf{x}_t \in \mathcal{A}(\boldsymbol{x}_t)}\Big[f(\mathbf{x}_t)\mathbb{1}\Big\{\boldsymbol{x}_t \in C_t(1) \cap \{\widetilde{C}_t(1) \cap S_t(\varepsilon, f)\}^c\Big\}\Big] \\
&\qquad\qquad\qquad\qquad\qquad\qquad\qquad\qquad\qquad\qquad\qquad\qquad\qquad\qquad\qquad (8) \\
&\quad - B_2\big\|\hat{\mu}_t(1) - \mu_t(1)\big\|_2 \\
&= \frac{\mathbb{P}_t\{\widetilde{C}_t(1) \cap S_t(\varepsilon, f)\}}{p_t(1)} f(\boldsymbol{x}_t)^\top \mathbb{E}_{\boldsymbol{x}_t \in \widetilde{C}_t(1) \cap S_t(\varepsilon, f)} \mathbb{E}_{\mathbf{x}_t \in \mathcal{A}(\boldsymbol{x}_t)}\big\{f(\mathbf{x}_t)\big\} \\
&\quad + \frac{1}{p_t(1)} \mathbb{E}_{\boldsymbol{x}_t \sim \mathbb{P}_t}\Big[\mathbb{E}_{\mathbf{x}_t \in \mathcal{A}(\boldsymbol{x}_t)}\big\{f(\boldsymbol{x}_t)^\top f(\mathbf{x}_t)\big\}\mathbb{1}\big[\boldsymbol{x}_t \in C_t(1)\backslash\{\widetilde{C}_t(1) \cap S_t(\varepsilon, f)\}\big]\Big] \\
&\quad - B_2\|\hat{\mu}_t(1) - \mu_t(1)\|_2 \\
&\ge \frac{\mathbb{P}_t\{\widetilde{C}_t(1) \cap S_t(\varepsilon, f)\}}{p_t(1)} f(\boldsymbol{x}_t)^\top \mathop{\mathbb{E}}_{\boldsymbol{x}_t \in \widetilde{C}_t(1) \cap S_t(\varepsilon, f)} \mathop{\mathbb{E}}_{\mathbf{x}_t \in \mathcal{A}(\boldsymbol{x}_t)}\big\{f(\mathbf{x}_t)\big\} \\
&\quad - \frac{B_2^2}{p_t(1)}\mathbb{P}_t\Big[C_t(1)\backslash\{\widetilde{C}_t(1) \cap S_t(\varepsilon, f)\}\Big] - B_2\|\hat{\mu}_t(1) - \mu_t(1)\|_2. \qquad (9)
\end{aligned}
$$

The second row follows from the Cauchy–Schwarz inequality. The third and last rows are derived from the condition $\|f\|_2 \leq B_2$. Note that

$$\mathbb{P}_t\Big[C_t(1)\backslash\{\widetilde{C}_t(1)\cap S_t(\varepsilon,f)\}\Big] = \mathbb{P}_t\Big[\{C_t(1)\backslash\widetilde{C}_t(1)\}\cup\big[\widetilde{C}_t(1)\cap\{S_t(\varepsilon,f)\}^c\big]\Big] \tag{10}$$

$$\leq (1-\sigma_t)p_t(1) + R_t(\varepsilon,f), \tag{11}$$

and

$$\mathbb{P}_t\big(\widetilde{C}_t(1)\cap S_t(\varepsilon,f)\big) = \mathbb{P}_t\big(C_t(1)\big) - \mathbb{P}_t\big(C_t(1)\backslash(\widetilde{C}_t(1)\cap S_t(\varepsilon,f))\big) \tag{12}$$

$$\geq p_t(1) - \big\{(1-\sigma_t)p_t(1) + R_t(\varepsilon,f)\big\}$$

$$= \sigma_t p_t(1) - R_t(\varepsilon,f). \tag{13}$$

Plugging (10) and (12) into (8) yields

$$f(\boldsymbol{x}_t)^\top\hat{\mu}_t(1) \geq \Big(\sigma_t - \frac{R_t(\varepsilon,f)}{p_t(1)}\Big)f(\boldsymbol{x}_t)^\top \underset{\boldsymbol{x}_t\in\widetilde{C}_t(1)\cap S_t(\varepsilon,f)}{\mathbb{E}}\underset{\mathbf{x}_t\in\mathcal{A}(\boldsymbol{x}_t)}{\mathbb{E}}\{f(\mathbf{x}_t)\} - B_2^2\Big(1 - \sigma_t + \frac{R_t(\varepsilon,f)}{p_t(1)}\Big)$$

$$- B_2\left\|\hat{\mu}_t(1) - \mu_t(1)\right\|_2. \tag{14}$$

Notice that $\boldsymbol{x}_t \in \widetilde{C}_t(1) \cap S_t(\varepsilon,f)$. Thus, for any $\boldsymbol{x}'_t \in \widetilde{C}_t(1) \cap S_t(\varepsilon,f)$, by the definition of $\widetilde{C}_t(1)$, we have $\min_{\mathbf{x}_t\in\mathcal{A}(\boldsymbol{x}_t),\mathbf{x}'_t\in\mathcal{A}(\boldsymbol{x}_t)}\|\mathbf{x}_t - \mathbf{x}'_t\|_2 \leq \delta_t$. Further, denote $(\mathbf{x}_t^*, \mathbf{x}'^*_t) = \arg\min_{\mathbf{x}_t\in\mathcal{A}(\boldsymbol{x}_t),\mathbf{x}'_t\in\mathcal{A}(\boldsymbol{x}_t)}\|\mathbf{x}_t - \mathbf{x}'_t\|_2$. Then, we have $\|\mathbf{x}_t^* - \mathbf{x}'^*_t\|_2 \leq \delta_t$. Combining this with the $\mathcal{K}$-Lipschitz property of $f$, we obtain $\|f(\mathbf{x}_t^*) - f(\mathbf{x}'^*_t)\|_2 \leq \mathcal{K}\|\mathbf{x}_t^* - \mathbf{x}'^*_t\|_2 \leq \mathcal{K}\delta_t$. Moreover, since $\boldsymbol{x}_t \in S_t(\varepsilon,f)$, it follows that for all $\mathbf{x}'_t \in \mathcal{A}(\boldsymbol{x}_t)$, $\|f(\mathbf{x}'_t) - f(\mathbf{x}'^*_t)\|_2 \leq \varepsilon$. Similarly, as $\boldsymbol{x}_t \in S_t(\varepsilon,f)$ and both $\mathbf{x}_t$ and $\mathbf{x}_t^*$ belong to $\mathcal{A}(\boldsymbol{x}_t)$, we know $\|f(\mathbf{x}_t) - f(\mathbf{x}_t^*)\|_2 \leq \varepsilon$.

Therefore,

$$f(\boldsymbol{x}_t)^\top\mathbb{E}_{\boldsymbol{x}_t\in\widetilde{C}_t(1)\cap S_t(\varepsilon,f)}\mathbb{E}_{\mathbf{x}_t\in\mathcal{A}(\boldsymbol{x}_t)}\{f(\mathbf{x}_t)\} = \mathbb{E}_{\boldsymbol{x}_t\in\widetilde{C}_t(1)\cap S_t(\varepsilon,f)}\mathbb{E}_{\mathbf{x}_t\in\mathcal{A}(\boldsymbol{x}_t)}\{f(\boldsymbol{x}_t)^\top f(\mathbf{x}_t)\}$$

$$= \mathbb{E}_{\boldsymbol{x}_t\in\widetilde{C}_t(1)\cap S_t(\varepsilon,f)}\mathbb{E}_{\mathbf{x}_t\in\mathcal{A}(\boldsymbol{x}_t)}\Big[f(\boldsymbol{x}_t)^\top\{f(\mathbf{x}_t) - f(\boldsymbol{x}_t) + f(\boldsymbol{x}_t)\}\Big]$$

$$\geq B_1^2 + \mathbb{E}_{\boldsymbol{x}_t\in\widetilde{C}_t(1)\cap S_t(\varepsilon,f)}\mathbb{E}_{\mathbf{x}_t\in\mathcal{A}(\boldsymbol{x}_t)}\Big[f(\boldsymbol{x}_t)^\top\{f(\mathbf{x}_t) - f(\boldsymbol{x}_t)\}\Big]$$

$$= B_1^2 + \mathbb{E}_{\boldsymbol{x}_t\in\widetilde{C}_t(1)\cap S_t(\varepsilon,f)}\mathbb{E}_{\mathbf{x}_t\in\mathcal{A}(\boldsymbol{x}_t)}\Big[f(\boldsymbol{x}_t)^\top\{\underbrace{f(\mathbf{x}_t) - f(\mathbf{x}'^*_t)}_{\|\cdot\|_2\leq\varepsilon} + \underbrace{f(\mathbf{x}'^*_t) - f(\mathbf{x}_t^*)}_{\|\cdot\|_2\leq\mathcal{K}\delta_t} + \underbrace{f(\mathbf{x}_t^*) - f(\boldsymbol{x}_t)}_{\|\cdot\|_2\leq\varepsilon}\}\Big]$$

$$\geq B_1^2 - \big(B_2\varepsilon + B_2\mathcal{K}\delta_t + B_2\varepsilon\big)$$

$$= B_1^2 - B_2\big(\mathcal{K}\delta_t + 2\varepsilon\big), \tag{15}$$

where the fourth line is derived from $\|f\|_2 \geq B_1$.

Plugging (15) into the inequality (14) yields

$$f(\boldsymbol{x}_t)^\top\hat{\mu}_t(1) \geq \Big(\sigma_t - \frac{R_t(\varepsilon,f)}{p_t(1)}\Big)f(\boldsymbol{x}_t)^\top\underset{\boldsymbol{x}_t\in\widetilde{C}_t(1)\cap S_t(\varepsilon,f)}{\mathbb{E}}\underset{\mathbf{x}_t\in\mathcal{A}(\boldsymbol{x}_t)}{\mathbb{E}}\{f(\mathbf{x}_t)\} - B_2^2\Big(1 - \sigma_t + \frac{R_t(\varepsilon,f)}{p_t(1)}\Big)$$

$$- B_2\big\|\hat{\mu}_t(1) - \mu_t(1)\big\|_2$$

$$\geq \Big(\sigma_t - \frac{R_t(\varepsilon,f)}{p_t(1)}\Big)\Big(B_1^2 - B_2(\mathcal{K}\delta_t + 2\varepsilon)\Big) - B_2^2\Big\{1 - \sigma_t + \frac{R_t(\varepsilon,f)}{p_t(1)}\Big\}$$

$$- B_2\big\|\hat{\mu}_t(1) - \mu_t(1)\big\|_2$$

$$= B_2^2\Big\{\Big(1 + \Big(\frac{B_1}{B_2}\Big)^2\Big)\Big(\sigma_t - \frac{R_t(\varepsilon,f)}{p_t(1)}\Big) - \Big(\sigma_t - \frac{R_t(\varepsilon,f)}{p_t(1)}\Big)\Big(\frac{\mathcal{K}\delta_t}{B_2} + \frac{2\varepsilon}{B_2}\Big) - 1\Big\}$$

$$- B_2\big\|\hat{\mu}_t(1) - \mu_t(1)\big\|_2$$

$$= B_2^2\Big\{\Big(\sigma_t - \frac{R_t(\varepsilon,f)}{p_t(1)}\Big)\Big(1 + \Big(\frac{B_1}{B_2}\Big)^2 - \frac{\mathcal{K}\delta_t}{B_2} - \frac{2\varepsilon}{B_2}\Big) - 1\Big\} - B_2\big\|\hat{\mu}_t(1) - \mu_t(1)\big\|_2$$

$$= B_2^2\Gamma_1(\sigma_t,\delta_t,\varepsilon,f) - B_2\big\|\hat{\mu}_t(1) - \mu_t(1)\big\|_2.$$

Similar process can also turn out

$$f(\boldsymbol{x}_t)^\top\mu_t(1) \geq B_2^2\Gamma_1(\sigma_t,\delta_t,\varepsilon,f). \tag{16}$$

Combining with $\left\|\mu_t(k)\right\|_2 = \left\|\mathbb{E}_{\boldsymbol{x}_t \in \widetilde{C}_t(k)}\mathbb{E}_{\mathrm{x}_t \in \mathcal{A}(\boldsymbol{x}_t)}\{f(\mathrm{x}_t)\}\right\|_2 \le \mathbb{E}_{\boldsymbol{x}_s \in \widetilde{C}_t(k)}\mathbb{E}_{\mathrm{x}_t \in \mathcal{A}(\boldsymbol{x}_t)}\left\|f(\mathrm{x}_t)\right\|_2 \le B_2$ yields

$$
\begin{aligned}
f(\boldsymbol{x}_t)^\top \hat{\mu}_t(k) &\le f(\boldsymbol{x}_t)^\top \mu_t(k) + f(\boldsymbol{x}_t)^\top\big(\hat{\mu}_t(k) - \mu_t(k)\big)\\
&\le f(\boldsymbol{x}_t)^\top \mu_t(k) + \left\|f(\boldsymbol{x}_t)\right\|_2\left\|\hat{\mu}_t(k) - \mu_t(k)\right\|_2\\
&\le f(\boldsymbol{x}_t)^\top \mu_t(k) + B_2\left\|\hat{\mu}_t(k) - \mu_t(k)\right\|_2\\
&= \{f(\boldsymbol{x}_t) - \mu_t(1)\}^\top \mu_t(k) + \mu_t(1)^\top \mu_t(k) + B_2\left\|\hat{\mu}_t(k) - \mu_t(k)\right\|_2\\
&\le \left\|f(\boldsymbol{x}_t) - \mu_t(1)\right\|_2 \cdot \left\|\mu_t(k)\right\|_2 + \mu_t(1)^\top \mu_t(k) + B_2\left\|\hat{\mu}_t(k) - \mu_t(k)\right\|_2\\
&\le B_2\sqrt{\left\|f(\boldsymbol{x}_t)\right\|_2^2 - 2f(\boldsymbol{x}_t)^\top \mu_t(1) + \left\|\mu_t(1)_2\right\|^2} + \mu_t(1)^\top \mu_t(k) + B_2\left\|\hat{\mu}_t(k) - \mu_t(k)\right\|_2\\
&\le B_2\sqrt{2B_2^2 - 2f(\boldsymbol{x}_t)^\top \mu_t(1)} + \mu_t(1)^\top \mu_t(k) + B_2\left\|\hat{\mu}_t(k) - \mu_t(k)\right\|_2\\
&\le B_2\sqrt{2B_2^2 - 2B_2^2\Gamma_1(\sigma_t, \delta_t, \varepsilon, f)} + \mu_t(1)^\top \mu_t(k) + B_2\left\|\hat{\mu}_t(k) - \mu_t(k)\right\|_2\\
&= \sqrt{2}B_2^2\sqrt{1 - \Gamma_1(\sigma_t, \delta_t, \varepsilon, f)} + \mu_t(1)^\top \mu_t(k) + B_2\left\|\hat{\mu}_t(k) - \mu_t(k)\right\|_2,
\end{aligned}
$$

where the inequality in eighth row stems from (16). Moreover, we can conclude

$$
\begin{aligned}
&f(\boldsymbol{x}_t)^\top \hat{\mu}_t(1) - f(\boldsymbol{x}_t)^\top \hat{\mu}_t(k) - \Big(\frac{1}{2}\left\|\hat{\mu}_t(1)\right\|_2^2 - \frac{1}{2}\left\|\hat{\mu}_t(k)\right\|^2\Big)\\
&= f(\boldsymbol{x}_t)^\top \hat{\mu}_t(1) - f(\boldsymbol{x}_t)^\top \hat{\mu}_t(k) - \frac{1}{2}\left\|\hat{\mu}_t(1)\right\|_2^2 + \frac{1}{2}\left\|\hat{\mu}_t(k)\right\|_2^2\\
&\ge f(\boldsymbol{x}_t)^\top \hat{\mu}_t(1) - f(\boldsymbol{x}_t)^\top \hat{\mu}_t(k) - \frac{1}{2}B_2^2 + \frac{1}{2}\min_{k\in[K]}\left\|\hat{\mu}_t(k)\right\|_2^2\\
&\ge B_2^2\Gamma_1(\sigma_t, \delta_t, \varepsilon, f) - B_2\left\|\hat{\mu}_t(1) - \mu_t(1)\right\|_2 - \sqrt{2}B_2^2\sqrt{1 - \Gamma_1(\sigma_t, \delta_t, \varepsilon, f)} - \mu_t(1)^\top \mu_t(k)\\
&\quad - B_2\left\|\hat{\mu}_t(k) - \mu_t(k)\right\|_2 - \frac{1}{2}B_2^2\Big(1 - \frac{\min_{k\in[K]}\left\|\hat{\mu}_t(k)\right\|_2^2}{B_2^2}\Big) > 0,
\end{aligned}
$$

where the last inequality is derived from the given condition in Lemma 1, which finishes the proof. $\square$

## F.3 The effect of minimaxing Adv-SSL

In this section, we explore the effect of minimaxing the risk of Adv-SSL, as demonstrated in Lemma 5. We begin by showing that the required condition in Lemma 1 can indeed be satisfied by our method. To achieve this, we first introduce Lemma 2, Lemma 3 as preparatory steps. We will begin with reviewing and introducing some necessary notations at first.

Review that $p_s(k) = \mathbb{P}_s\big(\boldsymbol{x}_s \in C_s(k)\big)$ and $\mathbb{P}_s(k)(\cdot) = \mathbb{P}_s\big(\cdot \,|\, \boldsymbol{x}_s \in C_s(k)\big)$. Correspondingly, $p_t(k) = \mathbb{P}_t\big(\boldsymbol{x}_t \in C_t(k)\big)$ and $\mathbb{P}_t(k)(\cdot) = \mathbb{P}_t\big(\cdot \,|\, \boldsymbol{x}_t \in C_t(k)\big)$. We use the quantities

$$
\epsilon_1 = \max_{k\in[K]}\mathcal{W}\big(\mathbb{P}_s(k), \mathbb{P}_t(k)\big), \quad \epsilon_2 = \max_{k\in[K]}\left|p_s(k) - p_t(k)\right|, \tag{17}
$$

to measure the divergence between the source and the target domains. In addition, Following the notations in the target domain, we denote the center of the $k$-th latent class in the representation space as $\mu_s(k) := \mathbb{E}_{\boldsymbol{x}_s \in C_s(k)}\mathbb{E}_{\mathrm{x}_s \in \mathcal{A}(\boldsymbol{x}_s)}\{f(\mathrm{x}_s)\} = \frac{1}{p_s(k)}\mathbb{E}_{\boldsymbol{x}_s \sim \mathbb{P}_s}\mathbb{E}_{\mathrm{x}_s \in \mathcal{A}(\boldsymbol{x}_s)}\big[f(\mathrm{x}_s)\mathbb{1}\big\{\boldsymbol{x}_s \in C_s(k)\big\}\big]$. In this context, the Lemma 2 can be presented as follow:

**Lemma 2.** *If the encoder $f$ is $\mathcal{K}$-Lipschitz continuous, then for any $k \in [K]$,*

$$
\left\|\mu_s(k) - \mu_t(k)\right\|_2 \le \sqrt{d^*}M\mathcal{K}\epsilon_1.
$$

*Proof.* For all $k \in [K]$,

$$
\left\|\mu_s(k) - \mu_t(k)\right\|_2^2 = \sum_{l=1}^{d^*}\Big[\{\mu_s(k)\}_l - \{\mu_t(k)\}_l\Big]^2
$$

$$= \sum_{i=1}^{d^*} \Big[ \mathbb{E}_{\boldsymbol{x}_s \in C_s(k)} \mathbb{E}_{\mathbf{x}_s \in \mathcal{A}(\boldsymbol{x}_s)} \big\{ f_i(\mathbf{x}_s) \big\} - \mathbb{E}_{\boldsymbol{x}_t \in C_t(k)} \mathbb{E}_{\mathbf{x}_t \in \mathcal{A}(\boldsymbol{x}_t)} \big\{ f_i(\mathbf{x}_t) \big\} \Big]^2$$

$$= \sum_{i=1}^{d^*} \Big[ \frac{1}{m} \sum_{j=1}^{m} \big( \mathbb{E}_{\boldsymbol{x}_s \in C_s(k)} \{ f_i(A_j(\boldsymbol{x}_s)) \} - \mathbb{E}_{\boldsymbol{x}_t \in C_t(k)} \{ f_i(A_j(\boldsymbol{x}_t)) \} \big) \Big]^2$$

$$\leq d^* M^2 \mathcal{K}^2 \epsilon_1^2.$$

The final inequality is obtained from $\epsilon_1 = \max_{k \in [K]} \mathcal{W}\big(\mathbb{P}_s(k), \mathbb{P}_t(k)\big)$ and the definition of Wasserstein distance, along with the fact that $f\big(A_j(\cdot)\big)$ is $M\mathcal{K}$-Lipschitz continuous. In fact, since $f \in \mathrm{Lip}(\mathcal{K})$, it follows that for every $i \in [d^*]$, $f_i \in \mathrm{Lip}(\mathcal{K})$. Combining this with the property that $A_j(\cdot) \in \mathrm{Lip}(M)$ stated in Assumption 3, we conclude that $f\big(A_i(\cdot)\big)$ is $M\mathcal{K}$-Lipschitz continuous. So that

$$\big\| \mu_s(k) - \mu_t(k) \big\|_2 \leq \sqrt{d^*} M \mathcal{K} \epsilon_1.$$

$\square$

Next we present Lemma 3.

**Lemma 3.** *Given a $(\sigma_s, \sigma_t, \delta_s, \delta_t)$-augmentation, if the encoder $f$ with $\|f\|_2 \leq B_2$ is $\mathcal{K}$-Lipschitz continuous, then*

$$\mathbb{E}_{\boldsymbol{x}_s \in C_s(k)} \mathbb{E}_{\mathbf{x}_s \in \mathcal{A}(\boldsymbol{x}_s)} \big\| f(\mathbf{x}_s) - \mu_s(k) \big\|_2^2 \leq 4B_2^2 \Big\{ \Big( 1 - \sigma_s + \frac{\mathcal{K}\delta_s + 2\varepsilon}{2B_2} + \frac{R_s(\varepsilon, f)}{p_s(k)} \Big)^2 + \Big( 1 - \sigma_s + \frac{R_s(\varepsilon, f)}{p_s(k)} \Big) \Big\},$$

*where $R_s(\varepsilon, f) = \mathbb{P}_s \Big\{ \boldsymbol{x}_s \in \mathcal{X}_s : \sup_{\mathbf{x}_{s,1}, \mathbf{x}_{s,2} \in \mathcal{A}(\boldsymbol{x}_s)} \| f(\mathbf{x}_{s,1}) - f(\mathbf{x}_{s,2}) \|_2 > \varepsilon \Big\}.$*

*Proof.* Let $S_s(\varepsilon, f) := \Big\{ \boldsymbol{x}_s \in \mathcal{X}_s : \sup_{\mathbf{x}_{s,1}, \mathbf{x}_{s,2} \in \mathcal{A}(\boldsymbol{x}_s)} \big\| f(\mathbf{x}_{s,1}) - f(\mathbf{x}_{s,2}) \big\|_2 \leq \varepsilon \Big\}$, for each $k \in [K]$,

$$\mathbb{E}_{\boldsymbol{x}_s \in C_s(k)} \mathbb{E}_{\mathbf{x}_s \in \mathcal{A}(\boldsymbol{x}_s)} \big\| f(\mathbf{x}_s) - \mu_s(k) \big\|_2^2 = \frac{1}{p_s(k)} \mathbb{E}_{\boldsymbol{x}_s \sim \mathbb{P}_s} \mathbb{E}_{\mathbf{x}_s \in \mathcal{A}(\boldsymbol{x}_s)} \Big[ \mathbb{1} \big\{ \boldsymbol{x}_s \in C_s(k) \big\} \big\| f(\mathbf{x}_s) - \mu_s(k) \big\|_2^2 \Big]$$

$$= \frac{1}{p_s(k)} \mathbb{E}_{\boldsymbol{x}_s \sim \mathbb{P}_s} \mathbb{E}_{\mathbf{x}_s \in \mathcal{A}(\boldsymbol{x}_s)} \Big[ \mathbb{1} \big\{ \boldsymbol{x}_s \in \widetilde{C}_s(k) \cap S_s(\varepsilon, f) \big\} \big\| f(\mathbf{x}_s) - \mu_s(k) \big\|_2^2 \Big]$$

$$\quad + \frac{1}{p_s(k)} \mathbb{E}_{\boldsymbol{x}_s \sim \mathbb{P}_s} \mathbb{E}_{\mathbf{x}_s \in \mathcal{A}(\boldsymbol{x}_s)} \Big[ \mathbb{1} \big\{ \boldsymbol{x}_s \in C_s(k) \backslash (\widetilde{C}_s(k) \cap S_s(\varepsilon, f)) \big\} \big\| f(\mathbf{x}_s) - \mu_s(k) \big\|_2^2 \Big]$$

$$\leq \frac{1}{p_s(k)} \mathbb{E}_{\boldsymbol{x}_s \sim \mathbb{P}_s} \mathbb{E}_{\mathbf{x}_s \in \mathcal{A}(\boldsymbol{x}_s)} \Big[ \mathbb{1} \big\{ \boldsymbol{x}_s \in \widetilde{C}_s(k) \cap S_s(\varepsilon, f) \big\} \big\| f(\mathbf{x}_s) - \mu_s(k) \big\|_2^2 \Big] + \frac{4B_2^2 \mathbb{P}_s \big[ C_s(k) \backslash \{ \widetilde{C}_s(k) \cap S_s(\varepsilon, f) \} \big]}{p_s(k)}$$

$$\leq \frac{1}{p_s(k)} \mathbb{E}_{\boldsymbol{x}_s \sim \mathbb{P}_s} \mathbb{E}_{\mathbf{x}_s \in \mathcal{A}(\boldsymbol{x}_s)} \Big[ \mathbb{1} \big\{ \boldsymbol{x}_s \in \widetilde{C}_s(k) \cap S_s(\varepsilon, f) \big\} \big\| f(\mathbf{x}_s) - \mu_s(k) \big\|_2^2 \Big] + 4B_2^2 \Big( 1 - \sigma_s + \frac{R_s(\varepsilon, f)}{p_s(k)} \Big)$$

$$\leq \frac{\mathbb{P}_s \big( \widetilde{C}_s(k) \cap S_s(\varepsilon, f) \big)}{p_s(k)} \mathbb{E}_{\boldsymbol{x}_s \in \widetilde{C}_s(k) \cap S_s(\varepsilon, f)} \mathbb{E}_{\mathbf{x}_s \in \mathcal{A}(\boldsymbol{x}_s)} \big\| f(\mathbf{x}_s) - \mu_s(k) \big\|_2^2 + 4B_2^2 \Big( 1 - \sigma_s + \frac{R_s(\varepsilon, f)}{p_s(k)} \Big)$$

$$\leq \mathbb{E}_{\boldsymbol{x}_s \in \widetilde{C}_s(k) \cap S_s(\varepsilon, f)} \mathbb{E}_{\mathbf{x}_s \in \mathcal{A}(\boldsymbol{x}_s)} \big\| f(\mathbf{x}_s) - \mu_s(k) \big\|_2^2 + 4B_2^2 \Big( 1 - \sigma_s + \frac{R_s(\varepsilon, f)}{p_s(k)} \Big), \tag{18}$$

where the second inequality is due to

$$\mathbb{P}_s \Big[ C_s(k) \backslash \{ \widetilde{C}_s(k) \cap S_s(\varepsilon, f) \} \Big] = \mathbb{P}_s \Big[ \big\{ C_s(k) \backslash \widetilde{C}_s(k) \big\} \cup \big\{ C_s(k) \backslash S_s(\varepsilon, f) \big\} \Big]$$

$$\leq \big( 1 - \sigma_s \big) p_s(k) + R_s(\varepsilon, f).$$

Furthermore,

$$\mathbb{E}_{\boldsymbol{x}_s \in \widetilde{C}_s(k) \cap S_s(\varepsilon, f)} \mathbb{E}_{\mathbf{x}_s \in \mathcal{A}(\boldsymbol{x}_s)} \big\| f(\mathbf{x}_s) - \mu_s(k) \big\|_2^2 \tag{19}$$

$$= \mathbb{E}_{\boldsymbol{x}_s \in \widetilde{C}_s(k) \cap S_s(\varepsilon, f)} \mathbb{E}_{\mathbf{x}_s \in \mathcal{A}(\boldsymbol{x}_s)} \Big\| f(\mathbf{x}_s) - \mathbb{E}_{\boldsymbol{x}'_s \in C_s(k)} \mathbb{E}_{\mathbf{x}'_s \in \mathcal{A}(\boldsymbol{x}'_s)} \big\{ f(\mathbf{x}'_s) \big\} \Big\|_2^2$$

$$= \mathbb{E}_{\boldsymbol{x}_s \in \widetilde{C}_s(k) \cap S_s(\varepsilon, f)} \mathbb{E}_{\mathbf{x}_s \in \mathcal{A}(\boldsymbol{x}_s)} \Big\| f(\mathbf{x}_s) - \frac{\mathbb{P}_s \{ \widetilde{C}_s(k) \cap S_s(\varepsilon, f) \}}{p_s(k)} \mathbb{E}_{\boldsymbol{x}'_s \in \widetilde{C}_s(k) \cap S_s(\varepsilon, f)} \mathbb{E}_{\mathbf{x}'_s \in \mathcal{A}(\mathbf{x}_s)} \big\{ f(\mathbf{x}'_s) \big\}$$

$$- \frac{\mathbb{P}_s\big[C_s(k)\backslash\{\widetilde{C}_s(k)\cap S_s(\varepsilon,f)\}\big]}{p_s(k)}\mathbb{E}_{\boldsymbol{x}'_s\in C_s(k)\backslash\{\widetilde{C}_s(k)\cap S_s(\varepsilon,f)\}}\mathbb{E}_{\mathbf{x}'_s\in\mathcal{A}(\boldsymbol{x}'_s)}\big\{f(\mathbf{x}'_s)\big\}\Big\|_2^2$$

$$= \mathbb{E}_{\boldsymbol{x}_s\in\widetilde{C}_s(k)\cap S_s(\varepsilon,f)}\mathbb{E}_{\mathbf{x}_s\in\mathcal{A}(\boldsymbol{x}_s)}\Big\|\frac{\mathbb{P}_s\{\widetilde{C}_s(k)\cap S_s(\varepsilon,f)\}}{p_s(k)}\Big(f(\mathbf{x}_s)-\mathbb{E}_{\boldsymbol{x}'_s\in\widetilde{C}_s(k)\cap S_s(\varepsilon,f)}\mathbb{E}_{\mathbf{x}'_s\in\mathcal{A}(\boldsymbol{x}'_s)}\big\{f(\mathbf{x}'_s)\big\}\Big)$$

$$- \frac{\mathbb{P}_s\big[C_s(k)\backslash\{\widetilde{C}_s(k)\cap S_s(\varepsilon,f)\}\big]}{p_s(k)}\Big(f(\mathbf{x}_s)-\mathbb{E}_{\boldsymbol{x}'_s\in C_s(k)\backslash\{\widetilde{C}_s(k)\cap S_s(\varepsilon,f)\}}\mathbb{E}_{\mathbf{x}'_s\in\mathcal{A}(\boldsymbol{x}'_s)}\big\{f(\mathbf{x}'_s)\big\}\Big)\Big\|_2^2$$

$$\leq \mathop{\mathbb{E}}_{\boldsymbol{x}_s\in\widetilde{C}_s(k)\cap S_s(\varepsilon,f)}\mathop{\mathbb{E}}_{\mathbf{x}_s\in\mathcal{A}(\boldsymbol{x}_s)}\Big[\Big\|f(\mathbf{x}_s)-\mathop{\mathbb{E}}_{\boldsymbol{x}_s\in\widetilde{C}_s(k)\cap S_s(\varepsilon,f)}\mathop{\mathbb{E}}_{\mathbf{x}'_s\in\mathcal{A}(\boldsymbol{x}'_s)}\big\{f(\mathbf{x}'_s)\big\}\Big\|_2+2B_2\Big(1-\sigma_s+\frac{R_s(\varepsilon,f)}{p_s(k)}\Big)\Big]^2$$

$$\tag{20}$$

For any $\boldsymbol{x}_s,\boldsymbol{x}'_s\in\widetilde{C}_s(k)\cap S_s(\varepsilon,f)$, by the definition of $\widetilde{C}_s(k)$, we can yield

$$\min_{\mathbf{x}_s\in\mathcal{A}(\boldsymbol{x}_s),\mathbf{x}'_s\in\mathcal{A}(\boldsymbol{x}'_s)}\|\mathbf{x}_s-\mathbf{x}'_s\|_2\leq\delta_s,$$

Thus, let $(\mathbf{x}_s^*,\mathbf{x}_s'^*)=\arg\min_{\mathbf{x}_s\in\mathcal{A}(\boldsymbol{x}_s),\mathbf{x}'_s\in\mathcal{A}(\boldsymbol{x}'_s)}\|\mathbf{x}_s-\mathbf{x}'_s\|_2$, we have $\|\mathbf{x}_s^*-\mathbf{x}_s'^*\|_2\leq\delta_s$. Furthermore, by the $\mathcal{K}$-Lipschitz continuity of $f$, we yield $\|f(\mathbf{x}_s^*)-f(\mathbf{x}_s'^*)\|_2\leq\mathcal{K}\|\mathbf{x}_s^*-\mathbf{x}_s'^*\|_2\leq\mathcal{K}\delta_s$. In addition, since $\boldsymbol{x}_s\in S_s(\varepsilon,f)$, we know for any $\mathbf{x}_s\in\mathcal{A}(\boldsymbol{x}_s),\|f(\mathbf{x}_s)-f(\mathbf{x}_s^*)\|_2\leq\varepsilon$. Similarly, $\boldsymbol{x}'_s\in S_s(\varepsilon,f)$ implies $\|f(\mathbf{x}'_s)-f(\mathbf{x}_s'^*)\|_2\leq\varepsilon$ for any $\mathbf{x}'_s\in\mathcal{A}(\boldsymbol{x}'_s)$. Therefore, for any $\boldsymbol{x}_s,\boldsymbol{x}'_s\in\widetilde{C}_s(1)\cap S_s(\varepsilon,f)$ and $\mathbf{x}_s\in\mathcal{A}(\boldsymbol{x}_s),\mathbf{x}'_s\in\mathcal{A}(\boldsymbol{x}'_s)$,

$$\begin{aligned}\big\|f(\mathbf{x}_s)-f(\mathbf{x}'_s)\big\|_2 &\leq \big\|f(\mathbf{x}_s)-f(\mathbf{x}_s^*)\big\|_2+\big\|f(\mathbf{x}_s^*)-f(\mathbf{x}_s'^*)\big\|_2+\big\|f(\mathbf{x}_s'^*)-f(\mathbf{x}'_s)\big\|_2\\ &\leq 2\varepsilon+\mathcal{K}\delta_s.\end{aligned}\tag{21}$$

Combining inequalities (18), (19) and (21) concludes

$$\begin{aligned}&\mathbb{E}_{\boldsymbol{x}_s\in C_s(k)}\mathbb{E}_{\mathbf{x}_s\in\mathcal{A}(\boldsymbol{x}_s)}\big\|f(\mathbf{x}_s)-\mu_s(k)\big\|_2^2\\ &\leq\Big[2\varepsilon+\mathcal{K}\delta_s+2B_2\Big(1-\sigma_s+\frac{R_s(\varepsilon,f)}{p_s(k)}\Big)\Big]^2+4B_2^2\Big(1-\sigma_s+\frac{R_s(\varepsilon,f)}{p_s(k)}\Big)\\ &=4B_2^2\Big[\Big(1-\sigma_s+\frac{\mathcal{K}\delta_s}{2B_2}+\frac{\varepsilon}{B_2}+\frac{R_s(\varepsilon,f)}{p_s(k)}\Big)^2+\Big(1-\sigma_s+\frac{R_s(\varepsilon,f)}{p_s(k)}\Big)\Big]\end{aligned}$$

$\square$

Subsequently, we state Lemma 4 to establish the connection between Adv-SSL and the requirements shown in Lemma 1. Following lemma reveals the upstream task can indeed render the representation space well-structured.

**Lemma 4.** *Given a $(\sigma_s,\sigma_t,\delta_s,\delta_t)$-augmentation, if $d^*>K$ and the encoder $f$ with $B_1\leq\|f\|_2\leq B_2$ is $\mathcal{K}$-Lipschitz continuous, then for any $\varepsilon>0$,*

$$R_s^2(\varepsilon,f)\leq\frac{m^4}{\varepsilon^2}\mathcal{L}_{\mathrm{align}}(f),$$

$$R_t^2(\varepsilon,f)\leq\frac{m^4}{\varepsilon^2}\mathcal{L}_{\mathrm{align}}(f)+\frac{8m^4}{\varepsilon^2}B_2d^*M\mathcal{K}\epsilon_1+\frac{4m^4}{\varepsilon^2}B_2^2d^*K\epsilon_2,$$

*and*

$$\max_{i\neq j}\big|\mu_t(i)^\top\mu_t(j)\big|\leq\sqrt{\frac{2}{\min_{i\neq j}p_s(i)p_s(j)}\big\{\mathcal{R}(f)+\varphi(\sigma_s,\delta_s,\varepsilon,f)\big\}}+2\sqrt{d^*}B_2M\mathcal{K}\epsilon_1.$$

*where* $\varphi(\sigma_s,\delta_s,\varepsilon,f):=4B_2^2\Big[\Big(1-\sigma_s+\frac{\mathcal{K}\delta_s+2\varepsilon}{2B_2}\Big)^2+(1-\sigma_s)+KR_s(\varepsilon,f)\Big(3-2\sigma_s+\frac{\mathcal{K}\delta_s+2\varepsilon}{B_2}\Big)+R_s^2(\varepsilon,f)\Big(\sum_{k=1}^K\frac{1}{p_s(k)}\Big)\Big]+B_2\big(\varepsilon^2+4B_2^2R_s(\varepsilon,f)\big)^{\frac{1}{2}}.$

*Proof.* Since the measure on $\mathcal{A}$ is uniform, we have

$$\mathbb{E}_{\mathbf{x}_{t,1},\mathbf{x}_{t,2}\in\mathcal{A}(\boldsymbol{x}_t)}\big\|f(\mathbf{x}_{t,1})-f(\mathbf{x}_{t,2})\big\|_2=\frac{1}{m^2}\sum_{i,j=1}^m\big\|f\big(A_i(\boldsymbol{x}_t)\big)-f\big(A_j(\boldsymbol{x}_t)\big)\big\|_2,$$

hence,

$$\sup_{\mathbf{x}_{t,1},\mathbf{x}_{t,2}\in\mathcal{A}(\boldsymbol{x}_t)}\big\|f(\mathbf{x}_{t,1})-f(\mathbf{x}_{t,2})\big\|_2 = \sup_{i,j\in[m]}\big\|f\big(A_i(\boldsymbol{x}_t)\big)-f\big(A_j(\boldsymbol{x}_t)\big)\big\|_2$$

$$\leq \sum_{i,j=1}^{m}\big\|f\big(A_i(\boldsymbol{x}_t)\big)-f\big(A_j(\boldsymbol{x}_t)\big)\big\|_2$$

$$= m^2\mathbb{E}_{\mathbf{x}_{t,1},\mathbf{x}_{t,2}\in\mathcal{A}(\boldsymbol{x}_t)}\big\|f(\mathbf{x}_{t,1})-f(\mathbf{x}_{t,2})\big\|_2.$$

Denote $S := \big\{\boldsymbol{x}_t : \mathbb{E}_{\mathbf{x}_{t,1},\mathbf{x}_{t,2}\in\mathcal{A}(\boldsymbol{x}_t)}\big\|f(\mathbf{x}_{t,1})-f(\mathbf{x}_{t,2})\big\|_2 > \frac{\varepsilon}{m^2}\big\}$, by the definition of $R_t(\varepsilon, f)$ along with Markov inequality, we have

$$R_t^2(\varepsilon, f) \leq \mathbb{P}_t^2(S) \leq \Big(\frac{\mathbb{E}_{\boldsymbol{x}_t\sim\mathbb{P}_t}\mathbb{E}_{\mathbf{x}_{t,1},\mathbf{x}_{t,2}\in\mathcal{A}(\boldsymbol{x}_t)}\big\|f(\mathbf{x}_{t,1})-f(\mathbf{x}_{t,2})\big\|_2}{\frac{\varepsilon}{m^2}}\Big)^2 \tag{22}$$

$$\leq \frac{\mathbb{E}_{\boldsymbol{x}_t\sim\mathbb{P}_t}\mathbb{E}_{\mathbf{x}_{t,1},\mathbf{x}_{t,2}\in\mathcal{A}(\boldsymbol{x}_t)}\big\|f(\mathbf{x}_{t,1})-f(\mathbf{x}_{t,2})\big\|_2^2}{\frac{\varepsilon^2}{m^4}}$$

$$= \frac{m^4}{\varepsilon^2}\mathbb{E}_{\boldsymbol{x}_t\sim\mathbb{P}_t}\mathbb{E}_{\mathbf{x}_{t,1},\mathbf{x}_{t,2}\in\mathcal{A}(\boldsymbol{x}_t)}\big\|f(\mathbf{x}_{t,1})-f(\mathbf{x}_{t,2})\big\|_2^2. \tag{23}$$

Apart from that, similar process yields the first inequity to be justified in Lemma 4:

$$R_s^2(\varepsilon, f) \leq \frac{m^4}{\varepsilon^2}\mathbb{E}_{\boldsymbol{x}_s\sim\mathbb{P}_s}\mathbb{E}_{\mathbf{x}_{s,1},\mathbf{x}_{s,2}\in\mathcal{A}(\boldsymbol{x}_s)}\big\|f(\mathbf{x}_{s,1})-f(\mathbf{x}_{s,2})\big\|_2^2 = \frac{m^4}{\varepsilon^2}\mathcal{L}_{\mathrm{align}}(f).$$

Furthermore, we can turn out

$$\mathbb{E}_{\boldsymbol{x}_t\sim\mathbb{P}_t}\mathbb{E}_{\mathbf{x}_{t,1},\mathbf{x}_{t,2}\in\mathcal{A}(\boldsymbol{x}_t)}\big\|f(\mathbf{x}_{t,1})-f(\mathbf{x}_{t,2})\big\|_2^2$$

$$= \mathbb{E}_{\boldsymbol{x}_s}\mathbb{E}_{\mathbf{x}_{s,1},\mathbf{x}_{s,2}\in\mathcal{A}(\boldsymbol{x}_s)}\big\|f(\mathbf{x}_{s,1})-f(\mathbf{x}_{s,2})\big\|_2^2 + \mathbb{E}_{\boldsymbol{x}_t}\mathbb{E}_{\mathbf{x}_{t,1},\mathbf{x}_{t,2}\in\mathcal{A}(\boldsymbol{x}_t)}\big\|f(\mathbf{x}_{t,1})-f(\mathbf{x}_{t,2})\big\|_2^2$$

$$\quad - \mathbb{E}_{\boldsymbol{x}_s}\mathbb{E}_{\mathbf{x}_{s,1},\mathbf{x}_{s,2}\in\mathcal{A}(\boldsymbol{x}_s)}\big\|f(\mathbf{x}_{s,1})-f(\mathbf{x}_{s,2})\big\|_2^2$$

$$= \frac{1}{m^2}\sum_{i,j=1}^{m}\Big\{\mathbb{E}_{\boldsymbol{x}_t\sim\mathbb{P}_t}\big\|f\big(A_i(\boldsymbol{x}_t)\big)-f\big(A_j(\boldsymbol{x}_t)\big)\big\|_2^2 - \mathbb{E}_{\boldsymbol{x}_s\sim\mathbb{P}_s}\big\|f\big(A_i(\boldsymbol{x}_s)\big)-f\big(A_j(\boldsymbol{x}_s)\big)\big\|_2^2\Big\}$$

$$\quad + \mathbb{E}_{\boldsymbol{x}_s\sim\mathbb{P}_s}\mathbb{E}_{\mathbf{x}_{s,1},\mathbf{x}_{s,2}\in\mathcal{A}(\boldsymbol{x}_s)}\big\|f(\mathbf{x}_{s,1})-f(\mathbf{x}_{s,2})\big\|_2^2$$

$$= \frac{1}{m^2}\sum_{i,j=1}^{m}\sum_{l=1}^{d^*}\Big[\mathbb{E}_{\boldsymbol{x}_t\sim\mathbb{P}_t}\big\{f_l\big(A_i(\boldsymbol{x}_t)\big)-f_l\big(A_j(\boldsymbol{x}_t)\big)\big\}^2 - \mathbb{E}_{\boldsymbol{x}_s\sim\mathbb{P}_s}\big\{f_l\big(A_i(\boldsymbol{x}_s)\big)-f_l\big(A_j(\boldsymbol{x}_s)\big)\big\}^2\Big]$$

$$\quad + \mathbb{E}_{\boldsymbol{x}_s\sim\mathbb{P}_s}\mathbb{E}_{\mathbf{x}_{s,1},\mathbf{x}_{s,2}\in\mathcal{A}(\boldsymbol{x}_s)}\big\|f(\mathbf{x}_{s,1})-f(\mathbf{x}_{s,2})\big\|_2^2,$$

we subsequently focus on dealing with the first term. Since for all $\gamma\in[m], \beta\in[m]$ and $l\in[d^*]$,

$$\mathbb{E}_{\boldsymbol{x}_t\sim\mathbb{P}_t}\big\{f_l\big(A_i(\boldsymbol{x}_t)\big)-f_l\big(A_j(\boldsymbol{x}_t)\big)\big\}^2 - \mathbb{E}_{\boldsymbol{x}_s\sim\mathbb{P}_s}\big\{f_l\big(A_i(\boldsymbol{x}_s)\big)-f_l\big(A_j(\boldsymbol{x}_s)\big)\big\}^2$$

$$= \sum_{k=1}^{K}\Big[p_t(k)\mathbb{E}_{\boldsymbol{x}_t\in C_t(k)}\big\{f_l\big(A_i(\boldsymbol{x}_t)\big)-f_l\big(A_j(\boldsymbol{x}_t)\big)\big\}^2 - p_s(k)\mathbb{E}_{\boldsymbol{x}_s\in C_s(k)}\big\{f_l\big(A_i(\boldsymbol{x}_s)\big)-f_l\big(A_j(\boldsymbol{x}_s)\big)\big\}^2\Big]$$

$$= \sum_{k=1}^{K}\Big[p_t(k)\Big\{\mathbb{E}_{\boldsymbol{x}_t\in C_t(k)}\big\{f_l\big(A_i(\boldsymbol{x}_t)\big)-f_l\big(A_j(\boldsymbol{x}_t)\big)\big\}^2 - \mathbb{E}_{\boldsymbol{x}_s\in C_s(k)}\underbrace{\big\{f_l\big(A_i(\boldsymbol{x}_s)\big)-f_l\big(A_j(\boldsymbol{x}_s)\big)\big\}^2}_{g(\boldsymbol{x}_s)}\Big\}$$

$$\quad + \big\{p_t(k)-p_s(k)\big\}\mathbb{E}_{\boldsymbol{x}_s\in C_s(k)}\big\{f_l\big(A_i(\boldsymbol{x}_s)\big)-f_l\big(A_j(\boldsymbol{x}_s)\big)\big\}^2\Big]$$

$$\leq 8B_2M\mathcal{K}\epsilon_1 + 4B_2^2K\epsilon_2.$$

To obtain the last inequality, it suffices to show $g(\boldsymbol{x}_s) \in \mathrm{Lip}(8B_2M\mathcal{K})$. In fact, we know $\forall l \in [d^*], f_l \in \mathrm{Lip}(\mathcal{K})$ as $f \in \mathrm{Lip}(\mathcal{K})$, along with the fact that $A_i(\cdot)$ and $A_j(\cdot)$ are both $M$-Lipschitz

continuous according to Assumption 3, we can conclude $f_l\big(A_i(\cdot)\big) - f_l\big(A_j(\cdot)\big) \in \mathrm{Lip}(2M\mathcal{K})$. Additionally, note that $\big|f_l\big(A_i(\cdot)\big) - f_l\big(A_j(\cdot)\big)\big| \le 2B_2$ as $\|f\|_2 \le B_2$, we can turn out outermost quadratic function remains locally $4B_2$-Lipschitz continuity in $[-2B_2, 2B_2]$, which implies that $g \in \mathrm{Lip}(8B_2M\mathcal{K})$. Furthermore, by the definition of Wasserstein distance, we yield

$$\sum_{k=1}^{K}\Big[p_t(k)\Big(\mathbb{E}_{\boldsymbol{x}_t\in C_t(k)}\{f_l\big(A_i(\boldsymbol{x}_t)\big) - f_l\big(A_j(\boldsymbol{x}_t)\big)\}^2 - \mathbb{E}_{\boldsymbol{x}_s\in C_s(k)}\{f_l\big(A_i(\boldsymbol{x}_s)\big) - f_l\big(A_j(\boldsymbol{x}_s)\big)\}^2\Big)\Big]$$

$$\le 8B_2M\mathcal{K}\epsilon_1 \sum_{k=1}^{K} p_t(k) = 8B_2M\mathcal{K}\epsilon_1,$$

As for the second term in the last inequality, note that $f_l\big(A_i(\boldsymbol{x}_s)\big) - f_l\big(A_j(\boldsymbol{x}_s)\big) \le 2B_2$ to yield

$$\sum_{k=1}^{K}\Big[\{p_t(k) - p_s(k)\}\mathbb{E}_{\boldsymbol{x}_s\in C_s(k)}\{f_l\big(A_i(\boldsymbol{x}_s)\big) - f_l\big(A_j(\boldsymbol{x}_s)\big)\}^2\Big] \le 4B_2^2 K\epsilon_2.$$

Therefore,

$$\mathbb{E}_{\boldsymbol{x}_t\sim\mathbb{P}_t}\mathbb{E}_{\mathbf{x}_{t,1},\mathbf{x}_{t,2}\in\mathcal{A}(\boldsymbol{x}_t)}\big\|f(\mathbf{x}_{t,1}) - f(\mathbf{x}_{t,2})\big\|_2^2 \le \mathbb{E}_{\boldsymbol{x}_s\sim\mathbb{P}_s}\mathbb{E}_{\mathbf{x}_{s,1},\mathbf{x}_{s,2}\in\mathcal{A}(\boldsymbol{x}_s)}\big\|f(\mathbf{x}_{s,1}) - f(\mathbf{x}_{s,2})\big\|_2^2$$
$$+ 8B_2 d^* M\mathcal{K}\epsilon_1 + 4B_2^2 d^* K\epsilon_2. \tag{24}$$

Combining (22) and (24) turns out the second inequality of Lemma 4.

$$R_t^2(\varepsilon, f) \le \frac{m^4}{\varepsilon^2}\mathcal{L}_{\mathrm{align}}(f) + \frac{8m^4}{\varepsilon^2}B_2 d^* M\mathcal{K}\epsilon_1 + \frac{4m^4}{\varepsilon^2}B_2^2 d^* K\epsilon_2.$$

To justify the third part of this Lemma, first recall Lemma 2 that $\forall k \in [K]$, $\big\|\mu_s(k) - \mu_t(k)\big\|_2 \le \sqrt{d^*}M\mathcal{K}\epsilon_1$. Hence, for any $i \ne j$, we have

$$\big|\mu_t(i)^\top\mu_t(j) - \mu_s(i)^\top\mu_s(j)\big| = \big|\mu_t(i)^\top\mu_t(j) - \mu_t(i)^\top\mu_s(j) + \mu_t(i)^\top\mu_s(j) - \mu_s(i)^\top\mu_s(j)\big|$$

$$\le \big\|\mu_t(i)\big\|_2\big\|\mu_t(j) - \mu_s(j)\big\|_2 + \big\|\mu_s(j)\big\|_2\big\|\mu_t(i) - \mu_s(i)\big\|_2 \le 2\sqrt{d^*}B_2M\mathcal{K}\epsilon_1,$$

so that we can further yield the relationship of class center divergence between the source domain and the target domain as follows:

$$\max_{i\ne j}\big|\mu_t(i)^\top\mu_t(j)\big| \le \max_{i\ne j}\big|\mu_s(i)^\top\mu_s(j)\big| + 2\sqrt{d^*}B_2M\mathcal{K}\epsilon_1. \tag{25}$$

We next derive the upper bound of $\max_{i\ne j}\big|\mu_s(i)^\top\mu_s(j)\big|$. To this end, let $U = \big(\sqrt{p_s(1)}\mu_s(1), \ldots, \sqrt{p_s(K)}\mu_s(K)\big) \in \mathbb{R}^{d^*\times K}$, then

$$\Big\|\sum_{k=1}^{K}p_s(k)\mu_s(k)\mu_s(k)^\top - I_{d^*}\Big\|_F^2 = \Big\|UU^\top - I_{d^*}\Big\|_F^2$$

$$= \mathrm{Tr}(UU^\top UU^\top - 2UU^\top + I_{d^*}) \qquad (\|A\|_F^2 = \mathrm{Tr}(A^\top A))$$

$$= \mathrm{Tr}(U^\top UU^\top U - 2U^\top U) + \mathrm{Tr}(I_K) + d^* - K$$
$$(\mathrm{Tr}(AB) = \mathrm{Tr}(BA))$$

$$\ge \Big\|U^\top U - I_K\Big\|_F^2 \qquad (d^* > K)$$

$$= \sum_{i,j=1}^{K}\big(\sqrt{p_s(i)p_s(j)}\mu_s(i)^\top\mu_s(j) - \delta_{kl}\big)^2$$

$$\ge p_s(i)p_s(j)\big(\mu_s(i)^\top\mu_s(j)\big)^2.$$

Therefore,

$$\big(\mu_s(i)^\top\mu_s(j)\big)^2 \le \frac{\Big\|\sum_{k=1}^{K}p_s(k)\mu_s(k)\mu_s(k)^\top - I_{d^*}\Big\|_F^2}{p_s(i)p_s(j)}$$

$$= \frac{\left\| \mathbb{E}_{\boldsymbol{x}_s} \mathbb{E}_{\mathbf{x}_{s,1},\mathbf{x}_{s,2}\in\mathcal{A}(\boldsymbol{x}_s)} \left\{ f(\mathbf{x}_{s,1})f(\mathbf{x}_{s,2})^\top \right\} - I_{d^*} + \sum_{k=1}^{K} p_s(k)\mu_s(k)\mu_s(k)^\top - \mathbb{E}_{\boldsymbol{x}_s} \mathbb{E}_{\mathbf{x}_{s,1},\mathbf{x}_{s,2}\in\mathcal{A}(\boldsymbol{x}_s)} \left\{ f(\mathbf{x}_{s,1})f(\mathbf{x}_{s,2})^\top \right\} \right\|_F^2}{p_s(i)p_s(j)}$$

$$\leq \frac{2\left\| \mathbb{E}_{\boldsymbol{x}_s} \mathbb{E}_{\mathbf{x}_{s,1},\mathbf{x}_{s,2}\in\mathcal{A}(\boldsymbol{x}_s)} \left\{ f(\mathbf{x}_{s,1})f(\mathbf{x}_{s,2})^\top \right\} - I_{d^*} \right\|_F^2 + 2\left\| \sum_{k=1}^{K} p_s(k)\mu_s(k)\mu_s(k)^\top - \mathbb{E}_{\boldsymbol{x}_s} \mathbb{E}_{\mathbf{x}_{s,1},\mathbf{x}_{s,2}\in\mathcal{A}(\boldsymbol{x}_s)} \left\{ f(\mathbf{x}_{s,1})f(\mathbf{x}_{s,2})^\top \right\} \right\|_F^2}{p_s(i)p_s(j)}$$

$$(26)$$

For the term $\left\| \sum_{k=1}^{K} p_s(k)\mu_s(k)\mu_s(k)^\top - \mathbb{E}_{\boldsymbol{x}_s\sim\mathbb{P}_s}\mathbb{E}_{\mathbf{x}_{s,1},\mathbf{x}_{s,2}\in\mathcal{A}(\boldsymbol{x}_s)}\left\{ f(\mathbf{x}_{s,1})f(\mathbf{x}_{s,2})^\top \right\} \right\|_F^2$, note that

$$= \sum_{k=1}^{K} p_s(k)\mathbb{E}_{\boldsymbol{x}_s\in C_s(k)}\mathbb{E}_{\mathbf{x}_{s,1}\in\mathcal{A}(\boldsymbol{x}_s)}\left\{ f(\mathbf{x}_{s,1})f(\mathbf{x}_{s,1})^\top \right\} - \sum_{k=1}^{K} p_s(k)\mu_s(k)\mu_s(k)^\top$$

$$+ \sum_{k=1}^{K} p_s(k)\mathbb{E}_{\boldsymbol{x}_s\in C_s(k)}\mathbb{E}_{\mathbf{x}_{s,1},\mathbf{x}_{s,2}\in\mathcal{A}(\boldsymbol{x}_s)}\left[ f(\mathbf{x}_{s,1})\{f(\mathbf{x}_{s,2})-f(\mathbf{x}_{s,1})\}^\top \right]$$

$$= \sum_{k=1}^{K} p_s(k)\mathbb{E}_{\boldsymbol{x}_s\in C_s(k)}\mathbb{E}_{\mathbf{x}_{s,1}\in\mathcal{A}(\boldsymbol{x}_s)}\left[ \{f(\mathbf{x}_{s,1})-\mu_s(k)\}\{f(\mathbf{x}_{s,1})-\mu_s(k)\}^\top \right] \qquad (27)$$

$$+ \mathbb{E}_{\boldsymbol{x}_s\sim\mathbb{P}_s}\mathbb{E}_{\mathbf{x}_{s,1},\mathbf{x}_{s,2}\in\mathcal{A}(\boldsymbol{x}_s)}\left[ f(\mathbf{x}_{s,1})\{f(\mathbf{x}_{s,2})-f(\mathbf{x}_{s,1})\}^\top \right], \qquad (28)$$

where the last equation is derived from

$$\mathbb{E}_{\boldsymbol{x}_s\in C_s(k)}\mathbb{E}_{\mathbf{x}_{s,1}\in\mathcal{A}(\boldsymbol{x}_s)}\left\{ f(\mathbf{x}_{s,1})f(\mathbf{x}_{s,1})^\top \right\} - \mu_s(k)\mu_s(k)^\top$$

$$= \mathbb{E}_{\boldsymbol{x}_s\in C_s(k)}\mathbb{E}_{\mathbf{x}_{s,1}\in\mathcal{A}(\boldsymbol{x}_s)}\left\{ f(\mathbf{x}_{s,1})f(\mathbf{x}_{s,1})^\top \right\} + \mu_s(k)\mu_s(k)^\top$$

$$- \left( \mathbb{E}_{\boldsymbol{x}_s\in C_s(k)}\mathbb{E}_{\mathbf{x}_{s,1}\in\mathcal{A}(\boldsymbol{x}_s)}\left\{ f(\mathbf{x}_{s,1}) \right\} \right)\mu_s(k)^\top - \mu_s(k)\left( \mathbb{E}_{\boldsymbol{x}_s\in C_s(k)}\mathbb{E}_{\mathbf{x}_{s,1}\in\mathcal{A}(\boldsymbol{x}_s)}\left\{ f(\mathbf{x}_{s,1}) \right\} \right)^\top$$

$$= \mathbb{E}_{\boldsymbol{x}_s\in C_s(k)}\mathbb{E}_{\mathbf{x}_{s,1}\in\mathcal{A}(\boldsymbol{x}_s)}\left[ \left( f(\mathbf{x}_{s,1})-\mu_s(k) \right)\left( f(\mathbf{x}_{s,1})-\mu_s(k) \right)^\top \right].$$

So its norm is

$$\left\| \sum_{k=1}^{K} p_s(k)\mu_s(k)\mu_s(k)^\top - \mathbb{E}_{\boldsymbol{x}_s\sim\mathbb{P}_s}\mathbb{E}_{\mathbf{x}_{s,1},\mathbf{x}_{s,2}\in\mathcal{A}(\boldsymbol{x}_s)}\left\{ f(\mathbf{x}_{s,1})f(\mathbf{x}_{s,2})^\top \right\} \right\|_F$$

$$\leq \sum_{k=1}^{K} p_s(k)\mathbb{E}_{\boldsymbol{x}_s\in C_s(k)}\mathbb{E}_{\mathbf{x}_{s,1}\in\mathcal{A}(\boldsymbol{x}_s)}\left[ \left\| \{f(\mathbf{x}_{s,1})-\mu_s(k)\}\{f(\mathbf{x}_{s,1})-\mu_s(k)\}^\top \right\|_F \right]$$

$$+ \mathbb{E}_{\boldsymbol{x}_s\sim\mathbb{P}_s}\mathbb{E}_{\mathbf{x}_{s,1},\mathbf{x}_{s,2}\in\mathcal{A}(\boldsymbol{x}_s)}\left[ \left\| f(\mathbf{x}_{s,1})\{f(\mathbf{x}_{s,2})-f(\mathbf{x}_{s,1})\}^\top \right\|_F \right]$$

$$\leq \sum_{k=1}^{K} p_s(k)\mathbb{E}_{\boldsymbol{x}_s\in C_s(k)}\mathbb{E}_{\mathbf{x}_{s,1}\in\mathcal{A}(\boldsymbol{x}_s)}\left\{ \left\| f(\mathbf{x}_{s,1})-\mu_s(k) \right\|_2^2 \right\} + \mathbb{E}_{\boldsymbol{x}_s}\mathbb{E}_{\mathbf{x}_{s,1},\mathbf{x}_{s,2}\in\mathcal{A}(\boldsymbol{x}_s)}\left\{ \left\| f(\mathbf{x}_{s,1}) \right\|_2 \left\| f(\mathbf{x}_{s,2})-f(\mathbf{x}_{s,1}) \right\|_2 \right\}$$

$$\leq \sum_{k=1}^{K} p_s(k)\mathbb{E}_{\boldsymbol{x}_s\in C_s(k)}\mathbb{E}_{\mathbf{x}_{s,1}\in\mathcal{A}(\boldsymbol{x}_s)}\left\{ \left\| f(\mathbf{x}_{s,1})-\mu_s(k) \right\|_2^2 \right\}$$

$$+ \left\{ \mathbb{E}_{\boldsymbol{x}_s\sim\mathbb{P}_s}\mathbb{E}_{\mathbf{x}_{s,1}\in\mathcal{A}(\boldsymbol{x}_s)}\left\| f(\mathbf{x}_{s,1}) \right\|_2^2 \right\}^{\frac{1}{2}}\left\{ \mathbb{E}_{\boldsymbol{x}_s\sim\mathbb{P}_s}\mathbb{E}_{\mathbf{x}_{s,1},\mathbf{x}_{s,2}\in\mathcal{A}(\boldsymbol{x}_s)}\left\| f(\mathbf{x}_{s,2})-f(\mathbf{x}_{s,1}) \right\|_2^2 \right\}^{\frac{1}{2}}$$

$$\leq \sum_{k=1}^{K} p_s(k)\mathbb{E}_{\boldsymbol{x}_s\in C_s(k)}\mathbb{E}_{\mathbf{x}_{s,1}\in\mathcal{A}(\boldsymbol{x}_s)}\left[ \left\| f(\mathbf{x}_{s,1})-\mu_s(k) \right\|_2^2 \right]$$

$$+ B_2\left( \varepsilon^2 + \mathbb{E}_{\boldsymbol{x}_s\sim\mathbb{P}_s}\mathbb{E}_{\mathbf{x}_{s,1},\mathbf{x}_{s,2}\in\mathcal{A}(\boldsymbol{x}_s)}\left[ \left\| f(\mathbf{x}_{s,2})-f(\mathbf{x}_{s,1}) \right\|_2^2 \mathbb{1}\left\{ \boldsymbol{x}_s\notin S_s(\varepsilon,f) \right\} \right] \right)^{\frac{1}{2}}$$

$$\left( \text{Review } S_s(\varepsilon,f) := \left\{ \boldsymbol{x}_s\in\mathcal{X}_s : \sup_{\mathbf{x}_{s,1},\mathbf{x}_{s,2}\in\mathcal{A}(\boldsymbol{x}_s)} \left\| f(\mathbf{x}_{s,1})-f(\mathbf{x}_{s,2}) \right\|_2 \leq \varepsilon \right\} \right)$$

$$\leq \sum_{k=1}^{K} p_s(k)\mathbb{E}_{\boldsymbol{x}_s\in C_s(k)}\mathbb{E}_{\mathbf{x}_{s,1}\in\mathcal{A}(\boldsymbol{x}_s)}\left\{ \left\| f(\mathbf{x}_{s,1})-\mu_s(k) \right\|_2^2 \right\}$$

$$+ B_2\Big(\varepsilon^2 + 4B_2^2\mathbb{E}_{\boldsymbol{x}_s\sim\mathbb{P}_s}\big[\mathbb{1}\{\boldsymbol{x}_s\notin S_s(\varepsilon,f)\}\big]\Big)^{\frac{1}{2}}$$

$$= \sum_{k=1}^{K} p_s(k)\mathbb{E}_{\boldsymbol{x}_s\in C_s(k)}\mathbb{E}_{\mathbf{x}_{s,1}\in\mathcal{A}(\boldsymbol{x}_s)}\Big[\big\|f(\mathbf{x}_{s,1}) - \mu_s(k)\big\|_2^2\Big] + B_2\Big(\varepsilon^2 + 4B_2^2 R_s(\varepsilon,f)\Big)^{\frac{1}{2}}$$

$$\leq 4B_2^2\sum_{k=1}^{K} p_s(k)\Big\{\Big(1-\sigma_s + \frac{\mathcal{K}\delta_s}{2B_2} + \frac{\varepsilon}{B_2} + \frac{R_s(\varepsilon,f)}{p_s(k)}\Big)^2 + \Big(1-\sigma_s + \frac{R_s(\varepsilon,f)}{p_s(k)}\Big)\Big\} + B_2\big\{\varepsilon^2 + 4B_2^2 R_s(\varepsilon,f)\big\}^{\frac{1}{2}}$$

(Lemma 3)

$$= 4B_2^2\Big\{\Big(1-\sigma_s + \frac{\mathcal{K}\delta_s + 2\varepsilon}{2B_2}\Big)^2 + (1-\sigma_s) + KR_s(\varepsilon,f)\Big(3 - 2\sigma_s + \frac{\mathcal{K}\delta_s + 2\varepsilon}{B_2}\Big)$$

$$+ R_s^2(\varepsilon,f)\Big(\sum_{k=1}^{K}\frac{1}{p_s(k)}\Big)\Big\} + B_2\Big\{\varepsilon^2 + 4B_2^2 R_s(\varepsilon,f)\Big\}^{\frac{1}{2}}$$

If we define $\varphi(\sigma_s,\delta_s,\varepsilon,f) := 4B_2^2\Big\{\Big(1-\sigma_s + \frac{\mathcal{K}\delta_s+2\varepsilon}{2B_2}\Big)^2 + (1-\sigma_s) + KR_s(\varepsilon,f)\Big(3 - 2\sigma_s + \frac{\mathcal{K}\delta_s+2\varepsilon}{B_2}\Big) + R_s^2(\varepsilon,f)\Big(\sum_{k=1}^{K}\frac{1}{p_s(k)}\Big)\Big\} + B_2\Big(\varepsilon^2 + 4B_2^2 R_s(\varepsilon,f)\Big)^{\frac{1}{2}}$, above derivation implies

$$\Big\|\sum_{k=1}^{K} p_s(k)\mu_s(k)\mu_s(k)^\top - \mathbb{E}_{\boldsymbol{x}_s\sim\mathbb{P}_s}\mathbb{E}_{\mathbf{x}_{s,1},\mathbf{x}_{s,2}\in\mathcal{A}(\boldsymbol{x}_s)}\big\{f(\mathbf{x}_{s,1})f(\mathbf{x}_{s,2})^\top\big\}\Big\|_F \leq \varphi(\sigma_s,\delta_s,\varepsilon,f). \quad (29)$$

Besides that, Note that

$$\mathcal{R} = \Big\|\mathbb{E}_{\boldsymbol{x}_s\sim\mathbb{P}_s}\mathbb{E}_{\mathbf{x}_{s,1},\mathbf{x}_{s,2}\in\mathcal{A}(\boldsymbol{x}_s)}\big\{f(\mathbf{x}_{s,1})f(\mathbf{x}_{s,2})^\top\big\} - I_{d^*}\Big\|_F^2, \quad (30)$$

Combining (26), (27), (29) and (30) yields for any $i\neq j$

$$\big(\mu_s(i)^\top\mu_s(j)\big)^2 \leq \frac{2}{p_s(i)p_s(j)}\Big\{\mathcal{R}(f) + \varphi(\sigma_s,\delta_s,\varepsilon,f)\Big\},$$

which implies that

$$\max_{i\neq j}\big|\mu_s(i)^\top\mu_s(j)\big| \leq \sqrt{\frac{2}{\min_{i\neq j} p_s(i)p_s(j)}\Big\{\mathcal{R}(f) + \varphi(\sigma_s,\delta_s,\varepsilon,f)\Big\}}.$$

So we can get what we desired according to (25)

$$\max_{i\neq j}\big|\mu_t(i)^\top\mu_t(j)\big| \leq \sqrt{\frac{2}{\min_{i\neq j} p_s(i)p_s(j)}\Big\{\mathcal{R}(f) + \varphi(\sigma_s,\delta_s,\varepsilon,f)\Big\}} + 2\sqrt{d^*}B_2 M\mathcal{K}\epsilon_1.$$

□

Next we present the population theorem as follows, which is a direct corollary of Lemma 4 because of the facts that $\mathcal{R}(f)\lesssim\mathcal{L}(f)$ and $\mathcal{L}_{\mathrm{align}}(f)\lesssim\mathcal{L}(f)$.

**Lemma 5.** *Given a $(\sigma_s,\sigma_t,\delta_s,\delta_t)$-augmentation, if $d^* > K$, Assumption 3 holds and the encoder $f$ with $B_1 \leq \|f\|_2 \leq B_2$ is $\mathcal{K}$-Lipschitz continuous, then for any $\varepsilon > 0$,*

$$\max_{i\neq j}\big|\mu_t(i)^\top\mu_t(j)\big| \lesssim \sqrt{\mathcal{L}(f) + \varphi(\sigma_s,\delta_s,\varepsilon,f)} + \mathcal{K}\epsilon_1.$$

*Furthermore, if $\max_{i\neq j}\mu_t(i)^\top\mu_t(j) < B_2^2\psi(\sigma_t,\delta_t,\varepsilon,f)$, then the misclassification rate of $Q_f$*

$$\mathrm{Err}(Q_f) \lesssim (1-\sigma_t) + \big\{\mathcal{L}_{\mathrm{align}}(f) + \mathcal{K}\epsilon_1 + \epsilon_2\big\}/\varepsilon^2,$$

*where the specific formulations of $\varphi(\sigma_s,\delta_s,\varepsilon,f)$ and $\psi(\sigma_t,\delta_t,\varepsilon,f)$ can be found in Lemma 4 and Lemma 1, respectively.*

We apply Lemma 5 to the optimizer at sample level $\hat{f}_{n_s}$ to yield following corollary 1.

**Corollary 1.** *Given a $(\sigma_s, \sigma_t, \delta_s, \delta_t)$-augmentation, for any $\varepsilon > 0$, we have*

$$\mathbb{E}_{\widetilde{D}_s}\big\{ \max_{i \neq j} \big| \mu_t(i)^\top \mu_t(j) \big| \big\} \lesssim \sqrt{\mathbb{E}_{\widetilde{D}_s}\{\mathcal{L}(\hat{f}_{n_s})\} + \mathbb{E}_{\widetilde{D}_s}\{\varphi(\sigma_s, \delta_s, \varepsilon, \hat{f}_{n_s})\}} + \mathcal{K}\epsilon_1. \tag{31}$$

*where $\mathbb{E}_{\widetilde{D}_s}\big\{\varphi\big(\sigma_s, \delta_s, \varepsilon, R_s(\varepsilon, \hat{f}_{n_s})\big)\big\} \lesssim \big(1 - \sigma_s + \mathcal{K}\delta_s + 2\varepsilon\big)^2 + \frac{1}{\varepsilon}\sqrt{\mathbb{E}_{\widetilde{D}_s}\{\mathcal{L}(\hat{f}_{n_s})\}}\big(3 - 2\sigma_s + \mathcal{K}\delta_s + 2\varepsilon\big) + \frac{1}{\varepsilon^2}\mathbb{E}_{\widetilde{D}_s}\{\mathcal{L}(\hat{f}_{n_s})\} + (1 - \sigma_s) + \big(\varepsilon^2 + \frac{1}{\varepsilon}\sqrt{\mathbb{E}_{\widetilde{D}_s}\{\mathcal{L}(\hat{f}_{n_s})\}}\big)^{\frac{1}{2}}$. Furthermore, if $\max_{i \neq j}\big|\mu_t(i)^\top \mu_t(j)\big| < B_2^2 \psi(\sigma_t, \delta_t, \varepsilon, f)$, then*

$$\mathbb{E}_{\widetilde{D}_s}\big\{\mathrm{Err}(Q_{\hat{f}_{n_s}})\big\} \lesssim (1 - \sigma_t) + \frac{1}{\varepsilon}\sqrt{\mathbb{E}_{\widetilde{D}_s}\{\mathcal{L}(\hat{f}_{n_s})\} + \mathcal{K}\epsilon_1 + \epsilon_2}, \tag{32}$$

*In addition, the following inequalities always hold*

$$\mathbb{E}_{\widetilde{D}_s}\big\{R_s(\varepsilon, \hat{f}_{n_s})\big\} \lesssim \frac{1}{\varepsilon}\sqrt{\mathbb{E}_{\widetilde{D}_s}\{\mathcal{L}(\hat{f}_{n_s})\}} \tag{33}$$

$$\mathbb{E}_{\widetilde{D}_s}\big\{R_t(\varepsilon, \hat{f}_{n_s})\big\} \lesssim \frac{1}{\varepsilon}\sqrt{\mathbb{E}_{\widetilde{D}_s}\{\mathcal{L}(\hat{f}_{n_s})\} + \mathcal{K}\epsilon_1 + \epsilon_2}. \tag{34}$$

*Proof.* Applying Lemma 4 to $\hat{f}_{n_s}$ yields

$$R_s^2(\varepsilon, \hat{f}_{n_s}) \leq \frac{m^4}{\varepsilon^2}\mathcal{L}(\hat{f}_{n_s}) \tag{35}$$

$$R_t^2(\varepsilon, \hat{f}_{n_s}) \leq \frac{m^4}{\varepsilon^2}\mathcal{L}(\hat{f}_{n_s}) + \frac{8m^4}{\varepsilon^2}B_2 d^* M\mathcal{K}\epsilon_1 + \frac{4m^4}{\varepsilon^2}B_2^2 d^* K\epsilon_2 \tag{36}$$

and

$$\max_{i \neq j}\big|\mu_t(i)^\top \mu_t(j)\big| \leq \sqrt{\frac{2}{\min_{i \neq j} p_s(i)p_s(j)}\Big(\frac{1}{\lambda}\mathcal{L}(\hat{f}_{n_s}) + \varphi(\sigma_s, \delta_s, \varepsilon, \hat{f}_{n_s})\Big)} + 2\sqrt{d^*}B_2 M\mathcal{K}\epsilon_1 \tag{37}$$

Take the expectation with respect to $\widetilde{D}_s$ on both sides of (35), (36), and (37), using Jensen's inequality to obtain (31), (33) and (34). where $\mathbb{E}_{\widetilde{D}_s}\{\varphi(\sigma_s, \delta_s, \varepsilon, \hat{f}_{n_s})\} = 4B_2^2\Big[\big(1 - \sigma_s + \frac{\mathcal{K}\delta_s + 2\varepsilon}{2B_2}\big)^2 + (1 - \sigma_s) + K\mathbb{E}_{\widetilde{D}_s}\{R_s(\varepsilon, \hat{f}_{n_s})\}\big(3 - 2\sigma_s + \frac{\mathcal{K}\delta_s + 2\varepsilon}{B_2}\big) + \mathbb{E}_{\widetilde{D}_s}\{R_s^2(\varepsilon, \hat{f}_{n_s})\}\big(\sum_{k=1}^{K}\frac{1}{p_s(k)}\big)\Big] + B_2\mathbb{E}_{\widetilde{D}_s}\Big[\big\{\varepsilon^2 + 4B_2^2 R_s(\varepsilon, \hat{f}_{n_s})\big\}^{\frac{1}{2}}\Big]$. In this regard, further by Jensen inequality, we know that

$$\begin{aligned}
\mathbb{E}_{\widetilde{D}_s}\big\{\varphi(\sigma_s, \delta_s, \varepsilon, R_s(\varepsilon, \hat{f}_{n_s}))\big\} &\leq 4B_2^2\Big[\big(1 - \sigma_s + \frac{\mathcal{K}\delta_s + 2\varepsilon}{2B_2}\big)^2 + (1 - \sigma_s) + K\mathbb{E}_{\widetilde{D}_s}\{R_s(\varepsilon, \hat{f}_{n_s})\} \\
&\quad \big(3 - 2\sigma_s + \frac{\mathcal{K}\delta_s + 2\varepsilon}{B_2}\big) + \mathbb{E}_{\widetilde{D}_s}\{R_s^2(\varepsilon, \hat{f}_{n_s})\}\big(\sum_{k=1}^{K}\frac{1}{p_s(k)}\big)\Big] + B_2\Big[\varepsilon^2 + 4B_2^2\mathbb{E}_{\widetilde{D}_s}\{R_s(\varepsilon, \hat{f}_{n_s})\}\Big]^{\frac{1}{2}} \\
&\leq 4B_2^2\Big[\big(1 - \sigma_s + \frac{\mathcal{K}\delta_s + 2\varepsilon}{2B_2}\big)^2 + \frac{Km^2}{\varepsilon}\sqrt{\mathbb{E}_{\widetilde{D}_s}\{\mathcal{L}(\hat{f}_{n_s})\}}\big(3 - 2\sigma_s + \frac{\mathcal{K}\delta_s + 2\varepsilon}{B_2}\big) \\
&\quad + \frac{m^4}{\varepsilon^2}\mathbb{E}_{\widetilde{D}_s}\{\mathcal{L}(\hat{f}_{n_s})\}\big(\sum_{k=1}^{K}\frac{1}{p_s(k)}\big)\Big] + (1 - \sigma_s) + B_2\big(\varepsilon^2 + \frac{4B_2^2 m^2}{\varepsilon}\sqrt{\mathbb{E}_{\widetilde{D}_s}\{\mathcal{L}(\hat{f}_{n_s})\}}\big)^{\frac{1}{2}}.
\end{aligned} \tag{38}$$

which is same as what we desired.

Moreover, since Lemma 1 reveals that if $\max_{i \neq j}\big|\mu_t(i)^\top \mu_t(j)\big| < B_2^2\psi(\sigma_t, \delta_t, \varepsilon, \hat{f}_{n_s})$, then $\mathrm{Err}(Q_{\hat{f}_{n_s}}) \leq (1 - \sigma_t) + R_t(\varepsilon, \hat{f}_{n_s})$. Combining with what we have had to yield 32, which completes the proof. $\qquad\square$

Above corollary 1 reveals the necessity of exploring the sample complexity of $\mathbb{E}_{\widetilde{D}_s}\{\mathcal{L}(\hat{f}_{n_s})\}$ for proving Theorem 1. To this end, we need to introduce some basic concepts of learning theory in advance.

## F.4 The sample complexity of $\mathbb{E}_{\widetilde{D}_s}\big\{\mathcal{L}(\hat{f}_{n_s})\big\}$

Recall that for given $\boldsymbol{x}_s \in \mathcal{X}_s$, $\mathrm{x}_{s,1}, \mathrm{x}_{s,2}$ is uniformly and independently sampled from $A(\boldsymbol{x}_s)$, we denote $\tilde{\mathrm{x}}_s = (\mathrm{x}_{s,1}, \mathrm{x}_{s,2}) \in \mathbb{R}^{2d^*}$. Moreover, we denote $\ell(\tilde{\mathrm{x}}_s, G) := \big\| f(\mathrm{x}_{s,1}) - f(\mathrm{x}_{s,2}) \big\|_2^2 + \lambda \big\langle f(\mathrm{x}_{s,1}) f(\mathrm{x}_{s,2})^\top - I_{d^*}, G \big\rangle_F$. In this context, our risk at the sample level can be rewritten as

$$\widehat{\mathcal{L}}(f, G) := \frac{1}{n_s} \sum_{i=1}^{n_s} \Big\{ \big\| f(\mathrm{x}_{s,1}^{(i)}) - f(\mathrm{x}_{s,2}^{(i)}) \big\|_2^2 + \lambda \big\langle f(\mathrm{x}_{s,1}^{(i)}) f(\mathrm{x}_{s,2}^{(i)})^\top - I_{d^*}, G \big\rangle_F \Big\} = \frac{1}{n_s} \sum_{i=1}^{n_s} \ell(\tilde{\mathrm{x}}_s^{(i)}, G).$$

Furthermore, let $\mathcal{G}_1 := \big\{ G \in \mathbb{R}^{d^* \times d^*} : \|G\|_F \le B_2^2 + \sqrt{d^*} \big\}$. It is obvious that $\mathcal{G}(f)$ is a subset of $\mathcal{G}_1$ for a given $f$ such that $\|f\|_2 \le B_2$. Likewise, $\widehat{\mathcal{G}}(f)$ is a subset as well for a given $f \in \mathcal{NN}_{d,d^*}(W, L, \mathcal{K}, B_1, B_2)$. On the other hand, following Proposition 2 reveals that $\ell(\boldsymbol{x}, G)$ is a Lipschitz function defined on $\big\{ \boldsymbol{x} \in \mathbb{R}^{2d^*} : \|\boldsymbol{x}\|_2 \le \sqrt{2}B_2 \big\} \times \mathcal{G}_1 \subseteq \mathbb{R}^{2d^* + (d^*)^2}$. Consider these two facts together, we can regard $\ell$ as a Lipschitz function in subsequent context. More specifically, we summary the Lipschitz constants of $\ell(\boldsymbol{x}, G)$ with respect to $\boldsymbol{x} \in \big\{ \boldsymbol{x} \in \mathbb{R}^{2d^*} : \|\boldsymbol{x}\|_2 \le \sqrt{2}B_2 \big\}$ and $G \in \mathcal{G}_1$ in Table 9, the corresponding calculating process is deferred to Proposition 2 for clarity of structure.

| Function | Lipschitz constant |
|---|---|
| $\ell(\boldsymbol{x}, \cdot)$ | $\sqrt{2}B_2$ |
| $\ell(\cdot, G)$ | $2\sqrt{2}B_2(B_2^2 + \sqrt{d^*})$ |
| $\ell(\cdot)$ | $\max\big\{ \sqrt{2}B_2, 2\sqrt{2}B_2(B_2^2 + \sqrt{d^*}) \big\}$ |

Table 9: Lipschitz constant of $\ell$ with respect to each component

Following Definition 4, 5 and Lemma 6, 7 are all typical elements of learning theory, which will be involved by our further derivation.

**Definition 4** (Rademacher complexity). Given a set $S \subseteq \mathbb{R}^n$, the Rademacher complexity of $S$ is defined as

$$\mathcal{R}_n(S) := \mathbb{E}_\xi \Big\{ \sup_{(s_1, \ldots, s_n) \in S} \frac{1}{n} \sum_{i=1}^n \xi_i s_i \Big\},$$

where $\{\xi_i\}_{i \in [n]}$ is a sequence of i.i.d Radmacher random variables which take the values $1$ and $-1$ with equal probability $1/2$.

Moreover, if we use $\ell_2$ to denote the Hilbert space of square summable sequences of real numbers, we have following vector-contraction principle.

**Lemma 6** (Vector-contraction principle). *Let $\mathcal{X}$ be any set, $(x_1, \ldots, x_n) \in \mathcal{X}^n$, let $F$ be a class of functions $f : \mathcal{X} \to \ell_2$ and let $h_i : \ell_2 \to \mathbb{R}$ have Lipschitz norm $L$. Then*

$$\mathbb{E} \sup_{f \in F} \Big| \sum_i \epsilon_i h_i(f(x_i)) \Big| \le 2\sqrt{2}L \mathbb{E} \sup_{f \in F} \Big| \sum_{i,j} \varepsilon_{ij} f_j(x_i) \Big|,$$

*where $\epsilon_{ij}$ is an independent doubly indexed Rademacher sequence and $f_j(x_i)$ is the $j$-th component of $f(x_i)$.*

*Proof.* Combining [29] and Theorem 3.2.1 of [17] obtains the desired result. □

**Definition 5** (Covering number). Given $n \in \mathbb{N}^+$, $\mathcal{S} \subseteq \mathbb{R}^n$ and $\varrho > 0$, the set $\mathcal{N}$ is referred to as an $\varrho$-net of $\mathcal{S}$ with respect to a norm $\|\cdot\|$ on $\mathbb{R}^n$, if $\mathcal{N} \subseteq \mathcal{S}$ and for any $\boldsymbol{x} \in \mathcal{S}$, there exists $\boldsymbol{v} \in \mathcal{N}$ such that $\|\boldsymbol{x} - \boldsymbol{v}\| \le \varrho$. Furthermore, the covering number of $\mathcal{S}$ is defined as

$$\mathcal{N}(\mathcal{S}, \|\cdot\|, \varrho) := \min\big\{ |\mathcal{Q}| : \mathcal{Q} \text{ is an } \varrho\text{-cover of } \mathcal{S} \big\}$$

where $|\mathcal{Q}|$ represents the cardinality of the set $\mathcal{Q}$.

In this context, denote $\mathcal{B}_2$ as the unit ball in $\mathbb{R}^n$. According to the Corollary 4.2.13 of [34], $|\mathcal{N}(\mathcal{B}_2, \|\cdot\|_2, \varrho)|$, which represents the the covering number of $\mathcal{B}_2$ regarding 2-norm, can be bounded by $(3/\varrho)^n$. Based on this fact, if we denote $\mathcal{N}_{\mathcal{G}_1}(\varrho)$ is a cover of $\mathcal{G}_1$ with radius $\varrho$, whose cardinality $|\mathcal{N}_{\mathcal{G}_1}(\varrho)|$ is identical with the covering number of $\mathcal{G}_1$, then $|\mathcal{N}_{\mathcal{G}_1}(\varrho)| \leq \left\{\frac{3}{(B_2^2 + \sqrt{d^*})\varrho}\right\}^{(d^*)^2}$.

**Lemma 7** (Finite maximum inequality). *For any $N \geq 1$, if $X_i, i \leq N$, are sub-Gaussian random variables admitting constants $\sigma_i$, then*

$$\mathbb{E}\left\{\max_{i \in [N]} |X_i|\right\} \leq \sqrt{2 \log 2N} \max_{i \leq N} \sigma_i$$

The proof of this lemma can be found in [17], Lemma 2.3.4.

Recall $\mathcal{NN}_{d_1,d_2}(W, L, \mathcal{K}) := \{f_\theta(\boldsymbol{x}_s) = A_L \sigma(A_{L-1}\sigma(\cdots \sigma(A_0 \boldsymbol{x}_s))) : \kappa(\theta) \leq \mathcal{K}\}$, as defined in eq 48. The second lemma we will employ is related to the upper bound for the Rademacher complexity of the hypothesis space consisting of norm-constrained neural networks, which was provided by [18].

**Lemma 8** (Theorem 3.2 of [18]). *Given $n \in \mathbb{N}^+$, and $\mathbf{x}_{s,1}, \ldots, \boldsymbol{x}_n \in [-B, B]^d$ with $B \geq 1$, define $S := \{(f(\boldsymbol{x}_1), \ldots, f(\boldsymbol{x}_n)) : f \in \mathcal{NN}_{d,1}(W, L, \mathcal{K})\} \subseteq \mathbb{R}^n$, then*

$$\mathcal{R}_n(S) \leq \frac{1}{n} \mathcal{K} \sqrt{2(L + 2 + \log(d+1))} \max_{1 \leq j \leq d+1} \sqrt{\sum_{i=1}^n x_{i,j}^2} \leq \frac{B\mathcal{K}\sqrt{2(L + 2 + \log(d+1))}}{\sqrt{n}},$$

*where $x_{i,j}$ is the $j$-th coordinate of the vector $(\boldsymbol{x}_i^\top, 1)^\top \in \mathbb{R}^{d+1}$, the definition of norm-constraint networks $\mathcal{NN}_{d_1,d_2}(W, L, \mathcal{K})$ is given by*

$$\mathcal{NN}_{d_1,d_2}(W, L, \mathcal{K}) := \{f_{\boldsymbol{\theta}}(\boldsymbol{x}_s) = A_L \sigma(A_{L-1}\sigma(\cdots \sigma(A_0 \boldsymbol{x}_s))) : \kappa(\boldsymbol{\theta}) \leq \mathcal{K}\},$$

*herein, review $\kappa(\boldsymbol{\theta}) = \|A_L\|_\infty \prod_{l=0}^{L-1} \max\{\|(A_l, \boldsymbol{b}_l)\|_\infty, 1\}$.*

### F.4.1 Risk decomposition

Let $\widehat{G}(f) = \frac{1}{n_s} \sum_{i=1}^{n_s} f(\mathbf{x}_{s,1}^{(i)}) f(\mathbf{x}_{s,2}^{(i)})^\top - I_{d^*}$, $G^*(f) = \mathbb{E}_{\boldsymbol{x}_s \sim \mathbb{P}_s} \mathbb{E}_{\mathbf{x}_{s,1},\mathbf{x}_{s,2} \in \mathcal{A}(\boldsymbol{x}_s)}\{f(\mathbf{x}_{s,1}) f(\mathbf{x}_{s,2})^\top\} - I_{d^*}$, we can decompose $\mathcal{E}(\hat{f}_{n_s})$ into three terms shown as follow and then deal each term successively. To achieve conciseness in subsequent conclusions, recall if $X$ and $Y$ are two quantities, we employ $X \lesssim Y$ or $Y \gtrsim X$ to indicate the statement that $X \leq CY$ form some $C > 0$. In addition, We denote $X \asymp Y$ when $X \lesssim Y \lesssim X$.

**Lemma 9.** *The $\mathbb{E}_{\widetilde{D}_s}\{\mathcal{L}(\hat{f}_{n_s})\}$ has following decomposition*

$$\mathbb{E}_{\widetilde{D}_s}\{\mathcal{L}(\hat{f}_{n_s})\} \lesssim \mathcal{L}(f^*) + \mathcal{E}_{\mathrm{sta}} + \mathcal{E}_{\mathcal{F}} + \mathcal{E}_{\mathcal{G}}.$$

*where $\mathcal{E}_{\mathrm{sta}} := \mathbb{E}_{\widetilde{D}_s}\{\sup_{f \in \mathcal{F}, G \in \widehat{\mathcal{G}}(f)} |\mathcal{L}(f, G) - \widehat{\mathcal{L}}(f, G)|\}$ is referred to the statistical error, $\mathcal{E}_{\mathcal{F}} := \inf_{f \in \mathcal{F}}\{\mathcal{L}(f) - \mathcal{L}(f^*)\}$ is called as the approximation error regarding $\mathcal{F}$, while $\mathcal{E}_{\mathcal{G}} := \mathbb{E}_{\widetilde{D}_s}\left[\sup_{f \in \mathcal{F}}\{G^*(f) - \widehat{G}(f)\}\right]$ is named as the error regarding $\mathcal{G}$.*

*Proof.* Notice that $\mathcal{L}(f) = \sup_{G \in \mathcal{G}(f)} \mathcal{L}(f, G)$ holds for both $\hat{f}_{n_s}$ and $f^*$, then for any $f \in \mathcal{F}$ we have

$$\mathcal{L}(\hat{f}_{n_s}) = \mathcal{L}(f^*) + \mathcal{L}(\hat{f}_{n_s}) - \mathcal{L}(f^*) = \mathcal{L}(f^*) + \sup_{G \in \mathcal{G}(\hat{f}_{n_s})} \mathcal{L}(\hat{f}_{n_s}, G) - \sup_{G \in \mathcal{G}(f^*)} \mathcal{L}(f^*, G)$$

$$= \mathcal{L}(f^*) + \left\{\sup_{G \in \mathcal{G}(\hat{f}_{n_s})} \mathcal{L}(\hat{f}_{n_s}, G) - \sup_{G \in \widehat{\mathcal{G}}(\hat{f}_{n_s})} \mathcal{L}(\hat{f}_{n_s}, G)\right\} + \left\{\sup_{G \in \widehat{\mathcal{G}}(\hat{f}_{n_s})} \mathcal{L}(\hat{f}_{n_s}, G) - \sup_{G \in \widehat{\mathcal{G}}(\hat{f}_{n_s})} \widehat{\mathcal{L}}(\hat{f}_{n_s}, G)\right\}$$

$$+ \left\{\sup_{G \in \widehat{\mathcal{G}}(\hat{f}_{n_s})} \widehat{\mathcal{L}}(\hat{f}_{n_s}, G) - \sup_{G \in \widehat{\mathcal{G}}(f)} \widehat{\mathcal{L}}(f, G)\right\} + \left\{\sup_{G \in \widehat{\mathcal{G}}(f)} \widehat{\mathcal{L}}(f, G) - \sup_{G \in \widehat{\mathcal{G}}(f)} \mathcal{L}(f, G)\right\}$$

$$+ \left\{ \sup_{G \in \widehat{\mathcal{G}}(f)} \mathcal{L}(f,G) - \sup_{G \in \mathcal{G}(f)} \mathcal{L}(f,G) \right\} + \left\{ \sup_{G \in \mathcal{G}(f)} \mathcal{L}(f,G) - \sup_{G \in \mathcal{G}(f^*)} \mathcal{L}(f^*,G) \right\},$$

Firstly, both the second and fourth terms can be bounded by $\mathcal{E}_{\text{sta}}$. Specifically, as for the fourth term, we have

$$\sup_{G \in \widehat{\mathcal{G}}(f)} \widehat{\mathcal{L}}(f,G) - \sup_{G \in \widehat{\mathcal{G}}(f)} \mathcal{L}(f,G) \leq \sup_{G \in \widehat{\mathcal{G}}(f)} \{ \widehat{\mathcal{L}}(f,G) - \mathcal{L}(f,G) \} \leq \sup_{G \in \widehat{\mathcal{G}}(f)} \left| \widehat{\mathcal{L}}(f,G) - \mathcal{L}(f,G) \right|$$

$$\leq \sup_{f \in \mathcal{F}, G \in \widehat{\mathcal{G}}(f)} \left| \widehat{\mathcal{L}}(f,G) - \mathcal{L}(f,G) \right|,$$

A similar bound holds for the second term.

Next, we note that the sum of the first and fifth terms can be bounded by $\mathcal{E}_{\mathcal{G}}$. In particular, for the first term

$$\sup_{G \in \mathcal{G}(\hat{f}_{n_s})} \mathcal{L}(\hat{f}_{n_s},G) - \sup_{G \in \widehat{\mathcal{G}}(\hat{f}_{n_s})} \mathcal{L}(\hat{f}_{n_s},G) \leq \sup_{f \in \mathcal{F}} \left\{ \sup_{G \in \mathcal{G}(f)} \mathcal{L}(f,G) - \sup_{G \in \widehat{\mathcal{G}}(f)} \mathcal{L}(f,G) \right\}$$

$$\leq \sup_{f \in \mathcal{F}} \left\{ \sup_{G \in \mathcal{G}(f)} \mathcal{L}(f,G) - \mathcal{L}\big(f,\widehat{G}(f)\big) \right\} = \sup_{f \in \mathcal{F}} \left\{ \mathcal{L}\big(f,G^*(f)\big) - \mathcal{L}\big(f,\widehat{G}(f)\big) \right\}$$

$$\leq \sqrt{2} B_2 \sup_{f \in \mathcal{F}} \left\| G^*(f) - \widehat{G}(f) \right\|_F$$

$$\leq \sqrt{2} B_2 \sup_{f \in \mathcal{F}} \left\| \mathbb{E}_{\boldsymbol{x}_s \sim \mathbb{P}_s} \mathbb{E}_{\mathbf{x}_{s,1},\mathbf{x}_{s,2} \in \mathcal{A}(\boldsymbol{x}_s)} \{ f(\mathbf{x}_{s,1}) f(\mathbf{x}_{s,2})^\top \} - \frac{1}{n_s} \sum_{i=1}^{n_s} f(\mathbf{x}_{s,1}^{(i)}) f(\mathbf{x}_{s,2}^{(i)})^\top \right\|_F. \quad (39)$$

where the second inequality follows from $\widehat{G}(f) \in \widehat{\mathcal{G}}(f)$, while the third inequality follows from the fact that $\ell(\boldsymbol{x}, \cdot) \in \text{Lip}(\sqrt{2}B_2)$, which can be found in Table 9. Meanwhile, regarding the fifth term, we can derive

$$\sup_{G \in \widehat{\mathcal{G}}(f)} \mathcal{L}(f,G) - \sup_{G \in \mathcal{G}(f)} \mathcal{L}(f,G) = \sup_{G \in \widehat{\mathcal{G}}(f)} \mathbb{E}_{\widetilde{D}_s} \left\{ \left\langle \widehat{G}(f), G \right\rangle_F \right\} - \sup_{G \in \mathcal{G}(f)} \langle G^*(f), G \rangle_F$$

$$\leq \mathbb{E}_{\widetilde{D}_s} \left\{ \sup_{G \in \widehat{\mathcal{G}}(f)} \left\langle \widehat{G}(f), G \right\rangle_F \right\} - \sup_{G \in \mathcal{G}(f)} \langle G^*(f), G \rangle_F = \mathbb{E}_{\widetilde{D}_s} \{ \|\widehat{G}(f)\|_F^2 \} - \| G^*(f) \|_F^2$$

$$\leq 2\big(B_2^2 + \sqrt{d^*}\big) \Big( \mathbb{E}_{\widetilde{D}_s} \{ \|\widehat{G}(f)\|_F \} - \| G^*(f) \|_F \Big)$$

$$\leq 2\big(B_2^2 + \sqrt{d^*}\big) \Big( \sup_{f \in \mathcal{F}} \Big[ \mathbb{E}_{\widetilde{D}_s} \{ \|\widehat{G}(f)\|_F \} - \| G^*(f) \|_F \Big] \Big)$$

$$\lesssim \sup_{f \in \mathcal{F}} \left[ \mathbb{E}_{\widetilde{D}_s} \left\{ \left\| \frac{1}{n_s} \sum_{i=1}^{n_s} f(\mathbf{x}_{s,1}^{(i)}) f(\mathbf{x}_{s,2}^{(i)})^\top - I_{d^*} \right\|_F - \left\| \mathbb{E}_{\boldsymbol{x}_s \sim \mathbb{P}_s} \mathbb{E}_{\mathbf{x}_{s,1},\mathbf{x}_{s,2} \in \mathcal{A}(\boldsymbol{x}_s)} \{ f(\mathbf{x}_{s,1}) f(\mathbf{x}_{s,2})^\top \} - I_{d^*} \right\|_F \right\} \right]$$

$$\leq \sup_{f \in \mathcal{F}} \left[ \mathbb{E}_{\widetilde{D}_s} \left\{ \left\| \frac{1}{n_s} \sum_{i=1}^{n_s} f(\mathbf{x}_{s,1}^{(i)}) f(\mathbf{x}_{s,2}^{(i)})^\top - \mathbb{E}_{\boldsymbol{x}_s \sim \mathbb{P}_s} \mathbb{E}_{\mathbf{x}_{s,1},\mathbf{x}_{s,2} \in \mathcal{A}(\boldsymbol{x}_s)} \{ f(\mathbf{x}_{s,1}) f(\mathbf{x}_{s,2})^\top \} \right\|_F \right\} \right]$$

$$\leq \mathbb{E}_{\widetilde{D}_s} \left[ \sup_{f \in \mathcal{F}} \left\{ \left\| \frac{1}{n_s} \sum_{i=1}^{n_s} f(\mathbf{x}_{s,1}^{(i)}) f(\mathbf{x}_{s,2}^{(i)})^\top - \mathbb{E}_{\boldsymbol{x}_s \sim \mathbb{P}_s} \mathbb{E}_{\mathbf{x}_{s,1},\mathbf{x}_{s,2} \in \mathcal{A}(\boldsymbol{x}_s)} \{ f(\mathbf{x}_{s,1}) f(\mathbf{x}_{s,2})^\top \} \right\|_F \right\} \right] \quad (40)$$

where the first equality is derived from $\langle G^*(f), G \rangle_F = \mathbb{E}_{\widetilde{D}_s} \left\{ \left\langle \widehat{G}(f), G \right\rangle_F \right\}$, while the second inequality is derived from $\|\widehat{G}(f)\|_F \leq B_2^2 + \sqrt{d^*}$ and $\|G^*(f)\|_F \leq B_2^2 + \sqrt{d^*}$. Combining (39) and (40) yields $\mathcal{E}_{\mathcal{G}}$.

Furthermore, it is easy to conclude the third term $\sup_{G \in \widehat{\mathcal{G}}(\hat{f}_{n_s})} \widehat{\mathcal{L}}(\hat{f}_{n_s}, G) - \sup_{G \in \widehat{\mathcal{G}}(f)} \widehat{\mathcal{L}}(f, G) \leq 0$ according to the definition of $\hat{f}_{n_s}$. Taking infimum over all $f \in \mathcal{NN}_{d,d^*}(W,L,\mathcal{K},B_1,B_2)$ on both sides yields

$$\mathbb{E}_{\widetilde{D}_s} \{ \mathcal{L}(\hat{f}_{n_s}) \} \lesssim \mathcal{L}(f^*) + \mathcal{E}_{\text{sta}} + \mathcal{E}_{\mathcal{F}} + \mathcal{E}_{\mathcal{G}},$$

which completes the proof. $\qquad \square$

Next, the remaining task is to handle each term on the right-hand side individually.

### F.4.2 Vanishing $\mathcal{L}(f^*)$

In this section we will show the optimal encoder $f^*$ can indeed vanish $\mathcal{L}(f^*)$. The justification comprises a total of two steps. At first, we assert that if there exists a measurable map $f$ such that $\Sigma = \mathbb{E}_{\boldsymbol{x}_s \sim \mathbb{P}_s}\{f(\boldsymbol{x}_s)f(\boldsymbol{x}_s)^\top\}$ be positive definite, then we can conduct a series of minor modifications on $f$ to obtain a $\tilde{f}$ such that $\mathcal{L}(\tilde{f}) = 0$. In the second step, we will demonstrate that the required $f$ indeed exists under Assumption 2, and that the modification $\tilde{f}$ also satisfies the constraint $B_1 \le \|\tilde{f}\|_2 \le B_2$, which implies that $\mathcal{L}(f^*) = 0$, since the definition of $f^*$ indicates that $\mathcal{L}(f^*) \le \mathcal{L}(\tilde{f})$.

To this end, it suffices to find a $\tilde{f} : B_1 \le \|\tilde{f}\|_2 \le B_2$ satisfying both $\mathcal{L}_{\text{align}}(\tilde{f}) = 0$ and $\left\|\mathbb{E}_{\boldsymbol{x}_s \sim \mathbb{P}_s}\mathbb{E}_{\boldsymbol{x_1},\boldsymbol{x_2} \in \mathcal{A}(\boldsymbol{x}_s)}\{f(\mathbf{x}_{s,1})f(\mathbf{x}_{s,2})^\top\} - I_{d^*}\right\|_F = 0$. First note that

$$
\left\|\mathbb{E}_{\boldsymbol{x}_s \sim \mathbb{P}_s}\mathbb{E}_{\boldsymbol{x_1},\boldsymbol{x_2} \in \mathcal{A}(\boldsymbol{x}_s)}\{f(\mathbf{x}_{s,1})f(\mathbf{x}_{s,2})^\top\} - I_{d^*}\right\|_F
$$
$$
= \left\|\mathbb{E}_{\boldsymbol{x}_s \sim \mathbb{P}_s}\mathbb{E}_{\boldsymbol{x_1},\boldsymbol{x_2} \in \mathcal{A}(\boldsymbol{x}_s)}\{f(\mathbf{x}_{s,1})f(\mathbf{x}_{s,1})^\top\} + \mathbb{E}_{\boldsymbol{x}_s \sim \mathbb{P}_s}\mathbb{E}_{\boldsymbol{x_1},\boldsymbol{x_2} \in \mathcal{A}(\boldsymbol{x}_s)}\left[f(\mathbf{x}_{s,1})\{f(\mathbf{x}_{s,2}) - f(\mathbf{x}_{s,1})\}^\top\right] - I_{d^*}\right\|_F
$$
$$
\le \left\|\mathbb{E}_{\boldsymbol{x}_s \sim \mathbb{P}_s}\mathbb{E}_{\mathbf{x}_{s,1} \in \mathcal{A}(\boldsymbol{x}_s)}\{f(\mathbf{x}_{s,1})f(\mathbf{x}_{s,1})^\top\} - I_{d^*}\right\|_F + \mathbb{E}_{\boldsymbol{x}_s \sim \mathbb{P}_s}\mathbb{E}_{\mathbf{x}_{s,1},\mathbf{x}_{s,2} \in \mathcal{A}(\boldsymbol{x}_s)}\left\{\|f(\mathbf{x}_{s,1})\|_2\|f(\mathbf{x}_{s,1}) - f(\mathbf{x}_{s,2})\|_2\right\}
$$
$$
\le \left\|\mathbb{E}_{\boldsymbol{x}_s \sim \mathbb{P}_s}\mathbb{E}_{\mathbf{x}_s \in \mathcal{A}(\boldsymbol{x}_s)}\{f(\mathbf{x}_s)f(\mathbf{x}_s)^\top\} - I_{d^*}\right\|_F + B_2 \mathbb{E}_{\boldsymbol{x}_s \sim \mathbb{P}_s}\mathbb{E}_{\mathbf{x}_{s,1},\mathbf{x}_{s,2} \in \mathcal{A}(\boldsymbol{x}_s)}\|f(\mathbf{x}_{s,1}) - f(\mathbf{x}_{s,2})\|_2.
$$
$$
(\|f\|_2 \le B_2)
$$

It reveals that, to achieve our destination, we just need to construct a $\tilde{f} : B_1 \le \|\tilde{f}\|_2 \le B_2$ such that $\mathcal{L}_{\text{align}}(\tilde{f}) = 0$, and well as $\left\|\mathbb{E}_{\boldsymbol{x}_s \sim \mathbb{P}_s}\mathbb{E}_{\mathbf{x}_s \in \mathcal{A}(\boldsymbol{x}_s)}\{\tilde{f}(\mathbf{x}_s)\tilde{f}(\mathbf{x}_s)^\top\} - I_{d^*}\right\|_F = 0$. To this end, we provide following lemma:

**Lemma 10.** *If there exists a measurable encoder $f$ making $\Sigma = \mathbb{E}_{\boldsymbol{x}_s \sim \mathbb{P}_s}\{f(\boldsymbol{x}_s)f(\boldsymbol{x}_s)^\top\}$ positive definite, then there exists a measurable encoder $\tilde{f}$ such that*

$$
\mathcal{L}_{\text{align}}(\tilde{f}) = 0, \quad \left\|\mathbb{E}_{\boldsymbol{x}_s \sim \mathbb{P}_s}\mathbb{E}_{\mathbf{x}_s \in \mathcal{A}(\boldsymbol{x}_s)}\{\tilde{f}(\mathbf{x}_s)\tilde{f}(\mathbf{x}_s)^\top\} - I_{d^*}\right\|_F = 0.
$$

*Proof.* We conduct following modifications on the given $f$ as follows: for any $\boldsymbol{x}_s \in \mathcal{X}_s$, define

$$
\tilde{f}_{\boldsymbol{x}_s}(\mathbf{x}_s) = \begin{cases} V^{-1}f(\boldsymbol{x}_s) & \text{if } \mathbf{x}_s \in \mathcal{A}(\boldsymbol{x}_s) \\ f(\boldsymbol{x}_s) & \text{if } \mathbf{x}_s \notin \mathcal{A}(\boldsymbol{x}_s) \end{cases}
$$

where $\Sigma = VV^\top$, which is the Cholesky decomposition of $\Sigma$. Here the positivity of $\Sigma$ ensure $V$ is well-defined. Iteratively repeat this modification for all $\boldsymbol{x}_s \in \mathcal{X}$ to yield $\tilde{f}$. As the result, we have

$$
\mathbb{E}_{\boldsymbol{x}_s \sim \mathbb{P}_s}\mathbb{E}_{\mathbf{x}_s \in \mathcal{A}(\boldsymbol{x}_s)}\{\tilde{f}(\mathbf{x}_s)\tilde{f}(\mathbf{x}_s)^\top\} = V^{-1}\mathbb{E}_{\boldsymbol{x}_s \sim \mathbb{P}_s}\{f(\boldsymbol{x}_s)f(\boldsymbol{x}_s)^\top\}V^{-\top} = I_{d^*}
$$

and

$$
\forall \boldsymbol{x}_s \in \mathcal{X}, \mathbf{x}_{s,1}, \mathbf{x}_{s,2} \in \mathcal{A}(\boldsymbol{x}_s), \left\|\tilde{f}(\mathbf{x}_{s,1}) - \tilde{f}(\mathbf{x}_{s,2})\right\|_2 = \left\|\tilde{f}(\boldsymbol{x}_s) - \tilde{f}(\boldsymbol{x}_s)\right\|_2 = 0.
$$

That is precisely what we desire. $\qquad\square$

*Remark* 2. Based on the construction approach in Lemma 10, we just need to show there exists a encoder $f$ such that $\Sigma$ are positive definite. In fact, if we have a measurable partition $\mathcal{X} = \cup_{i=1}^{d^*}\mathcal{P}_i$ as shown in Assumption 2 such that $\mathcal{P}_i \cap \mathcal{P}_j = \emptyset$ and $\forall i \in [d^*], \frac{1}{B_2^2} \le \mathbb{P}_s(\mathcal{P}_i) \le \frac{1}{B_1^2}$, just set the $f(\boldsymbol{x}_s) = \boldsymbol{e}_i$ if $\boldsymbol{x}_s \in \mathcal{P}_i$, where $\boldsymbol{e}_i$ is the standard basis of $\mathbb{R}^{d^*}$, then $\Sigma = \text{diag}\{\mathbb{P}_s(\mathcal{P}_1), \ldots, \mathbb{P}_s(\mathcal{P}_i), \ldots, \mathbb{P}_s(\mathcal{P}_{d^*})\}, V^{-1} = \text{diag}\left\{\sqrt{\frac{1}{\mathbb{P}_s(\mathcal{P}_1)}}, \ldots, \sqrt{\frac{1}{\mathbb{P}_s(\mathcal{P}_i)}}, \ldots, \sqrt{\frac{1}{\mathbb{P}_s(\mathcal{P}_{d^*})}}\right\}, \tilde{f}(\boldsymbol{x}_s) = \sqrt{\frac{1}{\mathbb{P}_s(\mathcal{P}_i)}}\boldsymbol{e}_i$ if $\boldsymbol{x}_s \in \mathcal{P}_i$, it is obvious that $B_1 \le \|\tilde{f}\|_2 \le B_2$.

### F.4.3 Upper bound of $\mathcal{E}_{\mathrm{sta}}$

**Lemma 11.** *Regarding the statistical error $\mathcal{E}_{\mathrm{sta}}$, we have*

$$\mathcal{E}_{\mathrm{sta}} \lesssim \frac{\mathcal{K}\sqrt{L}}{\sqrt{n_s}}.$$

*Proof.* To obtain the desired conclusion, it is necessary to clarify several definitions in advance. For any $f : \mathbb{R}^d \to \mathbb{R}^{d^*}$, define $\tilde{f} : \mathbb{R}^{2d} \to \mathbb{R}^{2d^*}$ such that $\tilde{f}(\tilde{x}_s) = (f(x_{s,1}), f(x_{s,2}))$, where $\tilde{x}_s = (x_{s,1}, x_{s,2}) \in \mathbb{R}^{2d}$. Furthermore, let $\widetilde{\mathcal{F}} := \{\tilde{f} : f \in \mathcal{NN}_{d,d^*}(W, L, \mathcal{K})\}$. In addition, denote $\widetilde{D}'_s = \{\tilde{x}_s'^{(i)}\}_{i=1}^{n_s}$, which is a collection consisting of $n_s$ independent samples. The distribution of these samples is identical to that of $\widetilde{D}_s$; $\widetilde{D}'_s$ is therefore referred to as the ghost samples of $\widetilde{D}_s$. Moreover, recall $\mathcal{G}_1 := \{G \in \mathbb{R}^{d^* \times d^*} : \|G\|_F \le B_2^2 + \sqrt{d^*}\}$, by the definition of $\mathcal{E}_{\mathrm{sta}}$, we have:

$$\mathcal{E}_{\mathrm{sta}} = \mathbb{E}_{\widetilde{D}_s}\left\{ \sup_{f \in \mathcal{NN}_{d,d^*}(W,L,\mathcal{K},B_1,B_2), G \in \widehat{\mathcal{G}}(f)} \left| \mathcal{L}(f, G) - \widehat{\mathcal{L}}(f, G) \right| \right\}$$

$$\le \mathbb{E}_{\widetilde{D}_s}\left\{ \sup_{(f,G) \in \mathcal{NN}_{d,d^*}(W,L,\mathcal{K},B_1,B_2) \times \mathcal{G}_1} \left| \mathcal{L}(f, G) - \widehat{\mathcal{L}}(f, G) \right| \right\}$$

$$(\forall f \in \mathcal{NN}_{d,d^*}(W, L, \mathcal{K}, B_1, B_2), \widehat{\mathcal{G}}(f) \subseteq \mathcal{G}_1)$$

$$\le \mathbb{E}_{\widetilde{D}_s}\left\{ \sup_{(f,G) \in \mathcal{NN}_{d,d^*}(W,L,\mathcal{K}) \times \mathcal{G}_1} \left| \mathcal{L}(f, G) - \widehat{\mathcal{L}}(f, G) \right| \right\}$$

$$(\mathcal{NN}_{d,d^*}(W, L, \mathcal{K}, B_1, B_2) \subseteq \mathcal{NN}_{d,d^*}(W, L, \mathcal{K}))$$

$$= \mathbb{E}_{\widetilde{D}_s}\left\{ \sup_{(\tilde{f},G) \in \widetilde{\mathcal{F}} \times \mathcal{G}_1} \left| \frac{1}{n_s} \sum_{i=1}^{n_s} \mathbb{E}_{\widetilde{D}'_s}\{\ell(\tilde{f}(\tilde{x}_s'^{(i)}), G)\} - \frac{1}{n_s} \sum_{i=1}^{n_s} \ell(\tilde{f}(\tilde{x}_s^{(i)}), G) \right| \right\}$$

$$\le \mathbb{E}_{\widetilde{D}_s, \widetilde{D}'_s}\left\{ \sup_{(\tilde{f},G) \in \widetilde{\mathcal{F}} \times \mathcal{G}_1} \left| \frac{1}{n_s} \sum_{i=1}^{n_s} \ell(\tilde{f}(\tilde{x}_s'^{(i)}), G) - \frac{1}{n_s} \sum_{i=1}^{n_s} \ell(\tilde{f}(\tilde{x}_s^{(i)}), G) \right| \right\}$$

$$= \mathbb{E}_{\widetilde{D}_s, \widetilde{D}'_s, \xi}\left\{ \sup_{(\tilde{f},G) \in \widetilde{\mathcal{F}} \times \mathcal{G}_1} \left| \frac{1}{n_s} \sum_{i=1}^{n_s} \xi_i \big( \ell(\tilde{f}(\tilde{x}_s'^{(i)}), G) - \ell(\tilde{f}(\tilde{x}_s^{(i)}), G) \big) \right| \right\} \tag{41}$$

$$\le 2\mathbb{E}_{\widetilde{D}_s, \xi}\left\{ \sup_{(\tilde{f},G) \in \widetilde{\mathcal{F}} \times \mathcal{G}_1} \left| \frac{1}{n_s} \sum_{i=1}^{n_s} \xi_i \ell(\tilde{f}(\tilde{x}_s^{(i)}), G) \right| \right\}$$

$$\le 4\sqrt{2}\, \|\ell\|_{\mathrm{Lip}} \left( \mathbb{E}_{\widetilde{D}_s, \xi}\left\{ \sup_{f \in \mathcal{NN}_{d,d^*}(W,L,\mathcal{K})} \left| \frac{1}{n_s} \sum_{i=1}^{n_s} \sum_{j=1}^{d^*} \xi_{i,j,1} f_j(x_{s,1}^{(i)}) + \xi_{i,j,2} f_j(x_{s,2}^{(i)}) \right| \right\} \right.$$

$$\left. + \mathbb{E}_\xi\left\{ \sup_{G \in \mathcal{G}_1} \left| \frac{1}{n_s} \sum_{i=1}^{n_s} \sum_{j=1}^{d^*} \sum_{k=1}^{d^*} \xi_{i,j,k} G_{jk} \right| \right\} \right) \tag{42}$$

$$\le 8\sqrt{2}\, \|\ell\|_{\mathrm{Lip}} \mathbb{E}_{\widetilde{D}_s, \xi}\left\{ \sup_{f \in \mathcal{NN}_{d,d^*}(W,L,\mathcal{K})} \left| \frac{1}{n_s} \sum_{i=1}^{n_s} \sum_{j=1}^{d^*} \xi_{i,j,1} f_j(x_{s,1}^{(i)}) \right| \right\} + 4\sqrt{2} d^* \|\ell\|_{\mathrm{Lip}}\, \varrho$$

$$+ 4\sqrt{2}\, \|\ell\|_{\mathrm{Lip}} \mathbb{E}_\xi\left\{ \max_{G \in \mathcal{N}_{\mathcal{G}_1}(\varrho)} \left| \frac{1}{n_s} \sum_{i=1}^{n_s} \sum_{j=1}^{d^*} \sum_{k=1}^{d^*} \xi_{i,j,k} G_{jk} \right| \right\} \tag{43}$$

$$\le 8\sqrt{2}\, \|\ell\|_{\mathrm{Lip}} \mathbb{E}_{\widetilde{D}_s, \xi}\left\{ \sup_{f \in \mathcal{NN}_{d,d^*}(W,L,\mathcal{K})} \left| \frac{1}{n_s} \sum_{i=1}^{n_s} \sum_{j=1}^{d^*} \xi_{i,j} f_j(x_{s,1}^{(i)}) \right| \right\} + 4\sqrt{2} d^* \|\ell\|_{\mathrm{Lip}}\, \varrho$$

$$+ 4\sqrt{2}(B_2^2 + \sqrt{d^*}) \|\ell\|_{\mathrm{Lip}} \sqrt{\frac{2\log\big(2\,|\mathcal{N}_{\mathcal{G}_1}(\varrho)|\big)}{n_s}} \tag{44}$$

$$\le 8\sqrt{2} d^* \|\ell\|_{\mathrm{Lip}} \mathbb{E}_{\widetilde{D}_s, \xi}\left\{ \sup_{f \in \mathcal{NN}_{d,1}(W,L,\mathcal{K})} \left| \frac{1}{n_s} \sum_{i=1}^{n_s} \xi_i f(x_{s,1}^{(i)}) \right| \right\} + 4\sqrt{2} d^* \|\ell\|_{\mathrm{Lip}}\, \varrho$$

$$+ 4\sqrt{2}(B_2^2 + \sqrt{d^*}) \|\ell\|_{\mathrm{Lip}} \sqrt{\frac{2\log\left(2(\frac{3}{(B_2^2+\sqrt{d^*})\varrho})^{(d^*)^2}\right)}{n_2}} \quad (|\mathcal{N}_{\mathcal{G}_1}(\varrho)| \le (\frac{3}{(B_2^2+\sqrt{d^*})\varrho})^{(d^*)^2})$$

$$\lesssim \frac{\mathcal{K}\sqrt{L}}{\sqrt{n_s}} + \sqrt{\frac{\log n_s}{n_s}} \qquad \qquad \text{(Lemma 8 and set } \varrho = \mathcal{O}(1/\sqrt{n_s}))$$

$$\lesssim \frac{\mathcal{K}\sqrt{L}}{\sqrt{n_s}} \qquad \qquad \text{(If } \mathcal{K} \gtrsim \sqrt{\log n_s})$$

Where (41) stems from the fact that $\xi_i\big(\ell(\tilde{f}(\tilde{\mathbf{x}}_s'^{(i)}), G) - \ell(\tilde{f}(\tilde{\mathbf{x}}_s^{(i)}), G)\big)$ has identical distribution with $\ell(\tilde{f}(\tilde{\mathbf{x}}_s'^{(i)}), G) - \ell(\tilde{f}(\tilde{\mathbf{x}}_s^{(i)}), G)$. In addition, notice that we have shown $\|\ell\|_{\mathrm{Lip}} < \infty$, applying Lemma 6 obtains (42). Regarding 43, since $\mathcal{N}_{\mathcal{G}_1}(\varrho)$ is a $\varrho$-covering, thus for any fixed $G \in \mathcal{G}_1$, we can find a $\widetilde{G} \in \mathcal{N}_{\mathcal{G}_1}(\varrho)$ satisfying $\left\|G - \widetilde{G}\right\|_F \le \varrho$, therefore we have

$$\mathbb{E}_\xi \Big\{ \max_{G \in \mathcal{G}_1} \Big| \frac{1}{n_s} \sum_{i=1}^{n_s} \sum_{j=1}^{d^*} \sum_{k=1}^{d^*} \xi_{i,j,k}\big(\widetilde{G}_{jk} + G_{jk} - \widetilde{G}_{jk}\big) \Big| \Big\}$$

$$\le \mathbb{E}_\xi \Big\{ \max_{G \in \mathcal{G}_1} \Big| \frac{1}{n_s} \sum_{i=1}^{n_s} \sum_{j=1}^{d^*} \sum_{k=1}^{d^*} \xi_{i,j,k}\widetilde{G}_{jk} \Big| \Big\} + \mathbb{E}_\xi \Big\{ \max_{G \in \mathcal{G}_1} \Big| \frac{1}{n_s} \sum_{i=1}^{n_s} \sum_{j=1}^{d^*} \sum_{k=1}^{d^*} \xi_{i,j,k}\big(G_{jk} - \widetilde{G}_{jk}\big) \Big| \Big\}$$

$$\le \mathbb{E}_\xi \Big\{ \max_{G \in \mathcal{N}_{\mathcal{G}_1}(\varrho)} \Big| \frac{1}{n_s} \sum_{i=1}^{n_s} \sum_{j=1}^{d^*} \sum_{k=1}^{d^*} \xi_{i,j,k}G_{jk} \Big| \Big\} + \frac{1}{n_s}\sqrt{(d^*)^2 n_s} \sqrt{n_s \sum_{j=1}^{d^*} \sum_{k=1}^{d^*} \big(G_{jk} - \widetilde{G}_{jk}\big)^2}$$

$$\text{(Cauchy-Schwarz inequality)}$$

$$\le \mathbb{E}_\xi \Big\{ \max_{G \in \mathcal{N}_{\mathcal{G}_1}(\varrho)} \Big| \frac{1}{n_s} \sum_{i=1}^{n_s} \sum_{j=1}^{d^*} \sum_{k=1}^{d^*} \xi_{i,j,k}G_{jk} \Big| \Big\} + d^*\varrho.$$

To handle the last term of (44), notice that $\|G\|_F \le B_2^2 + \sqrt{d^*}$ implies that $\sum_{j=1}^{d^*} \sum_{k=1}^{d^*} \xi_{i,j,k}G_{jk} \sim \mathrm{subG}(B_2^2 + \sqrt{d^*})$. Therefore, $\frac{1}{n_s}\sum_{i=1}^{n_s} \sum_{j=1}^{d^*} \sum_{k=1}^{d^*} \xi_{i,j,k}G_{jk} \sim \mathrm{subG}(B_2^2 + \sqrt{d^*})$, just apply Lemma 7 to complete the proof. $\qquad\qquad\qquad\qquad\qquad\qquad\qquad\qquad\qquad\square$

### F.4.4  Upper bound of $\mathcal{E}_\mathcal{F}$

If we define

$$\mathcal{E}(\mathcal{H}^\alpha, \mathcal{NN}_{d,1}(W, L, \mathcal{K})) := \sup_{g \in \mathcal{H}^\alpha} \inf_{f \in \mathcal{NN}_{d,1}(W,L,\mathcal{K})} \|f - g\|_{C([0,1]^d)},$$

where $C([0,1]^d)$ is the space of continuous functions on $[0,1]^d$ equipped with the sup-norm. According to Theorem 3.2 of [25], we have following lemma:

**Lemma 12** (Theorem 3.2 of [25]). *Let $d \in \mathbb{N}$ and $\alpha = r + \beta > 0$, where $r \in \mathbb{N}_0$ and $\beta \in (0,1]$. There exists $c > 0$ such that for any $\mathcal{K} \ge 1$, any $W \ge c\mathcal{K}^{(2d+\alpha)/(2d+2)}$ and $L \ge 2\lceil \log_2(d+r) \rceil + 2$,*

$$\mathcal{E}(\mathcal{H}^\alpha, \mathcal{NN}_{d,1}(W, L, \mathcal{K})) \lesssim \mathcal{K}^{-\alpha/(d+1)}.$$

Based on Lemma 12, we yield

$$\inf_{f \in \mathcal{NN}_{d,d^*}(W,L,\mathcal{K})} \|f(\boldsymbol{x}) - f^*(\boldsymbol{x})\|_2 = \inf_{f \in \mathcal{NN}_{d,d^*}(W,L,\mathcal{K})} \sqrt{\sum_{i=1}^{d^*} \big\{f_i(\boldsymbol{x}) - f_i^*(\boldsymbol{x})\big\}^2}$$

$$\le \inf_{f \in \mathcal{NN}_{d,d^*}(W,L,\mathcal{K})} \sqrt{\sum_{i=1}^{d^*} \|f_i - f_i^*\|_{C([0,1]^d)}^2} \le \sup_{g \in \mathcal{H}^\alpha} \inf_{f \in \mathcal{NN}_{d,d^*}(W,L,\mathcal{K})} \sqrt{\sum_{i=1}^{d^*} \|f_i - g\|_{C([0,1]^d)}^2}$$

$$\leq \sup_{g \in \mathcal{H}^\alpha} \sqrt{\sum_{i=1}^{d^*} \inf_{f \in \mathcal{NN}_{d,1}(\lfloor W/d^* \rfloor, L, \mathcal{K})} \|f - g\|^2_{C([0,1]^d)}} \leq \sqrt{d^*} \mathcal{E}\big(\mathcal{H}^\alpha, \mathcal{NN}_{d,1}(\lfloor W/d^* \rfloor, L, \mathcal{K})\big)$$

$$\lesssim \mathcal{K}^{-\alpha/(d+1)},$$

where the third inequality is because following fact: if we have a total of $d^*$ function $f_i \in \mathcal{NN}_{d,1}(\lfloor W/d^* \rfloor, L, \mathcal{K}), i \in [d^*]$ of independent parameters, according to following Proposition 1, the concatenation $f = (f_1, f_2, \cdots, f_{d^*})^\top$ can be regarded as an elements of $\mathcal{NN}_{d,d^*}(W, D, \mathcal{K})$ with specific parameters, that is, $f \in \mathcal{NN}_{d,d^*}(W, L, \mathcal{K})$.

**Proposition 1** ((iii) of Proposition 2.5 in [25]). *Let* $f_1 \in \mathcal{NN}_{d,d_1^*}(W_1, L_1, \mathcal{K}_1)$ *and* $f_2 \in \mathcal{NN}_{d,d_2^*}(W_2, L_2, \mathcal{K}_2)$, *define* $f(\boldsymbol{x}_s) := (f_1(\boldsymbol{x}_s), f_2(\boldsymbol{x}_s))$, *then* $f \in \mathcal{NN}_{d,d_1^*+d_2^*}(W_1 + W_2, \max\{L_1, L_2\}, \max\{\mathcal{K}_1, \mathcal{K}_2\})$.

Above conclusion implies optimal approximation element of $f^*$ in $\mathcal{NN}_{d,d^*}(W, L, \mathcal{K})$ can be arbitrarily close to $f^*$ under the setting that $\mathcal{K}$ is large enough. Hence we can conclude optimal approximation element of $f^*$ is also contained in $\mathcal{F} = \mathcal{NN}_{d,d^*}(W, L, \mathcal{K}, B_1, B_2)$ under the setting that $B_1 \leq \|f^*\|_2 \leq B_2$. Therefore, if we denote

$$\mathcal{T}(f) := \mathbb{E}_{\boldsymbol{x}_s \sim \mathbb{P}_s} \mathbb{E}_{\mathbf{x}_{s,1}, \mathbf{x}_{s,2} \in \mathcal{A}(\boldsymbol{x}_s)} \left\{ \left\| f(\mathbf{x}_{s,1}) - f(\mathbf{x}_{s,2}) \right\|_2^2 \right\} + \lambda \left\| \mathbb{E}_{\boldsymbol{x}_s \sim \mathbb{P}_s} \mathbb{E}_{\mathbf{x}_{s,1}, \mathbf{x}_{s,2} \in \mathcal{A}(\boldsymbol{x}_s)} \left\{ f(\mathbf{x}_{s,1}) f(\mathbf{x}_{s,2})^\top \right\} - I_{d^*} \right\|_F^2,$$

then we have

$$\mathcal{E}_\mathcal{F} = \inf_{f \in \mathcal{F}} \left\{ \sup_{G \in \mathcal{G}(f)} \mathcal{L}(f, G) - \sup_{G \in \mathcal{G}(f^*)} \mathcal{L}(f^*, G) \right\} = \inf_{f \in \mathcal{F}} \left\{ \mathcal{T}(f) - \mathcal{T}(f^*) \right\}$$

$$= \inf_{f \in \mathcal{NN}_{d,d^*}(W,L,\mathcal{K})} \left\{ \mathcal{T}(f) - \mathcal{T}(f^*) \right\} \leq \|\ell\|_{\text{Lip}} \inf_{f \in \mathcal{NN}_{d,d^*}(W,L,\mathcal{K})} \mathbb{E}_{\boldsymbol{x}_s \sim \mathbb{P}_s} \mathbb{E}_{\tilde{\mathbf{x}}_s} \left\| \tilde{f}(\tilde{\mathbf{x}}_s) - \tilde{f}^*(\tilde{\mathbf{x}}_s) \right\|_2$$

$$\leq \|\ell\|_{\text{Lip}} \inf_{f \in \mathcal{NN}_{d,d^*}(W,L,\mathcal{K})} \mathbb{E}_{\boldsymbol{x}_s \sim \mathbb{P}_s} \mathbb{E}_{\mathbf{x}_s \in \mathcal{A}(\boldsymbol{x}_s)} \sqrt{2 \sum_{i=1}^{d^*} \left\{ f_i(\mathbf{x}_s) - f_i^*(\mathbf{x}_s) \right\}^2}$$

$$\leq \sqrt{2d^*} \|\ell\|_{\text{Lip}} \sup_{g \in \mathcal{H}^\alpha} \inf_{f \in \mathcal{NN}_{d,1}(\lfloor W/d^* \rfloor, L, \mathcal{K}/\sqrt{d^*})} \|f - g\|_{C([0,1]^d)}$$

$$\leq \sqrt{2d^*} \|\ell\|_{\text{Lip}} \mathcal{E}\big(\mathcal{H}^\alpha, \mathcal{NN}_{d,1}(\lfloor W/d^* \rfloor, L, \mathcal{K}/\sqrt{d^*})\big)$$

$$\lesssim \mathcal{K}^{-\alpha/(d+1)}.$$

where the first inequality is because of Proposition 2.

### F.4.5 Upper bound of $\mathcal{E}_\mathcal{G}$

Let $\mathcal{M}(\boldsymbol{x}) = \boldsymbol{x}_1 \boldsymbol{x}_2^\top$, $\boldsymbol{x}_1, \boldsymbol{x}_2 \in \mathbb{R}^{d^*}$, which is a Lipchitz map on $\{\boldsymbol{x} \in \mathbb{R}^{2d^*} : \boldsymbol{x} \leq \sqrt{2} B_2\}$ as presented in Proposition 2. Then

$$\mathcal{E}_\mathcal{G} \lesssim \mathbb{E}_{\widetilde{D}_s} \left\{ \sup_{f \in \mathcal{F}} \left\| \mathbb{E}_{\boldsymbol{x}_s \sim \mathbb{P}_s} \mathbb{E}_{\mathbf{x}_{s,1}, \mathbf{x}_{s,2} \in \mathcal{A}(\boldsymbol{x}_s)} \left[ \frac{1}{n_s} \sum_{i=1}^{n_s} \left\{ \mathcal{M}(\tilde{f}(\tilde{\mathbf{x}}_s)) - \mathcal{M}(\tilde{f}(\tilde{\mathbf{x}}_s^{(i)})) \right\} \right] \right\|_F \right\}$$

$$\leq \|\mathcal{M}\|_{\text{Lip}} \mathbb{E}_{\widetilde{D}_s} \left[ \left\| \mathbb{E}_{\boldsymbol{x}_s \sim \mathbb{P}_s} \mathbb{E}_{\mathbf{x}_{s,1}, \mathbf{x}_{s,2} \in \mathcal{A}(\boldsymbol{x}_s)} \{\tilde{f}(\tilde{\mathbf{x}}_s)\} - \frac{1}{n_s} \sum_{i=1}^{n_s} \tilde{f}(\tilde{\mathbf{x}}_s^{(i)}) \right\|_2 \right]$$

Furthermore, according to the multidimensional Chebyshev's inequality, we can turn out
$$\mathbb{P}_s \left( \left\| \frac{1}{n_s} \sum_{i=1}^{n_s} \tilde{f}(\tilde{\mathbf{x}}_s^{(i)}) - \mathbb{E}_{\boldsymbol{x}_s \sim \mathbb{P}_s} \mathbb{E}_{\mathbf{x}_{s,1}, \mathbf{x}_{s,2} \in \mathcal{A}(\boldsymbol{x}_s)} \{\tilde{f}(\tilde{\mathbf{x}}_s)\} \right\|_2 \geq \frac{1}{n_s^{1/4}} \right) \leq \frac{\mathbb{E} \|\tilde{f}(\tilde{\mathbf{x}}_s) - \mathbb{E}\{\tilde{f}(\tilde{\mathbf{x}}_s)\}\|_2^2}{\sqrt{n_s}} \leq \frac{8B_2^2}{\sqrt{n_s}}$$
as $\|\tilde{f}(\tilde{\mathbf{x}}_s)\|_2 \leq \sqrt{2} B_2$. Therefore,

$$\mathcal{E}_\mathcal{G} \lesssim \frac{1}{n_s^{1/4}} \cdot \mathbb{P}_s \left( \left\| \frac{1}{n_s} \sum_{i=1}^{n_s} \tilde{f}(\tilde{\mathbf{x}}_s^{(i)}) - \mathbb{E}_{\boldsymbol{x}_s \sim \mathbb{P}_s} \mathbb{E}_{\mathbf{x}_{s,1}, \mathbf{x}_{s,2} \in \mathcal{A}(\boldsymbol{x}_s)} \{\tilde{f}(\tilde{\mathbf{x}}_s)\} \right\|_2 \geq \frac{1}{n_s^{1/4}} \right) + 2\sqrt{2} B_2 \cdot \frac{8B_2^2}{\sqrt{n_s}}$$

$$\leq \frac{1}{n_s^{1/4}} + 16\sqrt{2} B_2^3 \frac{1}{\sqrt{n_s}} \lesssim \frac{1}{n_s^{1/4}},$$

where the first inequity is due to $\left\| \tilde{f}(\tilde{\mathbf{x}}_s) \right\|_2 \leq \sqrt{2} B_2$.

### F.4.6 Trade-off on several errors

Let $W \geq c\mathcal{K}^{(2d+\alpha)/(2d+2)}$ and $L \geq 2\lceil \log_2(d+r) \rceil + 2$, combining all bounds yields

$$\mathbb{E}_{\widetilde{D}_s}\{\mathcal{L}(\hat{f}_{n_s})\} \lesssim \mathcal{E}_{\text{sta}} + \mathcal{E}_{\mathcal{F}} + \mathcal{E}_{\mathcal{G}} \lesssim \frac{\mathcal{K}}{\sqrt{n_s}} + \mathcal{K}^{-\alpha/(d+1)}.$$

Setting $\mathcal{K} \asymp n_s^{\frac{d+1}{2(\alpha+d+1)}}$ yields $\mathbb{E}_{\widetilde{D}_s}\{\mathcal{L}(\hat{f}_{n_s})\} \lesssim n_s^{-\frac{\alpha}{2(\alpha+d+1)}}$ under conditions $W \geq cn_s^{\frac{2d+\alpha}{4(\alpha+d+1)}}$ $L \geq 2\lceil \log_2(d+r) \rceil + 2$.

### F.4.7 The proof of primary theorem

Based on the previous preparation, we next prove the primary theorem 1. Before that, we summary here all crucial conclusions which have obtained so far.

- If $W \gtrsim n_s^{\frac{2d+\alpha}{4(\alpha+d+1)}}, L \geq 2\lceil \log_2(d+r) \rceil + 2, \mathcal{K} \asymp n_s^{\frac{d+1}{2(\alpha+d+1)}}$, then $\mathbb{E}_{\widetilde{D}_s}\{\mathcal{L}(\hat{f}_{n_s})\} \lesssim n_s^{-\frac{\alpha}{2(\alpha+d+1)}}$.

- According to Assumption 4, $\max\{\delta_s^{(n_s)}, \delta_t^{(n_s)}\} \lesssim n_s^{-\frac{\epsilon_\mathcal{A}+d+1}{2(\alpha+d+1)}}, \min\{\sigma_s^{(n_s)}, \sigma_t^{(n_s)}\} \to 1$ when $n_s \to \infty$.

- According to Assumption 5, $\epsilon_1 \lesssim n_s^{-\frac{\epsilon_{\text{ds}}+d+1}{2(\alpha+d+1)}}, \epsilon_2 \lesssim n_s^{-\frac{\epsilon_{\text{ds}}}{2(\alpha+d+1)}}$.

- According to Lemma 1, we have

$$\mathbb{E}_{\widetilde{D}_s}\left\{ \max_{i \neq j} |\mu_t(i)^\top \mu_t(j)| \right\} \lesssim \sqrt{\mathbb{E}_{\widetilde{D}_s}\{\mathcal{L}(\hat{f}_{n_s})\} + \mathbb{E}_{\widetilde{D}_s}\{\varphi(\sigma_s^{(n_s)}, \delta_s^{(n_s)}, \varepsilon_{n_s}, \hat{f}_{n_s})\}} + \mathcal{K}\epsilon_1.$$

where $\mathbb{E}_{\widetilde{D}_s}\left\{ \varphi(\sigma_s^{(n_s)}, \delta_s^{(n_s)}, \varepsilon_{(n_s)}, R_s(\varepsilon, \hat{f}_{n_s})) \right\} \lesssim \left(1 - \sigma_s^{(n_s)} + \mathcal{K}\delta_s^{(n_s)} + 2\varepsilon_{n_s}\right)^2 + \frac{1}{\varepsilon_{n_s}}\sqrt{\mathbb{E}_{\widetilde{D}_s}\{\mathcal{L}(\hat{f}_{n_s})\}}\left(3 - 2\sigma_s^{(n_s)} + \mathcal{K}\delta_s^{(n_s)} + 2\varepsilon_{n_s}\right) + \frac{1}{\varepsilon_{n_s}^2}\mathbb{E}_{\widetilde{D}_s}\{\mathcal{L}(\hat{f}_{n_s})\} + (1 - \sigma_s^{(n_s)}) + \left(\varepsilon_{n_s}^2 + \frac{1}{\varepsilon_{n_s}}\sqrt{\mathbb{E}_{\widetilde{D}_s}\{\mathcal{L}(\hat{f}_{n_s})\}}\right)^{\frac{1}{2}}$. Furthermore, if $\max_{i \neq j} |\mu_t(i)^\top \mu_t(j)| < B_2^2 \psi(\sigma_t, \delta_t, \varepsilon, f)$, then

$$\text{Err}(Q_{\hat{f}_{n_s}}) \lesssim (1 - \sigma_t) + \frac{1}{\varepsilon}\sqrt{\mathcal{L}(\hat{f}_{n_s}) + \mathcal{K}\epsilon_1 + \epsilon_2}, \tag{45}$$

In addition, the following inequalities always hold

$$\mathbb{E}_{\widetilde{D}_s}\{R_s(\varepsilon_{n_s}, \hat{f}_{n_s})\} \lesssim \frac{1}{\varepsilon}\sqrt{\mathbb{E}_{\widetilde{D}_s}\{\mathcal{L}(\hat{f}_{n_s})\}} \tag{46}$$

$$\mathbb{E}_{\widetilde{D}_s}\{R_t(\varepsilon_{n_s}, \hat{f}_{n_s})\} \lesssim \frac{1}{\varepsilon}\sqrt{\mathbb{E}_{\widetilde{D}_s}\{\mathcal{L}(\hat{f}_{n_s})\} + \mathcal{K}\epsilon_1 + \epsilon_2}.$$

**Theorem 1.** *When Assumptions 1-5 all hold, set $\varepsilon_{n_s} \asymp n_s^{-\frac{\min\{\alpha, \epsilon_{\text{ds}}, \epsilon_\mathcal{A}\}}{8(\alpha+d+1)}}, W \gtrsim n_s^{\frac{2d+\alpha}{4(\alpha+d+1)}}, L \geq 2\lceil \log_2(d+r) \rceil + 2, \mathcal{K} \asymp n_s^{\frac{d+1}{2(\alpha+d+1)}}$ and $\mathcal{A} = \mathcal{A}_{n_s}$ in Assumption 4, then we have*

$$\mathbb{E}_{\widetilde{D}_s, \widetilde{D}_t}\{\text{Err}(Q_{\hat{f}_{n_s}})\} \lesssim (1 - \sigma_s^{(n_s)}) + n_s^{-\frac{\min\{\alpha, \epsilon_\mathcal{A}, \epsilon_{\text{ds}}\}}{32(\alpha+d+1)}} + \frac{1}{\min_k \sqrt{n_t(k)}}$$

*for sufficiently large $n_s$.*

*Proof.* Define $\mathcal{C} = \left\{ \max_{i \neq j} |\mu_t(i)^\top \mu_t(j)| < B_2^2 \psi(\sigma_t^{(n_s)}, \delta_t^{(n_s)}, \varepsilon_{n_s}, \hat{f}_{n_s}) \right\}$, which is a event defined on the product measure space $(\mathcal{X}_s \times \mathcal{X}_t, \mathbb{P})$ with $\mathbb{P}$ is the joint distribution on $\mathcal{X}_s \times \mathcal{X}_t$.

$$\mathbb{E}_{\widetilde{D}_s, \widetilde{D}_t}\{\text{Err}(Q_{\hat{f}_{n_s}})\} = \mathbb{E}_{\widetilde{D}_s, \widetilde{D}_t}\{\text{Err}(Q_{\hat{f}_{n_s}})\mathbb{1}_\mathcal{C}\} + \mathbb{E}_{\widetilde{D}_s, \widetilde{D}_t}\{\text{Err}(Q_{\hat{f}_{n_s}})\mathbb{1}_{\mathcal{C}^c}\}$$
$$\leq \mathbb{E}_{\widetilde{D}_s, \widetilde{D}_t}\left[\{(1 - \sigma_t^{(n_s)}) + R_t(\varepsilon_{n_s}, \hat{f}_{n_s})\}\mathbb{1}_\mathcal{C}\right] + \mathbb{E}_{\widetilde{D}_s, \widetilde{D}_t}(\mathbb{1}_{\mathcal{C}^c})$$

$$\leq \left(1 - \sigma_t^{(n_s)}\right) + \mathbb{E}_{\widetilde{D}_s}\left\{R_t(\varepsilon_{n_s}, \hat{f}_{n_s})\right\} + \mathbb{P}(\mathcal{C}^c)$$

$$\lesssim \left(1 - \sigma_t^{(n_s)}\right) + \varepsilon^{-1}\mathbb{E}_{\widetilde{D}_s}\left[\left\{\mathcal{L}(\hat{f}_{n_s}) + \epsilon_1 + \epsilon_2\right\}^{\frac{1}{2}}\right] + \mathbb{P}(\mathcal{C}^c)$$

$$\leq \left(1 - \sigma_t^{(n_s)}\right) + \varepsilon^{-1}\left[\mathbb{E}_{\widetilde{D}_s}\left\{\mathcal{L}(\hat{f}_{n_s})\right\} + \epsilon_1 + \epsilon_2\right]^{\frac{1}{2}} + \mathbb{P}(\mathcal{C}^c) \qquad (47)$$

Since we have known the sample complexity of each terms except for $\mathbb{P}(\mathcal{C}^c)$, the remaining question is to estimate $\mathbb{P}(\mathcal{C}^c)$. To this end, first recall $\psi(\sigma_t^{(n_s)}, \delta_t^{(n_s)}, \varepsilon_{n_s}, \hat{f}_{n_s}) = \Gamma_{\min}(\sigma_t^{(n_s)}, \delta_t^{(n_s)}, \varepsilon_{n_s}, \hat{f}_{n_s}) - \sqrt{2 - 2\Gamma_{\min}(\sigma_t^{(n_s)}, \delta_t^{(n_s)}, \varepsilon_{n_s}, \hat{f}_{n_s})} - \frac{1}{2}\left(1 - \frac{\min_{k \in [K]}\|\hat{\mu}_t(k)\|_2^2}{B_2^2}\right) - \frac{2\max_{k \in [K]}\|\hat{\mu}_t(k) - \mu_t(k)\|_2}{B_2}$. Notice that (34) and dominated convergence theorem imply $R_t(\varepsilon_{n_s}, \hat{f}_{n_s}) \to 0$ a.s., thus

$$\Gamma_{\min}(\sigma_t^{(n_s)}, \delta_t^{(n_s)}, \varepsilon_{n_s}, \hat{f}_{n_s}) = \left(\sigma_t^{(n_s)} - \frac{R_t(\varepsilon_{n_s}, \hat{f}_{n_s})}{\min_i p_t(i)}\right)\left(1 + \left(\frac{B_1}{B_2}\right)^2 - \frac{\mathcal{K}\delta_t^{(n_s)}}{B_2} - \frac{2\varepsilon_{n_s}}{B_2}\right) - 1$$

$$\to \left(\frac{B_1}{B_2}\right)^2.$$

Combining with the fact that $\frac{1 - \min_{k \in [K]}\|\hat{\mu}_t(k)\|_2^2/B_2^2}{2} < \frac{1}{2}$ can yield

$$\Gamma_{\min}(\sigma_t^{(n_s)}, \delta_t^{(n_s)}, \varepsilon_{n_s}, \hat{f}_{n_s}) - \sqrt{2 - 2\Gamma_{\min}(\sigma_t^{(n_s)}, \delta_t^{(n_s)}, \varepsilon_{n_s}, \hat{f}_{n_s})} - \frac{1}{2}\left(1 - \min_{k \in [K]}\|\hat{\mu}_t(k)\|_2^2/B_2^2\right)$$

$$> \frac{1}{2},$$

if $B_1$ is sufficiently close to $B_2$. On the other hand, by Multidimensional Chebyshev's inequality, we yield

$$\mathbb{P}_t\left(\left\|\hat{\mu}_t(k) - \mu_t(k)\right\|_2 \geq \frac{B_2}{8}\right) \leq \frac{64\sqrt{\mathbb{E}_{\boldsymbol{x}_t \in \widetilde{C}_t(k)}\mathbb{E}_{\mathbf{x}_t \in \mathcal{A}(\boldsymbol{x}_t)}\left\|f(\mathbf{x}_t) - \mu_t(k)\right\|_2^2}}{B_2^2\sqrt{2n_t(k)}} \leq \frac{128}{B_2\sqrt{n_t(k)}},$$

which implies that $\psi(\sigma_t^{(n_s)}, \delta_t^{(n_s)}, \varepsilon_{n_s}, \hat{f}_{n_s}) \geq \frac{1}{4}$ with probability at least $1 - \frac{128K}{B_2\sqrt{\min_k n_t(k)}}$ when $n_s$ is sufficiently large. Therefore, with probability at least $1 - \mathcal{O}\left(\frac{1}{\min_k \sqrt{n_t(k)}}\right)$, we have $\mathcal{C}^c \subseteq \left\{\max_{i \neq j}\left|\mu_t(i)^\top\mu_t(j)\right| \geq \frac{B_2^2}{8}\right\}$

$$\mathbb{P}(\mathcal{C}^c) = \mathbb{P}_s\left(\mathcal{C}^c\Big|\mathcal{C}^c \subseteq \left\{\max_{i \neq j}\left|\mu_t(i)^\top\mu_t(j)\right| \geq \frac{B_2^2}{8}\right\}\right) \cdot \mathbb{P}_t\left(\mathcal{C}^c \subseteq \left\{\max_{i \neq j}\left|\mu_t(i)^\top\mu_t(j)\right| \geq \frac{B_2^2}{8}\right\}\right)$$

$$+ \mathbb{P}_s\left(\mathcal{C}^c\Big|\mathcal{C}^c \nsubseteq \left\{\max_{i \neq j}\left|\mu_t(i)^\top\mu_t(j)\right| \geq \frac{B_2^2}{8}\right\}\right) \cdot \mathbb{P}_t\left(\mathcal{C}^c \nsubseteq \left\{\max_{i \neq j}\left|\mu_t(i)^\top\mu_t(j)\right| \geq \frac{B_2^2}{8}\right\}\right)$$

$$\leq \mathbb{P}_s\left(\max_{i \neq j}\left|\mu_t(i)^\top\mu_t(j)\right| \geq \frac{B_2^2}{8}\Big|\mathcal{C}^c \subseteq \left\{\max_{i \neq j}\left|\mu_t(i)^\top\mu_t(j)\right| \geq \frac{B_2^2}{8}\right\}\right)$$

$$+ \mathbb{P}_t\left(\mathcal{C}^c \nsubseteq \left\{\max_{i \neq j}\left|\mu_t(i)^\top\mu_t(j)\right| \geq \frac{B_2^2}{8}\right\}\right)$$

$$\leq \mathbb{P}_s\left(\max_{i \neq j}\left|\mu_t(i)^\top\mu_t(j)\right| \geq \frac{B_2^2}{8}\right)\Big/\mathbb{P}_t\left(\mathcal{C}^c \subseteq \left\{\max_{i \neq j}\left|\mu_t(i)^\top\mu_t(j)\right| \geq \frac{B_2^2}{8}\right\}\right)$$

$$+ \mathbb{P}_t\left(\mathcal{C}^c \nsubseteq \left\{\max_{i \neq j}\left|\mu_t(i)^\top\mu_t(j)\right| \geq \frac{B_2^2}{8}\right\}\right)$$

$$\leq \frac{\mathbb{P}_s\left(\max_{i \neq j}\left|\mu_t(i)^\top\mu_t(j)\right| \geq \frac{B_2^2}{8}\right)}{1 - \mathcal{O}\left(1/\min_k \sqrt{n_t(k)}\right)} + \mathbb{P}_t\left(\mathcal{C}^c \nsubseteq \left\{\max_{i \neq j}\left|\mu_t(i)^\top\mu_t(j)\right| \geq \frac{B_2^2}{8}\right\}\right)$$

$$\lesssim \mathbb{P}_s\left(\max_{i \neq j}\left|\mu_t(i)^\top\mu_t(j)\right| \geq \frac{B_2^2}{8}\right) + \mathbb{P}_t\left(\mathcal{C}^c \nsubseteq \left\{\max_{i \neq j}\left|\mu_t(i)^\top\mu_t(j)\right| \geq \frac{B_2^2}{8}\right\}\right)$$

$$\lesssim (1 - \sigma_s^{(n_s)}) + n_s^{-\frac{\min\{\alpha, 2\epsilon_{\mathcal{A}}\}}{16(\alpha+d+1)}} + \frac{1}{\min_k \sqrt{n_t(k)}}.$$

wherein, as for the term $\mathbb{P}_s\Big( \max_{i \neq j} \big| \mu_t(i)^\top \mu_t(j) \big| \geq \frac{B_2^2}{8} \Big)$ in the last inequality, applying Markov inequality obtains

$$P_s \Big( \max_{i \neq j} \big| \mu_t(i)^\top \mu_t(j) \big| \geq \frac{B_2^2}{8} \Big) \lesssim \mathbb{E}_{\widetilde{D}_s} \Big[ \max_{i \neq j} \big| \mu_t(i)^\top \mu_t(j) \big| \Big]$$

$$\lesssim \sqrt{\mathbb{E}_{\widetilde{D}_s}\{\mathcal{L}(\hat{f}_{n_s})\} + \mathbb{E}_{\widetilde{D}_s}\{\varphi(\sigma_s^{(n_s)}, \delta_s^{(n_s)}, \varepsilon_{n_s}, \hat{f}_{n_s})\}} + \mathcal{K}\epsilon_1,$$

where the sample complexities of both $\mathbb{E}_{\widetilde{D}_s}\{\mathcal{L}(\hat{f}_{n_s})\}$ and $\epsilon_1$ on the right-hand side has been thoroughly explored. Therefore, the final step is to investigate the sample complexity of $\mathbb{E}_{\widetilde{D}_s}\{\varphi(\sigma_s^{(n_s)}, \delta_s^{(n_s)}, \varepsilon_{n_s}, \hat{f}_{n_s})\}$. In fact,

$$\mathbb{E}_{\widetilde{D}_s}\{\varphi(\sigma_s^{(n_s)}, \delta_s^{(n_s)}, \varepsilon_{(n_s)}, R_s(\varepsilon, \hat{f}_{n_s}))\} \lesssim \Big(1 - \sigma_s^{(n_s)} + \mathcal{K}\delta_s^{(n_s)} + 2\varepsilon_{n_s}\Big)^2 + \frac{1}{\varepsilon_{n_s}}\sqrt{\mathbb{E}_{\widetilde{D}_s}\{\mathcal{L}(\hat{f}_{n_s})\}}$$

$$\Big(3 - 2\sigma_s^{(n_s)} + \mathcal{K}\delta_s^{(n_s)} + 2\varepsilon_{n_s}\Big) + \frac{1}{\varepsilon_{n_s}^2}\mathbb{E}_{\widetilde{D}_s}\{\mathcal{L}(\hat{f}_{n_s})\} + (1 - \sigma_s^{(n_s)}) + \Big(\varepsilon_{n_s}^2 + \frac{1}{\varepsilon_{n_s}}\sqrt{\mathbb{E}_{\widetilde{D}_s}\{\mathcal{L}(\hat{f}_{n_s})\}}\Big)^{\frac{1}{2}}$$

$$\leq \Big(1 - \sigma_s^{(n_s)} + \mathcal{K}\delta_s^{(n_s)} + 2\varepsilon_{n_s}\Big) + \frac{2}{\varepsilon_{n_s}}\sqrt{\mathbb{E}_{\widetilde{D}_s}\{\mathcal{L}(\hat{f}_{n_s})\}} + \frac{1}{\varepsilon_{n_s}^2}\mathbb{E}_{\widetilde{D}_s}\{\mathcal{L}(\hat{f}_{n_s})\}$$

$$+ (1 - \sigma_s^{(n_s)} + \mathcal{K}\delta_s^{(n_s)} + 2\varepsilon_{n_s}) + \Big(\varepsilon_{n_s}^2 + \frac{1}{\varepsilon_{n_s}}\sqrt{\mathbb{E}_{\widetilde{D}_s}\{\mathcal{L}(\hat{f}_{n_s})\}}\Big)^{\frac{1}{2}}$$

$$\lesssim \Big(1 - \sigma_s^{(n_s)}\Big) + n_s^{-\frac{\min\{\alpha, \epsilon_{\mathcal{A}}, \epsilon_{\mathrm{ds}}\}}{16(\alpha+d+1)}}.$$

Substituting this conclusion back to (47) yields our conclusion, that is

$$\mathbb{E}_{\widetilde{D}_s, \widetilde{D}_t}\{\mathrm{Err}(Q_{\hat{f}_{n_s}})\} \lesssim \Big(1 - \sigma_s^{(n_s)}\Big) + n_s^{-\frac{\min\{\alpha, \epsilon_{\mathcal{A}}, \epsilon_{\mathrm{ds}}\}}{32(\alpha+d+1)}} + \frac{1}{\min_k \sqrt{n_t(k)}}.$$

$\square$

# G    Auxiliary Lemmas

## G.1    $\mathcal{K}$-Lipschitz property of $\mathcal{NN}_{d_1,d_2}(W, L, \mathcal{K}, B_1, B_2)$

*Proof.* To demonstrate that any function $\phi \in \mathcal{NN}_{d_1,d_2}(W, L, \mathcal{K}, B_1, B_2)$ is a $\mathcal{K}$-Lipschitz function, we first define two special classes. The first class is given by

$$\mathcal{NN}_{d_1,d_2}(W, L, \mathcal{K}) := \Big\{ \phi(\boldsymbol{x}) = A_L \sigma\big(A_{L-1}\sigma(\cdots \sigma(A_0 \boldsymbol{x}))\big) : \kappa(\boldsymbol{\theta}) \leq \mathcal{K} \Big\}, \qquad (48)$$

which is equivalent to $\mathcal{NN}_{d_1,d_2}(W, L, \mathcal{K}, B_1, B_2)$ when ignoring the condition $\|\phi\|_2 \in [B_1, B_2]$. The second class is defined as

$$\mathcal{SNN}_{d_1,d_2}(W, L, \mathcal{K}) := \{\tilde{\phi}(\boldsymbol{x}) = \tilde{A}_L \sigma(\tilde{A}_{L-1}\sigma(\cdots \sigma(\tilde{A}_0 \tilde{\boldsymbol{x}})) : \prod_{l=1}^{L} \|\tilde{A}_l\|_\infty \leq \mathcal{K}\}, \quad \tilde{\boldsymbol{x}} := \begin{pmatrix} \boldsymbol{x} \\ 1 \end{pmatrix},$$

where $\tilde{A}_l \in \mathbb{R}^{N_{l+1} \times N_l}$ with $N_0 = d_1 + 1$.

It is clear that $\mathcal{NN}_{d_1,d_2}(W, L, \mathcal{K}, B_1, B_2) \subseteq \mathcal{NN}_{d_1,d_2}(W, L, \mathcal{K})$, and every element in $\mathcal{SNN}_{d_1,d_2}(W, L, \mathcal{K})$ is a $\mathcal{K}$-Lipschitz function due to the 1-Lipschitz property of the ReLU activation function. Thus, it suffices to show that

$$\mathcal{SNN}_{d_1,d_2}(W, L, \mathcal{K}) \subseteq \mathcal{NN}_{d_1,d_2}(W, L, \mathcal{K}) \subseteq \mathcal{SNN}_{d_1,d_2}(W + 1, L, \mathcal{K})$$

to establish our claim.

To begin, any function $\phi(\boldsymbol{x}) = A_L \sigma\big(A_{L-1}\sigma(\cdots\sigma\big(A_0\boldsymbol{x} + \boldsymbol{b}_0\big)) + \boldsymbol{b}_{L-1}\big) \in \mathcal{NN}_{d_1,d_2}(W, L, \mathcal{K})$ can be restructured as $\tilde{\phi}(\boldsymbol{x}) = \tilde{A}_L\sigma(\tilde{A}_{L-1}\sigma(\cdots\sigma(\tilde{A}_0\tilde{\boldsymbol{x}})))$, where

$$\tilde{\boldsymbol{x}} := \begin{pmatrix} \boldsymbol{x} \\ 1 \end{pmatrix}, \quad \tilde{A} = \begin{pmatrix} A_L, \mathbf{0} \end{pmatrix}, \quad \tilde{A}_l = \begin{pmatrix} A_l & \boldsymbol{b}_l \\ \mathbf{0} & 1 \end{pmatrix}, \quad l = 0, \dots, L-1.$$

Notably, we have $\prod_{l=0}^{L} \|\tilde{A}_l\|_\infty = \|A_L\|_\infty \prod_{l=0}^{L-1} \max\{\|(A_l, \boldsymbol{b}_l)\|_\infty, 1\} = \kappa(\boldsymbol{\theta}) \le \mathcal{K}$, which implies that $\phi \in \mathcal{SNN}_{d_1,d_2}(W+1, L, \mathcal{K})$.

Conversely, since any $\tilde{\phi} \in \mathcal{SNN}(W, L, \mathcal{K})$ can also be parameterized as $A_L\sigma\big(A_{L-1}\sigma(\cdots\sigma\big(A_0\boldsymbol{x} + \boldsymbol{b}_0\big)) + \boldsymbol{b}_{L-1}\big)$ with $\boldsymbol{\theta} = (\tilde{A}_0, (\tilde{A}_1, \mathbf{0}), \dots, (\tilde{A}_{L-1}, \mathbf{0}), \tilde{A}_L)$, we can use the absolute homogeneity of the ReLU function to rescale $\tilde{A}_l$ such that $\|\tilde{A}_L\|_\infty \le \mathcal{K}$ and $\|\tilde{A}_l\|_\infty = 1$ for $l \ne L$. Consequently, we have $\kappa(\boldsymbol{\theta}) = \prod_{l=0}^{L} \|\tilde{A}_l\|_\infty \le \mathcal{K}$, which yields $\tilde{\phi} \in \mathcal{NN}(W, L, \mathcal{K})$. This completes the proof. $\square$

### G.2  Lipschitz property of $\ell$

**Proposition 2.** $\ell$ is a Lipschitz function on the domain $\{\boldsymbol{x} \in \mathbb{R}^{2d^*} : \|\boldsymbol{x}\|_2 \le \sqrt{2}B_2\} \times \mathcal{G}_1$.

*Proof.* We begin by proving $\|\ell(\cdot, G)\|_{\mathrm{Lip}} < \infty$ for any fixed $G \in \mathcal{G}_1$. Let $\boldsymbol{x} = (\boldsymbol{x}_1, \boldsymbol{x}_2) \in \mathbb{R}^{2d^*}$, where $\boldsymbol{x}_1, \boldsymbol{x}_2 \in \mathbb{R}^{d^*}$. We first demonstrate that $g(\boldsymbol{x}) = \|\boldsymbol{x}_1 - \boldsymbol{x}_2\|_2^2$ is a Lipschitz function. To this end, let $p(\boldsymbol{x}) := \boldsymbol{x}_1 - \boldsymbol{x}_2$, then we have:

$$\begin{aligned}
\|p(\boldsymbol{x}) - p(\boldsymbol{y})\|_2^2 &= \|\boldsymbol{x}_1 - \boldsymbol{x}_2 - \boldsymbol{y}_1 + \boldsymbol{y}_2\|_2^2 \le (\|\boldsymbol{x}_1 - \boldsymbol{y}_1\|_2 + \|\boldsymbol{x}_2 - \boldsymbol{y}_2\|_2)^2 \\
&= \|\boldsymbol{x}_1 - \boldsymbol{y}_1\|_2^2 + \|\boldsymbol{x}_2 - \boldsymbol{y}_2\|_2^2 + 2\|\boldsymbol{x}_1 - \boldsymbol{y}_1\|_2 \|\boldsymbol{x}_2 - \boldsymbol{y}_2\|_2 \\
&\le 2\left(\|\boldsymbol{x}_1 - \boldsymbol{y}_1\|_2^2 + \|\boldsymbol{x}_2 - \boldsymbol{y}_2\|_2^2\right) = 2\|\boldsymbol{x} - \boldsymbol{y}\|_2^2,
\end{aligned}$$

which implies that $p(\boldsymbol{x}) \in \mathrm{Lip}(\sqrt{2})$. Moreover, it is easy to notice that $p$ satisfies $\|p(\boldsymbol{x})\|_2 = \|\boldsymbol{x}_1 - \boldsymbol{x}_2\|_2 \le \|\boldsymbol{x}_1\|_2 + \|\boldsymbol{x}_2\|_2 \le 2\|\boldsymbol{x}\|_2 \le 2\sqrt{2}B_2$. on the other hand, let $q(\boldsymbol{y}) := \|\boldsymbol{y}\|_2^2$. We have:

$$\left\|\frac{\partial q}{\partial \boldsymbol{y}}\big(p(\boldsymbol{x})\big)\right\|_2 = 2\|p(\boldsymbol{x})\|_2 \le 4\sqrt{2}B_2.$$

Combining these facts together, we know $g(\boldsymbol{x}) = q(p(\boldsymbol{x})) = \|\boldsymbol{x}_1 - \boldsymbol{x}_2\|_2^2 \in \mathrm{Lip}(8B_2)$. Now, we show that $h(\boldsymbol{x}) = \langle \boldsymbol{x}_1\boldsymbol{x}_2^\top - I_{d^*}, G\rangle_F$ is also a Lipschitz function. Define $r(\boldsymbol{x}) := \boldsymbol{x}_1\boldsymbol{x}_2^\top$. We have:

$$\begin{aligned}
\|r(\boldsymbol{x}) - r(\boldsymbol{y})\|_F &= \|\boldsymbol{x}_1\boldsymbol{x}_2^\top - \boldsymbol{y}_1\boldsymbol{y}_2^\top\|_F = \|\boldsymbol{x}_1\boldsymbol{x}_2^\top - \boldsymbol{x}_1\boldsymbol{y}_2^\top + \boldsymbol{x}_1\boldsymbol{y}_2^\top - \boldsymbol{y}_1\boldsymbol{y}_2^\top\|_F \\
&= \|\boldsymbol{x}_1(\boldsymbol{x}_2 - \boldsymbol{y}_2)^\top + (\boldsymbol{x}_1 - \boldsymbol{y}_1)\boldsymbol{y}_2^\top\|_F \le \|\boldsymbol{x}_1\|_F\|\boldsymbol{x}_2 - \boldsymbol{y}_2\|_F + \|\boldsymbol{x}_1 - \boldsymbol{y}_1\|_F\|\boldsymbol{y}_2\|_F \\
&\le (\|\boldsymbol{x}_1\|_2 + \|\boldsymbol{y}_2\|_2)\|\boldsymbol{x} - \boldsymbol{y}\|_2 \le 2\sqrt{2}B_2\|\boldsymbol{x} - \boldsymbol{y}\|_2.
\end{aligned}$$

Additionally, define $t(A) := \langle A - I_{d^*}, G\rangle_F$. It is obvious that $\|\nabla t(A)\|_F = \|G\|_F \le B_2^2 + \sqrt{d^*}$. Based on these, we can conclude $h(\boldsymbol{x}) = t(r(\boldsymbol{x})) \in \mathrm{Lip}(2\sqrt{2}B_2(B_2^2 + \sqrt{d^*}))$. By combining above results, we yield for any $G \in \mathcal{G}_1$, $\|\ell(\cdot, G)\|_{\mathrm{Lip}} < \infty$ on the domain $\{\boldsymbol{x} : \|\boldsymbol{x}\|_2 \le \sqrt{2}B_2\}$. Furthermore, for a fixed $\boldsymbol{x} \in \mathbb{R}^{2d^*}$ such that $\|\boldsymbol{x}\|_2 \le \sqrt{2}B_2$, we have

$$|\ell(\boldsymbol{x}, G_1) - \ell(\boldsymbol{x}, G_2)| = |\langle \boldsymbol{x}, G_1 - G_2\rangle_F| \le \|\boldsymbol{x}\|_2\|G_1 - G_2\|_F = \sqrt{2}B_2\|G_1 - G_2\|_F,$$

which implies that $\ell(\boldsymbol{x}, \cdot) \in \mathrm{Lip}(\sqrt{2}B_2)$. Finally, we can conclude

$$\begin{aligned}
\big|\ell(\boldsymbol{x}_1, G_1) - \ell(\boldsymbol{x}_2, G_2)\big|^2 &\le \big\{|\ell(\boldsymbol{x}_1, G_1) - \ell(\boldsymbol{x}_2, G_1)| + |\ell(\boldsymbol{x}_2, G_1) - \ell(\boldsymbol{x}_2, G_2)|\big\}^2 \\
&\le \Big[\{\sqrt{2} + 2\sqrt{2}B_2(B_2^2 + \sqrt{d^*})\}\big\|\boldsymbol{x}_1 - \boldsymbol{x}_2\big\|_2 + \sqrt{2}B_2\big\|G_1 - G_2\big\|_F\Big]^2 \\
&\le 2\Big\{\sqrt{2} + 2\sqrt{2}B_2(B_2^2 + \sqrt{d^*})\Big\}^2\big\|\boldsymbol{x}_1 - \boldsymbol{x}_2\big\|_2^2 + 4B_2^2\big\|G_1 - G_2\big\|_F^2 \\
&\le C\big\|\mathrm{vec}(\boldsymbol{x}_1, G_1) - \mathrm{vec}(\boldsymbol{x}_2, G_2)\big\|_2^2,
\end{aligned}$$

where $C$ is a constant such that $C \ge \max\Big\{2(\sqrt{2} + 2\sqrt{2}B_2(B_2^2 + \sqrt{d^*}))^2, 4B_2^2\Big\}$ and $\mathrm{vec}(\cdot)$ represents vectorized operator. $\square$

