# OpenReview forum: "Adv-SSL: Adversarial Self-Supervised Representation Learning with Theoretical Guarantees"
_NeurIPS.cc/2025/Conference — NeurIPS 2025 poster_

### Official Review · Reviewer_sjVz · 2025-06-29

**Clarity:** 3
**Significance:** 2
**Originality:** 2
**Rating:** 3
**Confidence:** 2

**Summary:**

This paper introduced a new self-supervised learning approach named Adversarial Self-Supervised Representation Learning (ASSRL) for unbiased transfer learning. Through an innovative iteration format, ASSRL eliminates the bias between the population risk and its sample-level counterpart. In this paper, the end-to-end theoretical guarantee for the downstream classification task performance is provided. This theoretical understanding also explains the effectiveness of few-shot learning for downstream tasks with small sample sizes. Experiments are presented in this paper to show the effectiveness of the ASSRL method.

**Questions:**

1. For the question the authors raised in line 187, do the authors aim to answer this question just by the empirical experiments in Table 1? Or can this question also be answered from the theoretical perspective? Please elaborate on this question in more detail.

2. For the experimental results, why did the authors select some of the existing methods from [14] for comparison? Any reason for choosing only these self-supervised learning methods for comparison?

3. Why is the comparison not made with more state-of-the-art self-supervised learning methods? For example, what about those self-supervised learning methods that are mentioned in the introduction of this paper?

**Ethical Concerns:**

["NO or VERY MINOR ethics concerns only"]

**Final Justification:**

I tend to keep my original score. As discussed in the reviews and rebuttals, some important insights and clarification (such as those regarding the theorems and assumptions) should be included in the revised version, while maintaining a complete structure of the paper. I also noticed that other reviewers also raised some concerns about the experiments. Therefore, I sincerely hope that the authors can address these points in the revised version of this paper.

**Limitations:**

The limitations are discussed in the appendix of this paper, mainly about the minimax rates and the lower bound for the ASSRL method. Other limitations see  the question part.

**Paper Formatting Concerns:**

1. The structure of the paper may be incomplete. The authors could at least include a conclusion in the main paper to provide a complete structure of the paper.

2. More details about the insights and remarks could be given in the main text, for example, the experiments in Section 4. In the current version, it seems that some important information is missing.

**Quality:**

2

**Strengths And Weaknesses:**

Strengths:
1. This paper proposed a new adversarial contrastive learning method for unsupervised transfer learning.
2. End-to-end theoretical analysis for the downstream classification task is given in this paper.
3. Experimental results are given to show the improved accuracy for this new ASSRL method.

Weaknesses:
1. The experimental results comparison only covers a few existing self-supervised learning methods.

---

> ### Author Rebuttal · Authors · 2025-07-31
>
> We thank you for your thorough review of our manuscript and for your constructive suggestions. We have revised our manuscript by incorporating all of your suggestions. The revised version is much strengthened. Our point-by-point responses to your comments are given below.
>
> > **Comment**
> > * The experimental results comparison only covers a few existing self-supervised learning methods.
> > * For the experimental results, why did the authors select some of the existing methods from [14] for comparison? Any reason for choosing only these self-supervised learning methods for comparison?
> > * Why is the comparison not made with more state-of-the-art self-supervised learning methods? For example, what about those self-supervised learning methods that are mentioned in the introduction of this paper?
>
> **Response**
> Thank you for your valuable suggestions. We have add the comparison with [a] VICReg: Variance-Invariance-Covariance Regularization for Self-Supervised Learning, ICLR 2022. [b] Geometric View of Soft Decorrelation in Self-Supervised Learning, KDD 2024. The preliminarily results are shown below
>
>
> | Method | CIFAR-10 | CIFAR-10 | CIFAR-100 | CIFAR-100 |  Tiny ImageNet | Tiny ImageNet |
> | ---- | ----- | ----- | ----- | ----- | ---- | ----- |
> |  | Linear | $k$-nn | Linear | $k$-nn | Linear | $k$-nn |
> | VICReg | 91.23 | 89.15 | 67.61 | 57.04 | 48.55 | 35.62 |
> | LogDet | 92.47 | 90.19 | 67.32 | 57.56 | 49.13 | 35.78 |
> | ASSRL | **93.01** | **90.97** | **68.94** |  **58.50** | **50.21** | **37.40** |
>
> > **Comment**
> For the question the authors raised in line 187, do the authors aim to answer this question just by the empirical experiments in Table 1? Or can this question also be answered from the theoretical perspective? Please elaborate on this question in more detail.
>
> **Response**
> Exactly. The question you raised is undeniably important and extremely valuable. We posit that the improvements can be explained from an unbiased perspective, as discussed in the introduction. Roughly speaking, the original iteration method is driven by a biased sampling-level risk, while our revision leads to an unbiased sampling-level risk. This unbiasedness results in better performance of our methods.
>
> > **Comment** The structure of the paper may be incomplete. The authors could at least include a conclusion in the main paper to provide a complete structure of the paper.
>
> **Response** Thank you for your suggestions. I will add a conclusion in the main paper to provide a complete structure of the paper.
>
> > **Comment** More details about the insights and remarks could be given in the main text, for example, the experiments in Section 4. In the current version, it seems that some important information is missing.
>
> **Response** Thank you for your suggestions. Due to the limitation of the space, we move some remarks and illustrations to the appendices. For example, we provide explanations and toy examples of the assumptions in Appendix F, and the experimental details are offered in Appendix E. Some insightful ablation experiments are deferred to Appendix E. In our revised version, we reorganize our manuscript and move some important insights and remarks to our main text.
>
>
> Thank you once again for your careful reading, professional suggestions, and inspiring recognitions. If you have any questions, please feel free to contact us.

---

> > ### Comment · Reviewer_sjVz · 2025-08-05
> >
> > Thank you for your response and the general clarification. As discussed in the reviews and rebuttals, some important insights and clarification (such as those regarding the theorems and assumptions) should be included in the revised version, while maintaining a complete structure of the paper. I also noticed that other reviewers raised some concerns about the experiments (Reviewer 41Yc and Reviewer 5zMo). Therefore, I sincerely hope that the authors can address these points in the revised version of this paper.

---

> > > ### Author Response · Authors · 2025-08-07
> > > **Rebuttal by Authors**
> > >
> > > Thank you for your invaluable suggestions. We highly value your professional opinion and promise to incorporate them into our revised manuscript. However, NeurIPS 2025 does not allow authors to upload our revised manuscripts at this stage. Thank you for your understanding.

---

### Official Review · Reviewer_5zMo · 2025-07-03

**Clarity:** 3
**Significance:** 3
**Originality:** 4
**Rating:** 3
**Confidence:** 4

**Summary:**

This paper identifies that many self-supervised learning methods suffer from a biased sample risk, which causes optimization instability and prevents theoretical analysis. To solve this, the authors propose Adversarial Self-Supervised Representation Learning (ASSRL), a novel method that reframes the objective as a minimax problem to create an unbiased estimator. This adversarial approach leads to significantly improved classification accuracy, outperforming both the biased methods it aims to correct and other mainstream techniques on several benchmarks.

**Questions:**

Please see the comments above.

**Ethical Concerns:**

["NO or VERY MINOR ethics concerns only"]

**Limitations:**

yes

**Paper Formatting Concerns:**

No major formatting issue observed

**Quality:**

3

**Strengths And Weaknesses:**

Strengths
-	The paper clearly identifies a significant and practical problem in existing SSL methods—the bias in the sample-level risk approximation. The proposed solution, reformulating the objective as a minimax problem to create an unbiased estimator, is elegant and directly addresses this core issue.
-	A major strength of this work is the end-to-end theoretical analysis. Unlike prior work that often focused only on population-level risk or generalization error without considering approximation error, this paper provides a complete convergence analysis. The theory formally connects the number of unlabeled source samples to downstream task performance, which is a crucial aspect of transfer learning.
-	The theoretical results offer a formal explanation for the success of SSL in few-shot learning contexts. The paper proves that with a sufficient amount of unlabeled data, ASSRL can learn representations that are well-clustered by category, thereby enabling strong classification with very few labeled samples. This bridges an important gap between the empirical success and theoretical understanding of few-shot learning.

Weaknesses
-	The theoretical guarantees rely on a number of assumptions (Section 3.3), particularly regarding data augmentations (Assumption 4) and the shift between source and target domains (Assumption 5). These assumptions, such as the existence of an augmentation sequence where distances between similar samples converge to zero, might be difficult to verify or guarantee in practice. Could the authors provide more intuition on why these assumptions are reasonable for standard computer vision datasets and augmentation pipelines?
-	The paper introduces an alternating optimization algorithm (Algorithm 1) but does not analyze its practical implications in detail. There is no discussion of the computational overhead of the adversarial updates compared to the simpler direct minimization in methods like Barlow Twins. The ablation study on the regularization parameter λ shows that its optimal value varies significantly across datasets (e.g., 5.0×10−2 for CIFAR-10 vs. 1.0×10−2 for CIFAR-100). This suggests the method may be sensitive to this choice, requiring expensive tuning for new applications. A more in-depth analysis of this sensitivity would be valuable.
-	The baselines in the key comparison table (Table 1) are re-implementations by the authors. While the authors state that their results are close to those in the LightlySSL library, a direct comparison to the numbers reported in the original papers (e.g., [22, 38]) would provide a stronger and more objective benchmark. This would help ensure that the observed performance gap is entirely due to the methodological improvement of ASSRL and not differences in implementation.
-	The paper's checklist (Question 7) states that the experimental results report the best performance over three runs, rather than providing error bars or other measures of statistical significance. While the reported performance gains are large, this approach makes it difficult to assess the stability and variance of the results. Standard practice would be to report the mean and standard deviation over multiple runs, which would make the empirical claims much more robust and convincing.

---

> ### Author Rebuttal · Authors · 2025-07-31
>
> We thank you for your thorough review of our manuscript and for your constructive suggestions. We have revised our manuscript by incorporating all of your suggestions. The revised version is much strengthened. Our point-by-point responses to your comments are given below.
>
> > **Comment**  The theoretical guarantees rely on a number of assumptions (Section 3.3), particularly regarding data augmentations (Assumption 4) and the shift between source and target domains (Assumption 5). These assumptions, such as the existence of an augmentation sequence where distances between similar samples converge to zero, might be difficult to verify or guarantee in practice. Could the authors provide more intuition on why these assumptions are reasonable for standard computer vision datasets and augmentation pipelines?
>
> **Response** Thank you for your question. As you mentioned, it might be difficult to verify in practice. To provide more intuitions, we have illustrated Assumption 4 in Appendix F.2 with toy examples. Meanwhile, the corresponding explanation of Assumption 5 can be found in Appendix F.3. Thank you for your understanding.
>
> > **Comment**  The paper introduces an alternating optimization algorithm (Algorithm 1) but does not analyze its practical implications in detail. There is no discussion of the computational overhead of the adversarial updates compared to the simpler direct minimization in methods like Barlow Twins.
>
> **Response** We highly value this important question. First, it is clear that the additional cost of adversarial updates is negligible, as the inner maximization problem has an analytical solution, as shown in Step 8 of Algorithm 1. To further support this view, we provide the costs related to timing and memory below.
>
> | Methods | CIFAR-10 | CIFAR-10 | CIFAR-100 | CIFAR-100 | Tiny ImageNet | Tiny ImageNet |
> | ----- | ----- | ------ | ----- | ----- | ----- | ----- |
> | | Memory | Time | Memory | Time | Memory | Time |
> | Barlow Twins | 5598 MiB | 68s | 5598 MiB | 74s | 8307 MiB | 386s |
> | ASSRL | 5585 MiB | 51s | 5585 MiB | 52s | 8282 MiB | 352s |
>
> All experiments were conducted using a single Tesla V100 GPU. The time mentioned refers to the time spent per epoch.
>
> > **Comment** The ablation study on the regularization parameter $\lambda$ shows that its optimal value varies significantly across datasets (e.g., 5.0×10−2 for CIFAR-10 vs. 1.0×10−2 for CIFAR-100). This suggests the method may be sensitive to this choice, requiring expensive tuning for new applications. A more in-depth analysis of this sensitivity would be valuable.
>
> **Response** Thank you for your suggestion. We recognize that tuning costs are inevitable for almost all new methods. Additionally, we must emphasize that the results in Table 4 significantly outperform the biased methods shown in Table 1, regardless of the value of $\lambda$, indicating that the tuning parameters do not impact our experimental advantages. Thank you for your understanding.
>
> > **Comment** The baselines in the key comparison table (Table 1) are re-implementations by the authors. While the authors state that their results are close to those in the LightlySSL library, a direct comparison to the numbers reported in the original papers (e.g., [22, 38]) would provide a stronger and more objective benchmark. This would help ensure that the observed performance gap is entirely due to the methodological improvement of ASSRL and not differences in implementation
>
> **Response** Thank you for your suggestion. However, the original papers did not report corresponding results on the datasets used in our study. To facilitate a comparison, we are actively working to obtain results on ImageNet, but we will need more time to achieve this. Thank you for your understanding.
>
> > **Comment** The paper's checklist (Question 7) states that the experimental results report the best performance over three runs, rather than providing error bars or other measures of statistical significance. While the reported performance gains are large, this approach makes it difficult to assess the stability and variance of the results. Standard practice would be to report the mean and standard deviation over multiple runs, which would make the empirical claims much more robust and convincing.
>
> **Response** Thank you for your valuable suggestion. Our preliminary results are shown as below:
>
> | Method | CIFAR-10 | CIFAR-10 | CIFAR-100 | CIFAR-100 |
> | ----- | ----- | ----- | ----- | ----- |
> | | Mean | Deviation | Mean | deviation |
> | ASSRL | 92.49 | 0.71 | 68.53  | 0.67 |
>
> Thank you once again for your careful reading, professional suggestions, and inspiring recognitions. If you have any questions, please feel free to contact us.

---

### Official Review · Reviewer_41Yc · 2025-07-05

**Clarity:** 4
**Significance:** 3
**Originality:** 4
**Rating:** 5
**Confidence:** 4

**Summary:**

The authors propose an improvement (termed ASSRL) in how self-supervised learning methods regularize their optimization. Specifically, they improve upon the regulization term that encourages alignment of the feature covariance matrix with the identity matrix. Prior work relied on a biased estimate of this term, leading to instability in optimization and preventing some theoretical analysis. The author's proposed method reformulates this term as a minimax optimization problem that can be optimized through an alternating optimization algorithm. This new formulation provides an unbiased estimate. While the authors primarily focuse on the theoretical soundness of their method, they also include empirical results that show their method does in fact enable an improvement over baselines.

**Questions:**

- How much of an impact does the method have on (empirical) training time?
- How well does training scale with image/model size (the empirical evaluations all use relatively small image and model sizes).

**Ethical Concerns:**

["NO or VERY MINOR ethics concerns only"]

**Final Justification:**

I am keeping my rating. Authors addressed my concern about empirical results, but I feel the paper can still be accepted without the additional results.

**Limitations:**

Limitations adequately addressed.

**Paper Formatting Concerns:**

No major issues.

Minor issues:
- Typo in line 155: "used in throughout this" -> "used in this"

**Quality:**

4

**Strengths And Weaknesses:**

STRENGTHS

- Very clearly written.
- Motivations, background context, and logical flow of paper are very easy to follow.
- Theory looks good and is clearly explained. Comprehensiveness of the analysis is a big plus.
- Assumptions used seem reasonable / are similar to assumptions made in prior works. This is important--the theory would be less meaningful and useful if the assumptions were overly restrictive.
- Empirical evaluations show generalization of theory to practice.


WEAKNESSES

- Empirical evaluations are limited in scope. While primary focus is the theory, so I am largely okay with this, evaluations beyond CIFAR and tiny ImageNet would further bolster their results.
- Additionally, empirical results are all positive, but it is unclear how significant this is in terms of actual utility in situations where self-supervised learning is used in practice--CIFAR and tiny ImageNet are primarily used for small scale testing rather than real world use.
- Table 2 compares the author's results with results from a prior paper. I think this is mostly fine, but it does slightly detract from the results, since training conditions may not be identical.

---

> ### Author Rebuttal · Authors · 2025-07-31
>
> We thank you for your thorough review of our manuscript and for your constructive suggestions. We have revised our manuscript by incorporating all of your suggestions. The revised version is much strengthened. Our point-by-point responses to your comments are given below.
>
> > **Comment**
> > * Empirical evaluations are limited in scope. While primary focus is the theory, so I am largely okay with this, evaluations beyond CIFAR and tiny ImageNet would further bolster their results.
> > * Additionally, empirical results are all positive, but it is unclear how significant this is in terms of actual utility in situations where self-supervised learning is used in practice--CIFAR and tiny ImageNet are primarily used for small scale testing rather than real world use.
> > * How well does training scale with image/model size (the empirical evaluations all use relatively small image and model sizes).
>
> **Response**
> Thank you for your valuable suggestion. We are working to obtain experimental results on ImageNet, but we will need more time to do so. Thank you for your understanding.
>
> > **Comment**
> > Table 2 compares the author's results with results from a prior paper. I think this is mostly fine, but it does slightly detract from the results, since training conditions may not be identical.
>
> **Response**
> Thank you for your valuable questions. All implementations totally aligns with the experimental setting in [a] except for $\lambda$. Therefore, the training conditions are identical.
>
> [a] Whitening for Self-Supervised Representation Learning, ICML 2021
>
> > **Comment**
> > How much of an impact does the method have on (empirical) training time?
>
> **Response**
> We highly value this important question. First, it is clear that the additional cost of adversarial updates is negligible, as the inner maximization problem has an analytical solution, as shown in Step 8 of Algorithm 1. To further support this view, we provide the costs related to timing and memory below.
>
> | Methods | CIFAR-10 | CIFAR-10 | CIFAR-100 | CIFAR-100 | Tiny ImageNet | Tiny ImageNet |
> | ----- | ----- | ------ | ----- | ----- | ----- | ----- |
> | | Memory | Time | Memory | Time | Memory | Time |
> | Barlow Twins | 5598 MiB | 68s | 5598 MiB | 74s | 8307 MiB | 386s |
> | ASSRL | 5585 MiB | 51s | 5585 MiB | 52s | 8282 MiB | 352s |
>
> All experiments were conducted using a single Tesla V100 GPU. The time mentioned refers to the time spent per epoch.
>
> Thank you once again for your careful reading, professional suggestions, and inspiring recognitions. If you have any questions, please feel free to contact us.

---

### Official Review · Reviewer_2wB8 · 2025-07-07

**Clarity:** 3
**Significance:** 3
**Originality:** 2
**Rating:** 4
**Confidence:** 2

**Summary:**

The paper studies the regimen where unlabeled data is used to learn a representation that aligns an image with its augmentations with the hope of learning semantic information useful for a downstream supervised task. Of several methods towards this end surveyed in the introduction, they use a regularization approach where the covariance over augmentations is aligned towards the identity matrix (1) and thus encouraging separability. The authors introduce a modification of this regularizer that leads to a minimax learning algorithm they term Adversarial Self-Supervised Representation Learning (ASSRL). This leads way to a natural numerical alternating procedure, Algorithm 1, which they use in experiments to demonstrate their method; These experiments demonstrate the success of their method in comparison to prior work. Finally, Theorem 1 provides a theoretical guarantee for an RERM version of their algorithm for ReLU neural networks for classification.

**Questions:**

**About Augmentations**
* Line 150, the authors say that $m$ could be infinite, yet later we consider a uniform density over all augmentations. Could the authors clarify this?
* In order to approximate some augmentation classes we require a cover that is exponential in dimension. e.g., rotational transformations. Thus, there can be additional dimensional dependence for augmentations commonly used. How does this affect the rate?
*  It seems to me that a covering based argument over continuous augmentation classes would be more direct. Can the authors comment on the difficulty or feasibility of adapting their argument to this setting?

**Theorem 1**
* Is there a reason why a finite-sample bound is used for the source task?
*  Is the  $1/\mathrm{min}_k \sqrt{n_t(k)}$ natural here should we expect something that is more balanced as as we get more data for all classes except for one.


**On approach** (my biggest concern)
* For fixed $f$ and $G$ it is clear that  $\mathcal{R}(f, G)=\mathbb{E}_{\widetilde{D}_s}\{\widehat{\mathcal{R}}(f, G)\}$.

Yet, as I understand, this isn't true for $\mathbb{E}_{\widetilde{D}_s}$$ \sup _{G \in \widehat{\mathcal{G}}(f)} \widehat{\mathcal{R}}(f, G)$.

For example in (4) on line 174, on the left hand side we see taking the maximum over $G$ in $\hat{\mathcal{G}}(f)$ which is equal to a term for which there is no max over $G$. As I understand the latter is ASSRL because it's the adversarial minimax and thus it appears to me to be similarly biased in comparison to prior work. And therefore I'm not sure how Section 3.2 provides a justification for the method.

Am I misunderstanding the method?

**Miscellaneous**
* Are ReLU networks a Holder class? As their partials may not exist everywhere.

**Ethical Concerns:**

["NO or VERY MINOR ethics concerns only"]

**Final Justification:**

My biggest concern was addressed by the authors. The authors making additional remarks along the lines those made to me would be valuable to the reader. The work of unbiased estimator of a representation with end-to-end bounds adds to and clarifies to the literature. To my knowledge the contribution is novel and I raise my score to 4.

**Limitations:**

Repeating a two points mentioned above,  the function in Theorem 1 doesn't apply to the numerical procedure and the "mixed bounds" for Theorem 1 (asymptotic versus finite) .

**Paper Formatting Concerns:**

None noted.

**Quality:**

2

**Strengths And Weaknesses:**

**Strengths**
* The problem is well motivated and the algorithmic setting is valuable in practice.
* The work is extensive in the sense that it has experiments, practical numerical algorithms, a theoretical guarantee, etc.
* To my knowledge the work is original. Their modification of the regularizer is reminiscent of other methods that could be viewed as searching over all one dimensional projections.
* It's a generally well thought out paper. The Appendix has several sections that the reader will find useful, as I did. For example see the sections justifying the assumptions made.

**Weaknesses**
* Theorem 1's guarantee doesn't hold for the result of Algorithm 1. Of course, this is common in the literature as the problem is non-convex.  A note just specifying that Algorithm 1 likely doesn't return the RERM in Theorem 1 may be useful to the reader.
* I was confused by their augmentations discussion (see below)
* To my eye, which doesn't include much familiarity with prior work, I do not find the rate in Theorem 1 very interpretable. I'd appreciate more remarks on interpreting the rate of decay for each term in Theorem 1 perhaps, if possible, an example. Concretely there are with the several $\mathrm{min}$s that make it to tell which setting achieves which argument therein.
* Theorem 1's rate is a mixture of asymptotic and finite bounds. Asymptotic for the source task and finite for the target task.

General Writing notes:
* Although perhaps implicit in context I think the reader would be aided in learning that ReLU neural networks for classification is the model used.

---

> ### Author Rebuttal · Authors · 2025-07-31
>
> We thank you for your thorough review of our manuscript and for your constructive suggestions. We have revised our manuscript by incorporating all of your suggestions. The revised version is much strengthened. Our point-by-point responses to your comments are given below.
>
> > **Comment** Theorem 1's guarantee doesn't hold for the result of Algorithm 1. Of course, this is common in the literature as the problem is non-convex. A note just specifying that Algorithm 1 likely doesn't return the RERM in Theorem 1 may be useful to the reader.
>
> **Response** Thank you for your professional comment and understanding. As you mentioned, the algorithm does not return the RERM involved in Theorem 1. We have emphasized this difference in the revised manuscript.
>
> > **Comment** To my eye, which doesn't include much familiarity with prior work, I do not find the rate in Theorem 1 very interpretable. I'd appreciate more remarks on interpreting the rate of decay for each term in Theorem 1 perhaps, if possible, an example.
>
> **Response** Thank you for your invaluable suggestion. We found a typo regarding the term $\epsilon_{n_s} \asymp n_s^{-\frac{\min\\{\alpha, \nu, \varsigma, \epsilon_\mathcal{A}\\}}{8(\alpha + d + 1)}}$ in our theorem, it should be $\epsilon_{n_s} \asymp n_s^{-\frac{\min\\{\alpha, \epsilon_\mathrm{ds}, \epsilon_\mathcal{A}\\}}{8(\alpha + d + 1)}}$. we have revised our manuscript according to your advice. The rate respected to $n_s$ is $-\frac{\min\\{\alpha, \epsilon_\mathcal{A}, \epsilon_\mathrm{ds}\\}}{32(\alpha + d + 1)}$. We will interpret it term by term.
>
> * First, as the dimensionality of the data \( d \) increases, the convergence rate of ASSRL with respect to the sample size $n_s$ slows down.
> * The term $\epsilon_{\mathcal{A}}$ represents the quality of the data augmentation used. As $\epsilon_{\mathcal{A}}$ increases, it indicates a higher quality of data augmentation, which in turn leads to a faster convergence rate of the upper bound on the misclassification rate with respect to $n_s$.
> * The term $\epsilon_{\mathrm{ds}}$ indicates the extent of the shift between the source domain and the target domain. As it increases, the degree of shifting decreases, meaning that the difference between the source and target domains becomes smaller. This makes the transfer learning task significantly easier. Consequently, the convergence rate with respect to $n_s$ will increase as $\epsilon_{\mathrm{ds}}$ rises.
> * As for the term $\alpha$, Consider the case that both $\epsilon_{\mathcal{A}}$ and $\epsilon_{\mathrm{ds}}$ are larger than $\alpha$, In this scenario, the convergence rate takes on a typical form found in nonparametric statistics: $-\frac{\alpha}{32(\alpha + d + 1)}$.
>
> > **Comment** Concretely there are with the several $\min$s that make it to tell which setting achieves which argument therein.
>
> **Response** Thanks for your constructive suggestions. Let's consider a special case that $\epsilon_{\mathrm{ds}} > \alpha$ and $\epsilon_{\mathcal{A}} > \alpha$. This indicates that the distribution shift is minimal but significant enough to consider, while the augmentations used are sufficiently strong. In this context, the rate involved in our theorem degenerates to $n_s^{-\frac{\alpha}{\alpha + d + 1}}$, which is a typical rate in nonparametric statistics [a].
>
> [a] Nonparametric regression using deep neural networks with ReLU activation function
>
> > **Comment** Theorem 1's rate is a mixture of asymptotic and finite bounds. Asymptotic for the source task and finite for the target task.
>
> **Response** Thank you for your valuable comment. As you mentioned, our result does not provide a complete non-asymptotic upper bound for $n_s$. The primary challenge lies in the difficulty of determining a specific sample size $n_s$ based on the discussion from Lines 1000-1003 in our manuscript as the formulation of it is quite complex. Therefore, we adopt asymptotic analysis to circumvent this issue. This approach does not hinder our ability to further explore the finite-sample bound, provided that $n_s$ is sufficiently large. Thank you for your understanding.
>
> > **Comment** Although perhaps implicit in context I think the reader would be aided in learning that ReLU neural networks for classification is the model used.
>
> **Response** Thanks for your valuable advice. We have revised our manuscript according to it.
>
> > **Comment** Line 150, the authors say that $m$ could be infinite, yet later we consider a uniform density over all augmentations. Could the authors clarify this? It seems to me that a covering based argument over continuous augmentation classes would be more direct. Can the authors comment on the difficulty or feasibility of adapting their argument to this setting? In order to approximate some augmentation classes we require a cover that is exponential in dimension. e.g., rotational transformations. Thus, there can be additional dimensional dependence for augmentations commonly used. How does this affect the rate?
>
> **Response** Thank you for your question. We adopted a uniform distribution on a finite set $\mathcal{A}$, which refers to a discrete distribution, as we considered the case where $m$ is finite. In fact, it is reasonable since only a finite number of augmentations will be used in numerical experiments or practical applications.
>
> > **Comment** Is the $1/\min_k\sqrt{n_t(k)}$ natural here should we expect something that is more balanced as as we get more data for all classes except for one.
>
> **Response** Exactly! In the scenario you mentioned, the model's performance on the exclusive class will be very poor, but it performs well on the others. This makes a lot of sense in the context of meta-learning. In fact, this case can be also captured by our theory. Thanks for your valuable questions.
>
> > **Comment** Biggest concern.
>
> **Response** Thank you for your question. As you mentioned, it is incorrect that $\mathbb{E}\_{\widetilde{D}_{s}} \sup\_{G \in \widehat{\mathcal{G}}(f)} \widehat{\mathcal{R}}(f, G) = \mathcal{R}(f, G)$. We only have $\mathcal{R}(f,G)=\mathbb{E}\_{\widetilde{D}\_{s}}\widehat{\mathcal{R}}(f,G)$. As we discussed in Section 3.2, this unbiased property reduces the gap between the population risk and the empirical risk. It is sufficient to establish our theoretical analysis with the help of this unbiasedness.
>
> > **Comment** Are ReLU networks a Holder class? As their partials may not exist everywhere.
>
> **Response** The ReLU networks used in this work belong to a H"older's class. Since ReLU networks we used is Lipschitz continuous, they are H{\"o}lder continuous with index $\alpha=1$. It worth noting that we assume the target function $f^{*}$ belongs to a H{\"o}lder's class, instead of ReLU neural networks.
>
> If you have any questions, please feel free to further contact us. Thank you once again for your careful reading, professional suggestions, and inspiring recognitions.

---

> > ### Comment · Reviewer_2wB8 · 2025-08-04
> >
> > Thank you for the thoughtful and detailed reply.
> >
> > A notation clarification, to be clear w.r.t. Equation (4), we have that
> > $\arg \min _{f \in \mathcal{F}} \max _{G \in \widehat{\mathcal{G}}(f)} \widehat{\mathcal{L}}(f, G)$
> >
> > is equal to
> >
> > $\arg\min_{f \in \mathcal{F}} \max_{G \in \widehat{\mathcal{G}}(f)}\left \\{ \widehat{\mathcal{L}}_{\text {align}}(f)+\lambda \widehat{\mathcal{R}}(f, G) \right \\}$?

---

> > > ### Author Response · Authors · 2025-08-05
> > > **Rebuttal by Authors**
> > >
> > > Exactly. Thanks for your valuable discussions again.

---

> > > > ### Comment · Area_Chair_wSST · 2025-08-05
> > > >
> > > > Dear Reviewer 2wB8,
> > > >
> > > > Thank you for initiating the discussion with authors.
> > > >
> > > > However, at this stage, it seems that you have only made comments regarding notation in response to the authors' rebuttal comments. Would you please respond to the authors with your opinion on their rebuttal comments? Do you think their rebuttal adequately addresses your initial concerns? If you still have concerns even after considering the rebuttal comments, please express them specifically to the authors.
> > > >
> > > > If you have any difficulty in starting the discussion, please let us know.
> > > >
> > > > Sincerely,
> > > > AC

---

> > > > ### Comment · Reviewer_2wB8 · 2025-08-06
> > > >
> > > > Thank you for the clarifying remark. Taken with your originally reply, this resolves my main concern.
> > > >
> > > > As the authors provided to me, providing remarks the for interpreting the rates would aid the reader.
> > > >
> > > > Also, given the authors reply, a remark should also be given for the justification of the asymptotic term given the mismatch in the rate.
> > > >
> > > > A, I believe, final question, how does the rate change as $m$ increases? Although I agree that in practice $m$ is finite, it still may be large.

---

> > > > > ### Author Response · Authors · 2025-08-07
> > > > > **Rebuttal by Authors**
> > > > >
> > > > > Thank you for your one more professional question. If we consider the case where $m$ increases with $n_s$, according to our theoretical guarantee, we have following conclusions:
> > > > > $$\mathbb{E}\_{\widetilde{D}\_s, \widetilde{D}\_t}\\{\mathrm{Err}(Q\_{\hat{f}\_{n\_s}})\\} \lesssim (1 - \sigma\_s^{(n\_s)}) + m^2n\_s^{-\frac{\min\\{\alpha, \epsilon\_{\mathcal{A}}, \epsilon\_{\mathrm{ds}}\\}}{32(\alpha + d + 1)}} + \frac{1}{\min\_k\sqrt{n\_t(k)}}.
> > > > > $$
> > > > > - First of all, as long as $m \lesssim n\_s^{\frac{\min\\{\alpha, \epsilon\_\mathcal{A}, \epsilon\_{\mathrm{ds}}\\}}{64(\alpha + d + 1)}}$, the asymptotic property can be guaranteed. As the rate of $m$ increases, the final rate concerning $n\_s$ will be slower. This result makes sense, as the larger $m$ is, the more potential knowledge is needed to learn from the data, the requirements for data size get therefore stricter to ensure a misclassification rate of the same level.
> > > > > - For instance, if we set $m \asymp n\_s^{\frac{\min\\{\alpha, \epsilon\_\mathcal{A}, \epsilon\_{\mathrm{ds}}\\}}{64(\alpha + d + 1)}}$, the final rate regarding $n\_s$ becomes $n\_s^{-\frac{\min\\{\alpha, \epsilon\_\mathcal{A}, \epsilon\_{\mathrm{ds}}\\}}{64(\alpha + d + 1)}}$.
> > > > >
> > > > > Thank you for your invaluable suggestions. We highly value your professional opinion and promise to incorporate them into our revised manuscript. However, NeurIPS 2025 does not allow authors to upload our revised manuscripts at this stage. Thank you for your understanding.

---

> > > > > > ### Comment · Reviewer_2wB8 · 2025-08-07
> > > > > >
> > > > > > Thank you for taking the time to answer my questions. The discussion regarding growth rates of $m$, I believe, makes sense. Although wasn't clear partly due to the asymptotic nature of the bound. Taken our discussion together, I'll raise my score.

---

> > > > > > > ### Author Response · Authors · 2025-08-07
> > > > > > > **Rebuttal by Authors**
> > > > > > >
> > > > > > > Thank you for your effort in reviewing our work. We feel inspired by your recognition and are fortunate to engage in such a professional discussion at this stage.

---

### Official Review · Reviewer_N15g · 2025-07-23

**Clarity:** 2
**Significance:** 3
**Originality:** 3
**Rating:** 4
**Confidence:** 4

**Summary:**

This paper presents an unbiased adversarial self-supervised representation learning framework (ASSRL) that removes the estimation bias of covariance regularization through a min–max formulation. It provides end-to-end risk bounds for transfer and few-shot settings and achieves consistent improvements on several benchmarks.

**Questions:**

See weakness.

**Ethical Concerns:**

["NO or VERY MINOR ethics concerns only"]

**Final Justification:**

Taking everything into consideration, I tend to increase the score. I appreciate the author's efforts and exploration in addressing this valuable research question. However, it should be acknowledged that the current conclusions and phenomena have very limited impact on real applications. I am excited to see more progresses regarding this issue in the future.

**Limitations:**

See weakness.

**Quality:**

2

**Strengths And Weaknesses:**

Strengths:
1. The paper offers the first end-to-end risk bound for non-contrastive self-supervised learning under limited data and distribution shift, filling a gap in the literature.

2. By leveraging adversarial training, the method corrects the bias inherent in mini-batch estimation, avoiding the cumulative drift observed in Barlow Twins and WMSE.

Weaknesses:
1. The maximization step is realized by directly setting G equal to the empirical estimate, rather than through a dynamic min–max game. The main paper does not clarify this equivalence, which may lead readers to overinterpret the “adversarial” aspect. In practice, the procedure is not truly adversarial.

2. The empirical improvements, while consistent, are modest. Table 2 should include additional recent decorrelation-based non-contrastive methods, notably VICReg [a] and LogDet [b]. The reported accuracies slightly exceed VICReg but are below LogDet. Moreover, more recent contrastive and non-contrastive baselines should be included.
 [a] VICReg: Variance-Invariance-Covariance Regularization for Self-Supervised Learning, ICLR 2022.
 [b] Geometric View of Soft Decorrelation in Self-Supervised Learning, KDD 2024.

3. Empirical analyses are insufficient.
1）The claim that “ASSRL-learned representations cluster downstream data by class” should be supported by visualizations or quantitative metrics.
2）Assumptions 4 and 5 depend on parameters such as  $\sigma_{s}^{(n)},\sigma_{t}^{(n)},\delta_{s}^{(n)},\delta_{t}^{(n)}$ for augmentations and $\epsilon_{\mathrm{ds}}$ for domain shift. These should be measured in experiments and toy examples to validate the assumptions.

4. The theoretical setting does not fully align with the experiments. The required network width and depth, sample sizes, augmentation constraints, and downstream domain settings are stricter than those used in practice. Assumptions 4 and 5 are particularly stringent and difficult to verify. Additionally, a lower bound on the error is absent. These limitations hinder the theory’s direct applicability to implementation and engineering.

---

> ### Author Rebuttal · Authors · 2025-07-31
>
> We thank you for your thorough review of our manuscript and for your constructive suggestions. We have revised our manuscript by incorporating all of your suggestions. The revised version is much strengthened. Our point-by-point responses to your comments are given below.
>
> > **Comment** The maximization step is realized by directly setting G equal to the empirical estimate, rather than through a dynamic min–max game. The main paper does not clarify this equivalence, which may lead readers to overinterpret the “adversarial” aspect. In practice, the procedure is not truly adversarial.
>
> **Response** Thank you for your feedback. Our algorithm is still within a min-max framework (thus called "adversarial"), while the maximization step can be solved explicitly. Hence we directly set $G$ equal to the empirical estimate, which is the analytical solution to the inner maximization step.
>
> > **Comment** The empirical improvements, while consistent, are modest. Table 2 should include additional recent decorrelation-based non-contrastive methods, notably VICReg [a] and LogDet [b]. The reported accuracies slightly exceed VICReg but are below LogDet. Moreover, more recent contrastive and non-contrastive baselines should be included.  [a] VICReg: Variance-Invariance-Covariance Regularization for Self-Supervised Learning, ICLR 2022.  [b] Geometric View of Soft Decorrelation in Self-Supervised Learning, KDD 2024.
>
> **Response** Thank you for your valuable feedback. We highly value your professional suggestions. We notice that the paper you mentioned didn't provide their implementations. Therefore, we try to implement the methods your mentioned ([a], [b]) with the same setting as ours and successfully get The preliminary results shown as follows:
>
> | Method | CIFAR-10 | CIFAR-10 | CIFAR-100 | CIFAR-100 |  Tiny ImageNet | Tiny ImageNet |
> | ---- | ----- | ----- | ----- | ----- | ---- | ----- |
> |  | Linear | $k$-nn | Linear | $k$-nn | Linear | $k$-nn |
> | VICReg | 91.23 | 89.15 | 67.61 | 57.04 | 48.55 | 35.62 |
> | LogDet | 92.47 | 90.19 | 67.32 | 57.56 | 49.13 | 35.78 |
> | ASSRL | **93.01** | **90.97** | **68.94** |  **58.50** | **50.21** | **37.40** |
>
> Additionally, we have achieved improvements of 6\% to 12\% compared to the biased methods.
>
> > **Comment** The claim that “ASSRL-learned representations cluster downstream data by class” should be supported by visualizations or quantitative metrics
>
> **Response** To provide empirical evidence for our claim, we follow the definition of divergence term from [c], which is expressed as the average of $\Big(\frac{1}{\vert C_t(i)\vert}\sum_{x_t \in C_t(i)}f(x_t)\Big)^{\top}\Big(\frac{1}{\vert C_t(j)\vert}\sum_{x_t \in C_t(j)}f(x_t)\Big)$ among all $i \neq j$. This quantity measures the divergence between representations of data in different latent classes. Further, lower divergence term implies more divergence. We compute the divergence term of the representations learned by our method in CIFAR-10, resulting in a value of 0.231, which reveals that our representations cluster downstream data by class. Moreover, this quantity is smaller than the corresponding values of the other methods reported in [c].
>
> [c] Towards the Generalization of Contrastive Self-Supervised Learning, ICLR 2023.
>
> > **Comment** Assumptions 4 and 5 depend on parameters such as $\sigma_s^{(n)}, \sigma_t^{(n)}, \sigma_s^{(n)}, \sigma_t^{(n)}$ for augmentations and $\epsilon_{\mathrm{ds}}$ for domain shift. These should be measured in experiments and toy examples to validate the assumptions. Assumptions 4 and 5 are particularly stringent and difficult to verify.
>
> **Response** Thank you for your comments. We have illustrated Assumption 4 in Appendix F.2 of our initial manuscript with a toy example. Additionally, the corresponding explanation for Assumption 5 can be found in Appendix F.3. However, in practical applications, these assumptions are challenging to verify since we cannot access the image distribution. Thank you for your understanding.
>
> > **Comment** The required network width and depth, sample sizes, augmentation constraints, and downstream domain settings are stricter than those used in practice. Assumptions 4 and 5 are particularly stringent and difficult to verify.
>
> **Response** Thank you for your comments. As you mentioned, our assumptions may indeed be stricter than those typically used in practice. However, we want to emphasize that the width and depth represent only a sufficient but not necessary condition. Meanwhile, the sample sizes serve merely as a theoretical guarantee to illustrate the effectiveness of our proposed method. Additionally, the augmentation constraints provide insight into the types of augmentations we require. Overall, from a theoretical perspective, it is common to explore theoretical understanding in an ideal scenario. Thank you for your understanding.
>
> > **Comment** Additionally, a lower bound on the error is absent.
>
> **Response** Thank you for your feedback. The mini-max lower bound of self-supervised contrastive learning remains an open problem. It is undoubtedly a valuable and promising area for future research.
>
> Thank you once again for your careful reading, professional suggestions, and inspiring recognitions. If you have any questions, please feel free to contact us.

---

> > ### Comment · Reviewer_N15g · 2025-08-07
> > **Thanks for your reply.**
> >
> > Taking everything into consideration, I tend to increase the score. I appreciate the author's efforts and exploration in addressing this valuable research question. However, it must be acknowledged that the current conclusions and phenomena have very limited impact on real applications. I am excited to see more progresses regarding this issue in the future.
> >
> > Comment 1:
> > In my point of view, the maximization step is naturally exist and there is no need to “direct set G to the empirical estimate”. As such, I can not agree there is a proposed step used for explicitly maximization. The name of the method seems to be overdecorated.
> >
> > Comment 2:
> > Could you specify the difference between the setting of results reported in LogDet[b]. As I understand, compared with results in LogDet[b], your results, e.g., 93.01 of Linear on CIFAR-10, 68.94 of Linear on CIFAR-100, still not satisfied enough. Additionally, the results are lack of direct comparison with existing theories, e.g., results on ImageNet and ImageNet-100.
> >
> > Comment 3:
> > The divergence value is meaningful in the response. Actually, it also needs to be compared with other methods.
> >
> > Comment 4-6
> > Before pointing out these weaknesses, I had already read the author's discussion on Assumption 4 and 5. It should be admit the authors have try to derive the meaningful theory by leveraging some existing assumptions. I appreciate the authors efforts. However, in my point of view, the guidance of the proposed theory for practical implementation in real-world is limited. Because the theoretical guarantee rely on these strict assumptions that the current advanced learning paradigms can not satisfy explicitly.

---

> ### Author Response · Authors · 2025-08-07
> **Rebuttal by Authors**
>
> > **Comment** Taking everything into consideration, I tend to increase the score.
>
> **Response** We feel incredibly grateful and inspired by your recognition. We would like to express our appreciation once again for your professional reviews and invaluable suggestions.
>
> > **Comment** In my point of view, the maximization step is naturally exist and there is no need to “direct set G to the empirical estimate”. As such, I can not agree there is a proposed step used for explicitly maximization. The name of the method seems to be overdecorated.
>
> **Response** We have deeply understood your concerns and are considering revising our name according to your suggestions. However, we are unsure if this is permitted for NeurIPS 2025. If it is not allowed, we will emphasize this in the main text of our revised manuscript. Thank you.
>
> > **Comment** Could you specify the difference between the setting of results reported in LogDet[b]. As I understand, compared with results in LogDet[b], your results, e.g., 93.01 of Linear on CIFAR-10, 68.94 of Linear on CIFAR-100, still not satisfied enough.
>
> **Response** Thank you for your thoughtful question. Unfortunately, the authors did not provide a link to their implementation and specific parameters in their paper [b]. While there is a GitHub link related to logDet on the author's homepage, it appears that the practical code corresponds to other papers rather than [b].
>
> According to the limited information provided in the paper, they mentioned, 'we employ both the same **logistic regression classifier**.' In contrast, our implementation uses a **linear classifier** to evaluate the classification error, which is a standard paradigm in this field. Thank you for your understanding.
>
> > **Comment** Additionally, the results are lack of direct comparison with existing theories, e.g., results on ImageNet and ImageNet-100.
>
> **Response** We highly value your suggestions and believe that they are indeed helpful in improving the quality of our paper. We are working hard to implement them and thank you for your understanding.
>
>
> > **Comment** The divergence value is meaningful in the response. Actually, it also needs to be compared with other methods.
>
> **Response**
> Thank you for your professional suggestion, we individually calculate the same quantity of other methods, the primary result is shown as following table:
>
> | Method                  | Barlow Twins | Beyond Separability | VICReg | LogDet | ASSRL      |
> |-------------------------|--------------|---------------------|--------|--------|------------|
> | Divergence              | 0.379        | 0.383               | 0.255  | 0.261  | **0.231**  |
>
>
> > **Comment** Comment 4-6 Before pointing out these weaknesses, I had already read the author's discussion on Assumption 4 and 5. It should be admit the authors have try to derive the meaningful theory by leveraging some existing assumptions. I appreciate the authors efforts. However, in my point of view, the guidance of the proposed theory for practical implementation in real-world is limited. Because the theoretical guarantee rely on these strict assumptions that the current advanced learning paradigms can not satisfy explicitly.
>
> **Response** Thank you for your professional comments. We attach great importance to your opinions. The key reason for the strong assumptions is the overly rough estimation of Lipschitz constants in actual networks. To the best of our knowledge, there have been numerous studies focused on improving the precision of Lipschitz constants. We believe that this theoretical work can help us improve our results further. Thank you once again for your suggestions.

---

### Decision · Program_Chairs · 2025-09-17

**Decision:**

Accept (poster)

**Comment:**

This paper introduces adversarial self-supervised representation learning (ASSRL) to address the biased sample risk inherent in SSL that aims to align the covariance/correlation matrix to a diagonal matrix. By adopting the adversarial learning idea of performing representation learning via solving a min-max optimization, the paper provides theoretical guarantees on generalization error in transfer learning scenarios under misspecified settings.

Reviewer evaluations are split around the borderline. While the rigor of the theoretical claims is largely appreciated, concerns have been raised about the real-world plausibility of the assumptions used in the theory, the practical implications of the proposed algorithm, and the settings and evaluation methodology of the experiments. In particular, although the direct performance comparisons with existing methods are not necessarily presented in a convincing manner, I judge the authors’ responses to these points to be reasonable since the main contribution of this paper is on the theoretical side. I believe the work will attract interest of the contrastive learning community, so I recommend acceptance.